# A new species of *Brachycephalus* (Anura: Brachycephalidae) from Serra do Quiriri, northeastern Santa Catarina state, southern Brazil, with a review of the diagnosis among species of the *B. pernix* group and proposed conservation measures

**Marcos R. Bornschein**[1,2], **Marcio R. Pie**[2,3,4*], **Júnior Nadaline**[2,4], **André E. Confetti**[4,5], **David C. Blackburn**[5], **Edward L. Stanley**[5], **Renata de Britto Mari**[1], **Gabriel Silveira Alves**[1], **Giovanna Sandretti-Silva**[1,2,6], **Felipe Farias de Andrade Lima**[1], **Luiz F. Ribeiro**[2,7]

**1** Departamento de Ciências Biológicas e Ambientais, Instituto de Biociências, Universidade Estadual Paulista (UNESP), São Vicente, São Paulo, Brazil, **2** Mater Natura – Instituto de Estudos Ambientais, Curitiba, Paraná, Brazil, **3** Biology Department, Edge Hill University, Ormskirk, Lancashire, United Kingdom, **4** Departamento de Zoologia, Universidade Federal do Paraná, Curitiba, Paraná, Brazil, **5** Florida Museum of Natural History, University of Florida, Gainesville, Florida, United States of America, **6** Institute of Biology and Environmental Sciences, School of Mathematics and Science, Carl von Ossietzky Universität Oldenburg, Oldenburg, Lower Saxony, Germany, **7** Departamento de Genética, Universidade Federal do Paraná, Curitiba, Brazil

\* marcio.pie@gmail.com

## Abstract

*Brachycephalus* are miniaturized diurnal frogs inhabiting the leaf litter of the Brazilian Atlantic Forest, mainly in montane areas. The genus includes 42 currently recognized species, 35 of which being described since 2000. This study describes a new species of *Brachycephalus* from the *B. pernix* species group discovered at Serra do Quiriri, Santa Catarina, Brazil. Morphological and acoustic comparisons were made with other species in the species group, and high-resolution computed tomography was used for osteological examination. The phylogenetic position was based on partitioned Bayesian analysis of mitochondrial (16S rRNA) and nuclear DNA sequences (β–fibrinogen, ribosomal Protein L3, and tyrosinase exon 1). We collected 32 individuals and recorded 13 calls of the new species. It is distinguished by 18 characters including snout–vent length 8.9–11.3 mm for males and 11.7–13.4 mm for females, general bright orange coloration of the body with small green and brown irregular points, and advertisement call including note groups (two notes per group, with 1–4 pulses per note). Phylogenetic data indicate that the new species is closely related to *B. auroguttatus* and *B. quiririensis*, which also occur at Serra do Quiriri. A review of diagnoses among species of the *B. pernix* group is provided. We propose classifying the new species as Least Concern. Serra do Quiriri experienced semi-arid periods in the Quaternary, with forests likely occurring at lower altitudes. As the climate became

**Data availability statement:** All relevant data are available in the paper and its Supporting Information files. These files are also archived at the digital repository DARE (Institutional Research Data Repository of the Carl von Ossietzky Universität Oldenburg) at the following DOI: https://doi.org/10.57782/95CKKG.

**Funding:** The field work was funded by Fundação Grupo Boticário de Proteção à Natureza (through grant 1149_20191) through project conducted by Mater Natura – Instituto de Estudos Ambientais. MRP received a grant from CNPq/MCT (301636/2016-8). GS-S received grant from São Paulo Research Foundation (FAPESP; processes #2022/04847-7 and # 2023/09718-3). There was no additional external funding received for this study. The funders had no role in study design, data collection and analysis, decision to publish, or preparation of the manuscript.

**Competing interests:** The authors have declared that no competing interests exist.

wetter, these forests expanded upward as cloud forests, forming patches amidst grasslands, leading to speciation by allopatry (microrefugia) of *B. quiririensis*, *B. auroguttatus*, and the new species. This process continues, with recent observations of *Brachycephalus* colonizing newly formed cloud forests at high altitudes. We propose the creation of the Refúgio de Vida Silvestre (RVS) Serra do Quiriri to protect this and other endemic species, without requiring government acquisition of private land.

## Introduction

*Brachycephalus* species are miniaturized and, together with several other anuran lineages, are among the smallest adult tetrapods [1]. They inhabit the leaf litter of the eastern Atlantic Forest, from northeastern to southern Brazil, with some species having very restricted geographical distribution and, therefore, being prone to extinction [2]. Presumably, all species exhibit direct development, without an aquatic larval stage [3]. The species are mostly diurnal, and although they are easy to hear calling in high densities, they are difficult to see and locate [4].

*Brachycephalus* vary from brightly colored, such as *B. alipioi* [5] and *B. mirissimus* [6], to cryptically colored, such as *B. didactylus* [7] and *B. curupira* [4], and the bright coloration seems to have been lost at least twice in the genus [4,8]. Other intriguing discoveries in recent years involve the recognition that males of *B. ephippium* and *B. pitanga* seem unable to hear their own calls [9], and many species have distinctive features such as bone fluorescence (e.g., [10]) or specific skin morphology associated to different climate conditions within the Atlantic Forest mountains, which can even give a white color to certain parts of the body [11].

During most of the past two centuries, the genus *Brachycephalus* Fitzinger, 1826 encompassed only a few species. Remarkably, 35 of the 42 currently recognized species have been described since 2000 [12]. A previous study [13] divided *Brachycephalus* into three phenetic groups: the *B. ephippium*, *B. didactylus*, and *B. pernix* species groups, now with 15, seven, and 20 species, respectively (see below). The species in each group show consistent morphological and biogeographical differences. The species of the *B. ephippium* group are found in Southeastern Brazil (states of São Paulo, Rio de Janeiro, Minas Gerais, and Espírito Santo), and show dermal co-ossification and larger body sizes (up to 18.9 mm in snout-vent length) in comparison with members of the *B. pernix* group (up to 15.8 mm), which are found in Southeastern and Southern Brazil (states of São Paulo, Paraná, and Santa Catarina) [2,13–15] and lack dermal co-ossification. In contrast, members of the *B. didactylus* species group have broad distributions [2,14], a "leptodactyliform" body shape (as opposed to the "bufoniform" body shape found in the other two species groups), and absence of dermal co-ossification. Molecular genetic data support the monophyly of the *B. ephippium* and *B. pernix* species groups [16], but current evidence strongly indicates that *B. didactylus* species group is polyphyletic [17]. For this reason, the species in this group will be referred to by the common name "flea toads".

As more localities were surveyed and new species were uncovered, the known distribution of the genus expanded and revealed a recurring pattern of small geographic distributions and largely allopatric populations, with most species being distributed in only one or a few adjacent mountaintops [2,14,18]. The prevalence of such restricted ranges has profound consequences for their conservation, with many described species already threatened [2,15]. Therefore, the tasks of uncovering the diversity in the genus and searching for new populations should receive high priority to ensure their long-term conservation. In this study, we describe a new species of the *B. pernix* species group from the Serra do Quiriri, north-eastern state of Santa Catarina, Brazil. This description is accompanied by a comprehensive review of previously used diagnostic characteristics and additional specimens recently obtained in the field. Conservation measures are proposed to protect the new species and other anurans endemic to high altitudes of the Serra do Quiriri.

## Materials and methods

### Study area

Serra do Quiriri (*c.* 26°00'S, 48°57'W) is a mountain range located on the border of the states of Paraná and Santa Catarina, southern Brazil, which reaches a maximum altitude of 1,507 m above sea level (a.s.l.). The dominant vegetation is the highland grasslands (see the black polygons in Fig 1 of Nadaline *et al.* [19]), which are bordered by cloud forest (see [20]), while this forest transitions to montane forests on the slopes with lower altitudes. The Serra do Quiriri is also inhabited by *B. quiririensis* and *B. auroguttatus* (see Fig 1 of Bornschein *et al.* [21]), the former known from two localities and the second only from the type locality [2]. Towards the north, Serra do Quiriri is separated from Serra do Araçatuba, in the state of Paraná, by a valley 1,080 m a.s.l. In the Serra do Araçatuba, which reaches 1,656 m a.s.l., occurs *B. leopardus*, being known from two localities [2].

According to Köppen's climate classification, the eastern slopes of the Serra do Quiriri have a Cfa climate: a humid subtropical zone (C) with oceanic climate, without dry season (Cf), and with a hot summer (Cfa; [22]). At higher altitudes,

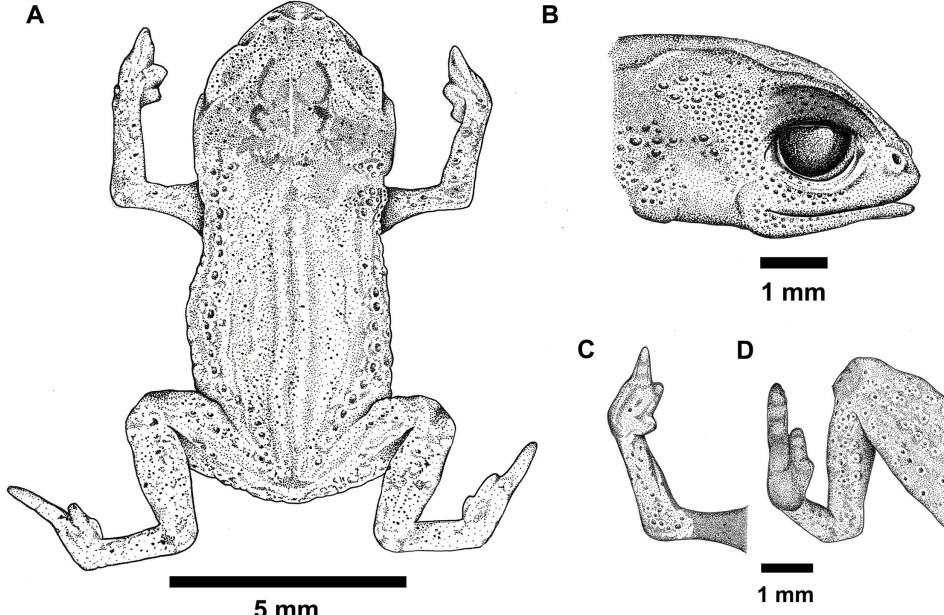

**Fig 1. Holotype of *Brachycephalus lulai* sp. nov. (MHNCI 11592), male.** (A) Dorsal view of the body. (B) Lateral view of the head. (C) Ventral view of right hand. (D) Ventral view of right foot. The specimen's image was projected using a stereomicroscope with a camera lucida, and the illustration was rendered in black ink using the pointillism technique Drawing by Verônica R. Apolônio.

the Serra do Quiriri exhibits a Cfb climate, characterized by a temperate summer [22]. The mean annual temperature ranges from 20–22 °C on the eastern slopes of the Serra do Quiriri to 12–14 °C at higher altitudes, with intermediate altitudes averaging between 16–18 °C [22]. Annual rainfall varies from 1,600–1,900 mm across most of the Serra do Quiriri, increasing to 1,900–2,200 mm in the highest areas [22].

## Specimen collection and processing

The Instituto Chico Mendes de Conservação da Biodiversidade (ICMBioc) granted the license to collect the specimens for this study (permission number #20416−2). No additional approval was necessary, given that the study follows the general guidelines for taxonomic work in herpetology. Populations of the new species were identified in the field through the recognition of the advertising call emitted by males. Specimens were collected through active search and during the day. The advertisement call aided the researchers in locating the males, whereas the females were collected haphazardly. Collection locations were recorded with coordinates using Garmin GPSmap 60CSx (DATUM WGS84) and the vegetation was classified according to the criteria of Veloso *et al.* [23]. Specimens collected in the field were placed alive in individual plastic bags, containing damp leaf litter. The plastic bags were placed inside Styrofoam boxes to maintain a mild temperature. The specimens were taken to the morphology laboratory, anesthetized and killed painlessly using 2% lidocaine hydrochloride. Each one was dissected to obtain a tissue sample, fixed with 10% formalin for 24 h and then preserved in 70% ethanol. All specimens, after being analyzed and measured, were deposited at the Museu de História Natural Capão da Imbuia (MHNCI), Curitiba, state of Paraná, Brazil. Other specimens of various congeneric species were analyzed, belonging to the following collections: Célio F. B. Haddad collection, Departamento de Zoologia, Universidade Estadual Paulista, Campus Rio Claro, state of São Paulo (CFBH); Coleção Herpetológica do Departamento de Zoologia, Universidade Federal do Paraná, Curitiba, state of Paraná (DZUP); Museu de História Natural Capão da Imbuia, state of Paraná (MHNCI); Museu de História Natural, Universidade Estadual de Campinas, Campinas, state of São Paulo (ZUEC); Museu de Zoologia da Universidade de São Paulo, São Paulo, state of São Paulo (MZUSP); and Museu Nacional, Rio de Janeiro, state of Rio de Janeiro (MNRJ). All specimens examined in the study are listed in S1 Appendix. Fifteen morphometric measurements [24] were used, according to Pie *et al.* [6], obtained using a micrometric eyepiece attached to a Zeiss Stemi 2000 stereomicroscope. These measurements are presented in millimeters (mm) and were taken by a single researcher (Júnior Nadaline). The sex of each specimen was determined based on the presence of the *linea masculinea* in males as described in Pie *et al.* [6].

We describe the skin texture of examined specimens of *Brachycephalus* as smooth, moderately rough, and densely rough (S1 Fig). Smooth texture refers to skin with small warts that are barely or not at all prominent, giving a homogeneously smooth appearance (S1A Fig). Moderately rough texture refers to skin with protruding warts that maintain spacing between them (S1B Fig). Lastly, densely rough texture refers to skin with prominently protruding warts that are densely packed, with little or no space between them (S1C Fig). We prefer not to create a fourth state of skin texture for a condition observed in individuals with few prominent warts on smooth skin, a condition we then characterize as smooth skin.

To produce the drawing of the holotype, a stereomicroscope equipped with a camera lucida was employed for direct observation of the specimen. Initially, a preliminary sketch was created using graphite. A black ink pen (Faber-Castell ECCO Pigment Gray 0.05 mm) with pointillism technique was then applied to achieve the final rendering. A millimeter scale was included adjacent to the specimen to ensure accurate proportions. Magnification levels ranging from 1x to 4.5x were utilized. Finally, the drawing was digitized. The design was executed by Verônica R. Apolônio.

## Computed tomography

High resolution computed tomography (CT) scans were produced for one paratype of *Brachycephalus* sp. nov. at LAMIR – Laboratório de Análise de Minerais e Rochas, Universidade Federal do Paraná, using a Skyscan 1172 desktop MicroCT

with a 100 kV X-ray tube following settings: 59 kV, 167 uA, a 1.4 s exposure time, four images per rotation and an image pixel size of 5.99 μm. Raw X-ray data were processed using NRecon reconstruction software (Micro Photonics Inc.) to produce a series of tomogram images. These MicroCT image stacks were then viewed, sectioned, measured, and analyzed using ITKSnap 3.2 and Meshlab 2020.12. Final figures were prepared with Adobe Photoshop and Illustrator (CS6; Adobe, San Jose, CA, USA).

## Skin histology

We examined three adult individuals from each locality where the new species was recorded. Two transverse skin cuts were taken from the medial and caudal regions in the dorsal surface of the body. Samples were dehydrated in increasing series from 70% to 100% ethanol, and cleared in xylol. Subsequently, samples were embedded in paraffin and cut into 5-μm thickness cross-sections, which were stained with the Hematoxylin and Eosin (HE) technique for measurement of the thickness of dermis and the von Kossa technique for measurement of the mineralized dermal layer (MDL) [11]. To obtain the measurements, each section was photomicrographed (200x magnification) and three thickness measurements (μm) were taken (one in each lateral portion of the body and one in the middle portion) using Image-Pro-Plus 3.0.1 software.

## Bioacoustics

Calls were obtained using the digital recorder Tascam DR-44WL with a Sennheiser ME 67/K6 microphone and digital recorder Sony PCM-D50 with a Sennheiser ME 66/K6 microphone, with sampling frequency rate of 44.1 kHz and 16-bit resolution. We deposited the recordings in the sound collections of the herpetological collection of the MHNCI. We evaluated calls under note-centered approach [25], as in Bornschein et al. [26–28] and Pie et al. [6]. We described the parameters based on Bornschein et al. [26,28] and Pie et al. [6], with modifications. For Bornschein et al. [28], we considered the parameter named attenuated notes. Attenuated notes generally precede the emission of notes with relatively higher sound energy/amplitude. In sum, our description of the call of the new species includes the following parameters, divided into the following sections: entire call (nine parameters), isolated notes (five parameters), note groups (eight parameters), and attenuated notes (seven parameters). Entire call: (1) call duration (s); (2) note rate (notes per minute; excluding attenuated notes); (3) number of notes per call (including attenuated notes); (4) number of notes per call (excluding attenuated notes); (5) note duration (s) (of isolated notes and notes within note groups); (6) number of pulses per notes (of isolated notes and notes of note groups); (7) note dominant frequency (kHz); (8) highest frequency (kHz); and (9) lowest frequency (kHz). Isolated notes: (10) duration of the call including only isolated notes (s) (when the calls present note groups among isolated notes, only the longest part with isolated notes was counted); (11) note rate of the call including only isolated notes (notes per minute); (12) number of isolated notes per call (excluding attenuated notes); (13) number of pulses per isolated notes; and (14) inter-note interval in isolated notes (s) (time from the end of one isolated note to the beginning of the next isolated note). Note groups: (15) duration of the call including only note groups (s) (only the longest part was counted, when the calls present isolated notes among note groups); (16) note rate of the call including only note groups (notes per minute; calculated only when at least two note groups were present); (17) number of note groups per call; (18) number of notes in each note group; (19) duration of note groups (s); (20) inter-note group interval (s) (time from the end of one note group to the beginning of the next note group); (21) inter-note interval within note groups (s) (time from the end of the first note to the beginning of the next note of the same note group); and (22) number of pulses per note in note groups. Attenuated notes: (23) number of notes per call associated with attenuated notes; (24) number of attenuated notes associated with each note of the call; (25) number of pulses in attenuated notes; (26) shortest interval between an attenuated note and its associated note (s); (27) attenuated note dominant frequency (kHz); (28) attenuated note highest frequency (kHz); and (29) attenuated

note lowest frequency (kHz). For rate calculation, we counted the time between the beginning of the first note to the beginning of the last note under consideration and the number of notes in this section [26].

Advertisement calls were analyzed in Raven Pro 1.6.3 with a 256-point Fast Fourier Transform and a 3-dB Filter bandwidth of 492 Hz, Hann window, and 50% overlap. Raven table selections were imported to the R environment using Rraven [29] and summarized using Summarytools [30]. Spectrograms for figures were generated using the Seewave package, v. 2.2.0 [31] of the R environment, v. 4.2.0 [32] using the same preset parameters as in Raven Pro. We examined calls from other 22 species in the sound collections of MHNCI, Coleção Audiovisual do Semiárido (CASA), Mossoró, state of Rio Grande do Norte, Brazil, and Fonoteca Neotropical Jacques Vielliard (FNJV), Campinas, state of São Paulo, Brazil, for comparative purposes (S2 Appendix). In the *B. pernix* group, there is still no available data on the calls of *B. mariaeterezae*.

## Molecular phylogeny

To determine the phylogenetic position of the new species, we analyzed DNA sequences from four paratypes (MHNCI 11596, MHNCI 11598–11600). The choice of species to be analyzed alongside the new species was based on previous work on the phylogeny of the group (i.e., [33,34]). Preliminary analyses indicated that the new species was closely related to a well-supported clade that included *B. auroguttatus*, *B. quiririensis*, *B. ferruginus*, *B. pernix*, and *B. pombali*. The monophyly of this clade was first suggested by Firkowski *et al.* [33] using Sanger sequencing and was later strongly supported by a phylogenomic analysis using UCEs by Pie *et al.* [34]. We therefore chose those species for exploring the phylogenetic position of the new species. The resulting tree was rooted according to the phylogenomic work by Pie *et al.* [34].

Whole genomic DNA was extracted using SPRI beads [35,36]. We amplified one mitochondrial locus (16S rRNA) and three nuclear loci (β-fibrinogen, ribosomal Protein L3, and tyrosinase exon 1) via polymerase chain reaction (PCR). Each PCR reaction had a total volume of 25 μL, containing 2 U AmpliTaq DNA polymerase, 1 × PCR buffer, 1.5 mM MgCl2, 0.5 mM dNTPs, 1.0 μM each primer, and approximately 40 ng of template DNA. The thermocycling protocol included an initial denaturation at 94 °C for 5 min, followed by 35 cycles of 94 °C for 1 min, 48–62 °C for 50 s, and 72 °C for 35–50 s, with a final extension at 72 °C for 5 min (S1 Table). PCR products were run on 1.5% agarose gels, and successful amplifications were purified using PEG 8000.

Sequencing reactions were prepared in a 10 μL volume, comprising 0.7 μL ABI Prism® BigDye™ v3.1 (Applied Biosystems Inc., Foster City, CA, USA), 1.0 μL 5 × buffer, one μL each primer (3.2 pmol), and about 30 ng of template DNA. The cycle sequencing conditions involved an initial denaturation at 96 °C for 1 min, followed by 35 cycles at 96 °C for 15 s, annealing at 50 °C for 15 s, and extension at 60 °C for 4 min. Each locus was sequenced in both directions using an ABI 3500 sequencer.

Sequences were aligned for each locus with those of other species in the *B. pernix* group (*B. auroguttatus*, *B. ferruginus*, *B. pombali*, *B. pernix*, and *B. quiririensis*, see [33]) using MUSCLE v3.8.31 [37] under default settings. The sequences were then concatenated, resulting in a final alignment length of 1828 bp. The optimal partitioning scheme was selected using PartitionFinder2 [38] based on the AICc criterion. The best scheme included six partitions: (1) HKY + I for 603 sites (16S and L3), (2) HKY + Γ for 133 sites (β-fibrinogen), (3) K80 model for 133 sites (β-fibrinogen), (4) F81 for 322 sites (β-fibrinogen and Tyr), (5) HKY + I + Γ for 448 sites (L3, L3, and Tyr), and (6) HKY + I + Γ model for 189 sites (Tyr).

A Bayesian phylogeny was inferred using MrBayes 3.2.7a [39]. Each analysis included two independent runs with four chains, run for 20 million generations, sampling every 1,000th generation. After confirming convergence of the chains, data sets were combined. Stationary distribution and effective sample sizes (ESS) for all parameters were assessed using Tracer v1.5 [40]. We discarded the initial 20% of trees as burn-in and used the remaining trees to estimate the maximum clade credibility consensus topology in TreeAnnotator v1.7.5 [41]. GenBank accession numbers are provided in S2 Table. Additional analyses using maximum likelihood provided qualitatively identical results and will not be shown for the sake of brevity.

## Conservation

We evaluated the conservation status of the new species and compared it with assessments of other species in the *B. pernix* group to better discuss and adjust recommended conservation measure. Although reassessing the conservation status of the others was not a primary objective, we also reassessed them because previous assessments for these species had some particularities in the spatial metrics based on their distribution pattern [2] that could complicate the consistency across taxa. To guide our evaluations, we followed the criteria and guidelines of the IUCN [42], IUCN SSC Red List Technical Working Group [43], and IUCN Standards and Petitions Committee [44].

We used the IUCN's geographical distribution criteria to assess the extinction risk of the species [42], as we lacked available data on population estimates and trends for the new species. Geographical distribution was delineated by constructing polygons based on the mapping of suitable habitat within and around the altitudinal range isolines of known records, in Google Earth Pro 7.3.6.10201, following and updating the mapping method of Bornschein *et al.* [2]. Polygons of potential habitat that extended several kilometers westward were not considered.

After this preliminary delimitation of the geographical distribution polygons, we connected their outermost points to generate an external polygon representing the extent of occurrence (EOO), using the Minimum Convex Polygon method [42–44] in Google Earth. Subsequently, we reviewed the entire EOO surface to identify and map additional potential habitats within the species' altitudinal range that may have been overlooked during the preliminary delimitation, also using Google Earth. If the resulting EOO was lower than the area of occupancy (AOO, see below), we equaled them to ensure consistency in the definition [43,44].

The resulting polygons of mapped habitat within the EOO represented the potential occupied habitat for the species, also referred to as limits of distribution, area of habitat, or field guide map [43]. To calculate the upper bound AOO, we overlapped these polygons of mapped habitat with a grid of 2 km x 2 km cells (= 4 km$^2$ of area), and then summed the area of overlapped cells [44] in QGIS 3.30.1. To calculate the lower bound AOO, we overlapped only the current records with the same grid [44]. We adjusted the grid to obtain the smallest number of occupied cells for each species under consideration [44], using the same position for upper and lower bound. We evaluated the range between the upper and lower bound values to potentially access the species as Data Deficient (DD) due to uncertainty [44]. For species in which we have not mapped the habitat because of the absence of an associated altitudinal range or because the mapping was unrealistic given the suitable habitat and altitude, we considered the lower bound AOO as the best estimate.

We assessed the IUCN Green Status of the new species and remaining species of the *B. pernix* group according to IUCN Green Status of Species Working Group [45], and IUCN [46], in order to compare the benefits of conservation measures. We considered the locations (*sensu* [42]) as conservation units for the assessment [46]. We used the conservation status (see above) in the unit to assess its states, and also the field information to assess its functionality. We used fine-resolution weights values (0, 1.5, 2.5, 3.5, 4.5, 5.5, 6.5, or 10) to better calculate the conservation impact [45]. Some species, despite their naturally small geographic distribution, do not seem to be threatened and occupy environments that are little or not at all disturbed, and, therefore, have been classified as functional at baseline levels in certain spatial units [45]. We assessed the Current and also Long-term potential Green Scores to calculate the Recovery Potential (percentage) of each species [46]. For the Long-term potential scenario, we considered stopping deforestation, promoting forest regeneration and mitigate wildfires as potential conservation actions. We created a new metric, called Strategic Weight (points), to allow numerical comparisons of species' conservation priorities (higher priorities for higher point values). It is calculated by adding the values we attribute to each conservation status (CR = 20; EN = 15; VU = 5; DD = 5; NT = 2; LC = 1) with the species' Recovery Potential value. Finally, we proposed the Potential Weight of a Conservation Unit (points) for a potential conservation unit to be created to protect each species or group of species with distributions close to each other, outside conservation units, adding Strategic Weight of each species occurring in this planned conservation unit.

## Nomenclatural acts

The electronic edition of this article conforms to the requirements of the amended International Code of Zoological Nomenclature, and hence the new names contained herein are available under that Code from the electronic edition of this article. This published work and the nomenclatural acts it contains have been registered in ZooBank, the online registration system for the ICZN. The ZooBank LSIDs (Life Science Identifiers) can be resolved and the associated information viewed through any standard web browser by appending the LSID to the prefix "http://zoobank.org/". The LSID for this publication is: urn:lsid:zoobank.org:pub:DD6AC76F-9D25-443B-A6B9-8AFC196DAD64. The electronic edition of this work was published in a journal with an ISSN, and has been archived and is available from the digital repository DARE (Institutional Research Data Repository of the Carl von Ossietzky Universität Oldenburg).

## Results

### *Brachycephalus lulai* sp. nov.

(Figs 1–4)

urn:lsid:zoobank.org:act:2A4472A8-55B9-4478-B65C-632D7B31B2A9

## Holotype

MHNCI 11592 (Figs 1–3), male, Pico Garuva (26°02'32"S, 48°53'27"W; 635 m a.s.l.), municipality of Garuva, state of Santa Catarina, southern Brazil, on 2 November 2016 by Luiz F. Ribeiro, André E. Confetti, Júnior Nadaline, and Marcio R. Pie.

## Paratopotypes

MHNCI 11593, male, and MHNCI 10847, female, both collected with the holotype; MHNCI 11594–8 and MHNCI 11600, males, and MHNCI 11599, female, 20 January 2017, Luiz F. Ribeiro, André E. Confetti, Júnior Nadaline, and Marcio R.

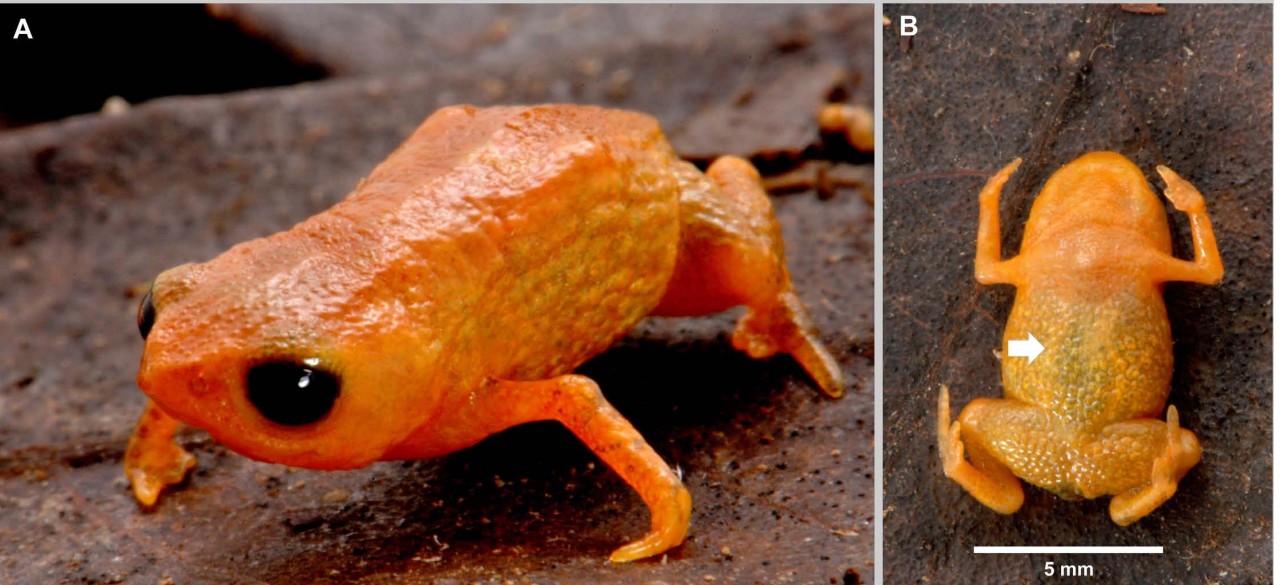

**Fig 2. Holotype of *Brachycephalus lulai* sp. nov. (MHNCI 11592), male, in life.** (A) Anterolateral view. (B) Ventral view. In B, white arrow indicates the presence of the *linea masculinea*. Photographs by Luiz F. Ribeiro.

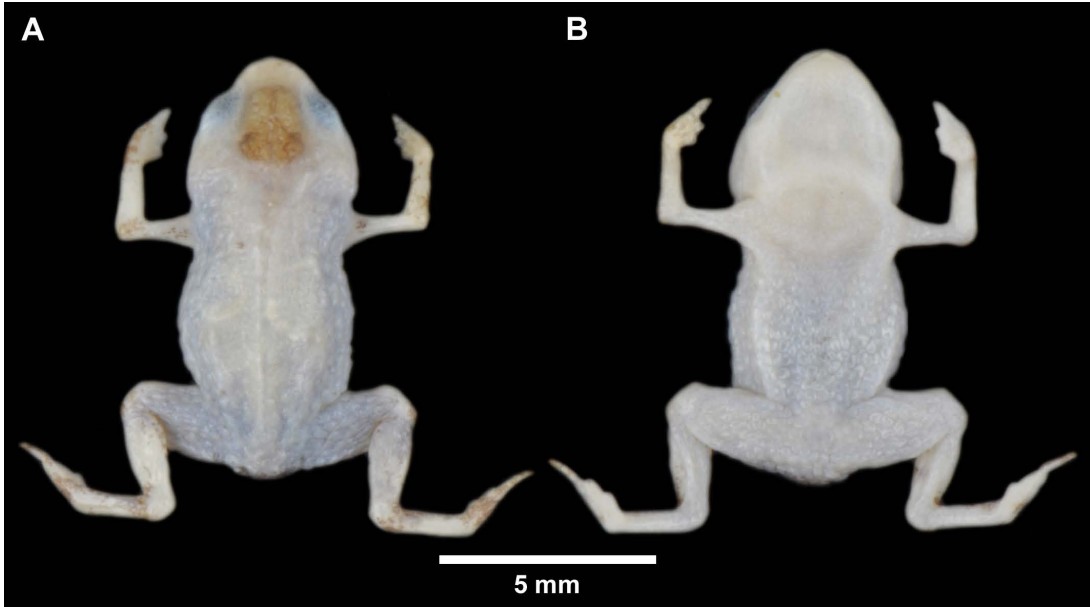

**Fig 3. Holotype of *Brachycephalus lulai* sp. nov. (MHNCI 11592), male, in preservative.** (A) Dorsal view. (B) Ventral view.

Pie; MHNCI 11601 and MHNCI 11603–10, males, and MHNCI 11602, female, 15 November 2018, Júnior Nadaline, André Luiz Ferreira Silva, and Marcio R. Pie. Specimens were collected in same locality as the holotype, between 635–990 m a.s.l.

## Paratypes

Specimens were collected at Monte Crista (26°05'33"S, 48°55'16"W), municipality of Garuva, state of Santa Catarina, southern Brazil, between 435–895 m a.s.l. MHNCI 11611 and MHNCI 11613, males, and MHNCI 11612, female, 15 November 2016, Luiz F. Ribeiro, Liliane Pires, and Marcio R. Pie; MHNCI 11614–5, MHNCI 11617, MHNCI 11619, and MHNCI 11621–2, males, and MHNCI 11616, MHNCI 11618, and MHNCI 11620, females, 26 February 2019, Júnior Nadaline, Philippe Fumaneri Teixeira, Tainara Thais Jory, and Marcio R. Pie.

## Diagnosis

*Brachycephalus lulai* sp. nov. is identified as a member of the *B. pernix* group (*sensu* Pie *et al.* [6] and Ribeiro *et al.* [13]) by having a bufoniform body shape (Figs 1–4), absence of dermal co–ossification, and presence of *linea masculinea* (Fig 2B). *Brachycephalus lulai* sp. nov. is distinguished from all of the species in the genus by the following combination of characters: 1) body shape bufoniform; 2) snout shape in dorsal view rounded; 3) SVL 8.9–11.3 mm for males and 11.7–13.4 mm for females (Table 1); 4) proportion of HL/SVL 32.5–41.4% for males and 31.8–34.9% for females; 5) presence of *linea masculinea* in males; 6) absence of dermal co-ossification; 7) dorsum with smooth texture; 8) sides of the body with densely rough texture; 9) tip of fingers I rounded, II rounded, and III pointed; 10) toe V externally absent; 11) outer metacarpal tubercle present; 12) iris black; 13) general color bright orange with small green irregular dots on sides of the body and belly and sometimes with brown dots on sides of the body; 14) general color in preservative pale cream with small light gray to dark gray irregular dots on sides of the body and belly and sometimes with dark spots on sides of the body; 15) advertisement call including note group; 16) two notes per note group; 17) advertisement call including attenuated notes; and 18) up to four pulses per note. The comparison with our analysis of other 20 species of the *B. pernix* group is

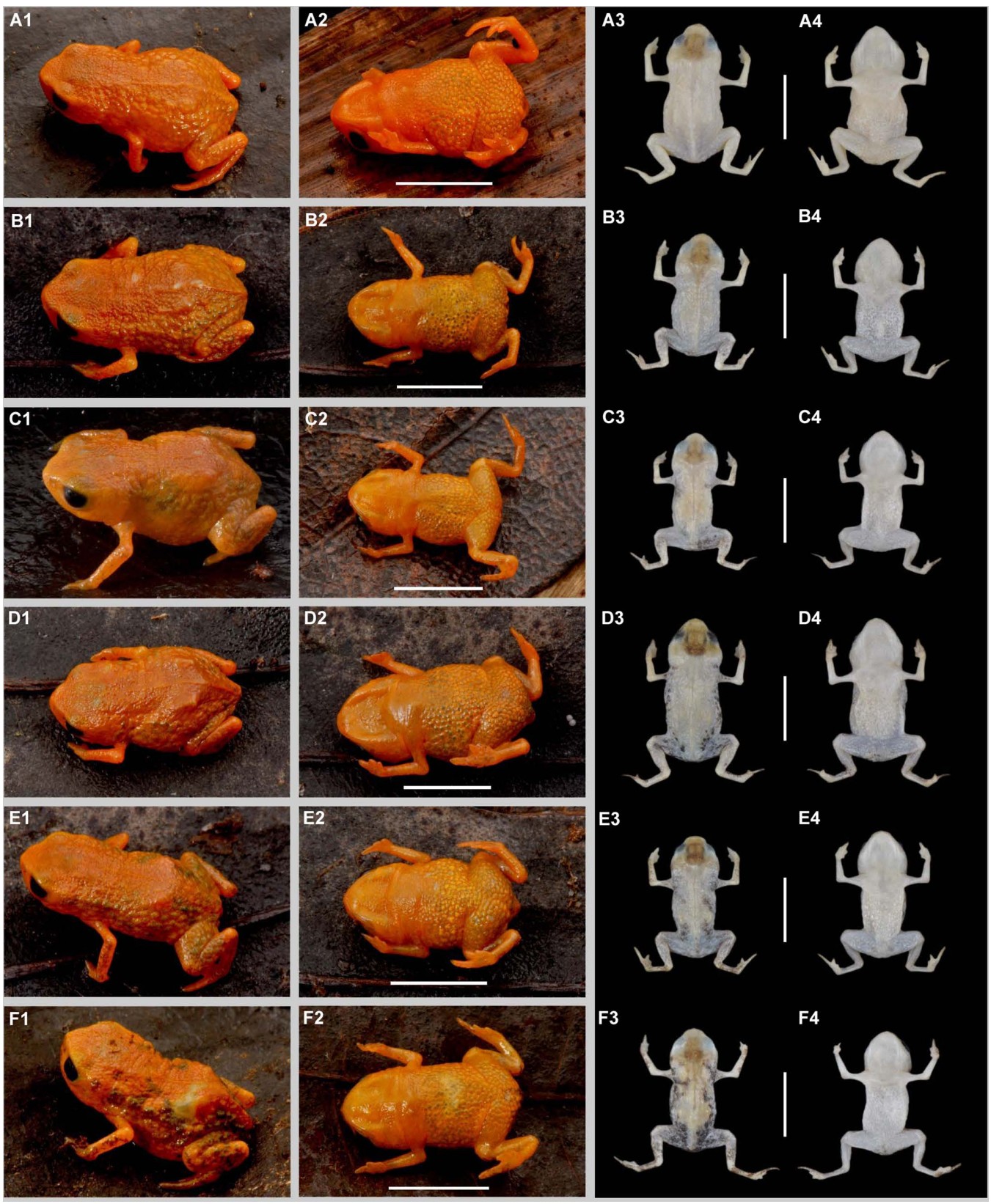

**Fig 4. Variation in coloration of paratypes of *Brachycephalus lulai* sp. nov. Column one and column three show specimens in dorsal view in life and in preservative, respectively.** Column two and column four show specimens in ventral view in life and in preservative, respectively. A1–A4 = MHNCI 11612. B1–B4 = MHNCI 11598. C1–C4 = MHNCI 11596. D1–D4 = MHNCI 11599. E1–E4 = MHNCI 11600. F1–F4 = MHNCI 11594. Scale bars equal 5 mm. Photographs by Luiz F. Ribeiro.

**Table 1. Measurements of the 15 variables (in mm) of the type series of *Brachycephalus lulai* sp. nov.**

| Variable | Males (N = 25) | | | Females (N = 7) | | |
|---|---|---|---|---|---|---|
| | Mean | SD | Range | Mean | SD | Range |
| SVL | 10.5 | 0.6 | 8.9–11.3 | 12.6 | 0.6 | 11.7–13.4 |
| HL | 3.6 | 0.2 | 3.0–4.1 | 4.1 | 0.2 | 3.9–4.3 |
| HW | 4.3 | 0.3 | 3.7–4.7 | 4.9 | 0.2 | 4.7–5.1 |
| ED | 1.2 | 0.1 | 1.0–1.4 | 1.3 | 0.1 | 1.1–1.5 |
| IOD | 2.2 | 0.1 | 1.8–2.4 | 2.4 | 0.2 | 2.2–2.7 |
| IND | 1.2 | 0.1 | 1.1–1.4 | 1.3 | 0.1 | 1.2–1.5 |
| EN | 0.6 | 0.1 | 0.6–0.8 | 0.7 | 0.0 | 0.7–0.8 |
| SL | 1.1 | 0.1 | 0.9–1.3 | 1.3 | 0.1 | 1.1–1.5 |
| UEW | 0.7 | 0.1 | 0.5–0.9 | 0.8 | 0.1 | 0.7–1.0 |
| FLL | 2.3 | 0.2 | 1.9–2.8 | 2.6 | 0.2 | 2.3–2.9 |
| HAL | 1.6 | 0.1 | 1.4–1.9 | 1.8 | 0.1 | 1.7–2.0 |
| THL | 3.9 | 0.2 | 3.4–4.3 | 4.5 | 0.2 | 4.3–4.9 |
| TL | 3.4 | 0.2 | 2.9–3.7 | 3.8 | 0.2 | 3.6–4.1 |
| TSL | 2.2 | 0.2 | 1.8–2.6 | 2.3 | 0.1 | 2.2–2.6 |
| FL | 2.6 | 0.2 | 2.2–3.0 | 2.9 | 0.3 | 2.6–3.3 |

Variable names and abbreviations are provided in the text. Abbreviation: SD = standard deviation.

summarized in Table 2, including the coloration in life based on specimens we collected, which sometimes differs from the color variation given in the original species description (Fig 5). *Brachycephalus tabuleiro* was assigned to the *B. pernix* group in its original description [47]. *Brachycephalus lulai* sp. nov. can be distinguished from the seven flea toad species (*B. clarissae*, *B. dacnis*, *B. didactylus*, *B. hermogenesi*, *B. pulex*, *B. puri*, *B. sulfuratus*) by having bufoniform body shape instead leptodactyliform body shape (but see below) and from 15 species of the *B. ephippium* group (*sensu* Ribeiro *et al.* [13]; *B. alipioi*, *B. bufonoides*, *B. crispus*, *B. darkside*, *B. ephippium*, *B. garbeanus*, *B. guarani*, *B. herculeus*, *B. ibitinga*, *B. margaritatus*, *B. nodoterga*, *B. pitanga*, *B. rotenbergae*, *B. toby*, *B. vertebralis*) by absence of dermal co-ossification [13,17,48–53]. *Brachycephalus rotenbergae*, *B. ibitinga*, and *B. herculeus* belong to the *B. ephippium* group due to the presence of dermal co-ossifications and a bufoniform body shape [49–51]. *Brachycephalus puri* was originally treated as a flea toad due to its leptodactyliform body shape and the absence of dermal co-ossification [48]. *Brachycephalus clarissae*, with its like-bufoniform body shape and lack of dermal co-ossification [17], was treated as a flea toad by Toledo *et al.* [54]. *Brachycephalus dacnis* was assigned to the *B. didactylus* group by Bornschein *et al.* [28], prior to its formal description by Toledo *et al.* [54], who treated it as a flea toad. We cannot provide distinguishing traits of *B. lulai* sp. nov. from *B. atelopoide* because its holotype is apparently missing [55] and there are no additional collections [56]. However, reanalysis of the original description [57] suggests that the holotype of *B. atelopoide* likely had a bufoniform body shape, dermal co-ossification, and warts [56]—the latter two features being absent in *B. lulai* sp. nov.

## Description of the holotype

Body shape bufoniform (Figs 1–4); head wider than long (proportion HW/HL = 120%); head length 34% of SVL; snout short (proportion SL/HL = 31%), with length almost equal to eye diameter (proportion SL/ED = 92%), rounded in dorsal and

**Table 2. Comparison of the diagnostic features of *Brachycephalus lulai* sp. nov. in relation to remaining species of *B. pernix* group. In bold we highlight the variables that are diagnostic in relation to *B. lulai* sp. nov. Sample size refers to the number of individuals in which the authors observed each variable or number of individuals from which one or more calls where recorded and analyzed by the authors.**

| Species | Measurement/ proportion | | | | Iris | Skin texture of the body | | Color | |
| | SVL male (mm) | SVL female (mm) | HL/SVL male (%) | HL/SVL female (%) | | Dorsum | Lateral | Alive | Preservative |
|---|---|---|---|---|---|---|---|---|---|
| *B. lulai* sp. nov. | 8.9–11.3 (N = 25) | 11.7–13.4 (N = 7) | 32.5–41.4 (N = 25) | 31.8–34.9 (N = 7) | Black (N = 32) | Smooth (N = 32) | Densely rough (N = 32) | Orange, orange with variable amount of small green rounded dots on sides of the body and belly and with or without brown spots on sides of the body (N = 32) | Pale cream with small dark dots on sides of the body and belly and with or without dark spots on sides of the body (N = 32) |
| *B. actaeus* | 10.4–11.3 (N = 9) | 11.7–12.2 (N = 3) | 30–34 (N = 9) | 33–34 (N = 3) | Black (N = 12) | Smooth (N = 12) | Densely rough (N = 12) | **Variable dorsum and sides of the body: these parts entirely dark green or brown; brown on middle dorsum with brown mixed orange on sides of the body, or dorsum brown mixed with orange and orange mixed with brown on sides of the body. Belly orange (N = 12)** | **Dark with pale cream belly or pale cream and dark on dorsum and belly (N = 12)** |
| *B. albo-lineatus* | 8.9–11.4 (N = 27) | 9.7–12.7 (N = 2) | 29–35 (N = 27) | 34–35 (N = 2) | Black (N = 29) | Smooth (N = 29) | Densely rough (N = 29) | **Green with orange patches on belly with or without a white stripe on middle dorsum (N = 29)** | **Dark with pale cream patches on belly with or without pale cream stripe on middle dorsum (N = 29)** |
| *B. auro-guttatus* | 10.0–12.6 (N = 10) | 11.3–13.6 (N = 4) | 31–36 (N = 10) | 32–37 (N = 4) | Black (N = 17) | **Densely rough (N = 17)** | Densely rough (N = 17) | **Orange and yellow mixed with dark on dorsum and belly, except in the head, entirely orange, or brown dorsum with orange head and stripe on middle dorsum and orange belly with scattered small brown dots (N = 17)** | **Pale cream and dark on dorsum and belly (N = 17)** |
| *B. boticario* | 10.9–11.7 (N = 5) | 10.6–12.2 (N = 3) | 29–32 (N = 5) | 31–31 (N = 3) | Black (N = 11) | **Densely rough (N = 11)** | Densely rough (N = 11) | **Dorsum dark with orange on head and along the vertebrae and orange belly (N = 11)** | **Dorsum dark; head, along the verte-brae, and belly pale cream (N = 11)** |
| *B. brunneus* | 8.7–10.6 (N = 16) | 9.2–11.4 (N = 10) | 31–35 (N = 16) | 31–35 (N = 10) | Black (N = 32) | Smooth (N = 32) | **Mod-erately rough (N = 32)** | **Brown with yellow patches on belly (N = 32)** | **Dark with pale cream patches on belly (N = 32)** |
| *B. coloratus* | 10.3–11.3 (N = 12) | 11.2–13.3 (N = 8) | 30–36 (N = 12) | 29–36 (N = 8) | Black (N = 20) | **Densely rough (N = 20)** | Densely rough (N = 20) | **Reddish on central dorsum with greenish sides of the body with yellow and greenish belly (N = 20)** | **Pale cream on central dorsum with dark sides of the body with pale cream and dark belly (N = 20)** |
| *B. curupira* | 8.9–10.7 (N = 10) | 8.3–12.3 (N = 4) | 30–35 (N = 10) | 30–34 (N = 4) | **Black and golden (N = 17)** | Smooth (N = 17) | **Mod-erately rough (N = 17)** | **Brown with yellow patches on belly (N = 17)** | **Dark with pale cream patches on belly (N = 17)** |

*(Continued)*

Table 2. (Continued)

| Species | Measurement/ proportion | | | | Iris | Skin texture of the body | | Color | |
|---|---|---|---|---|---|---|---|---|---|
| | SVL male (mm) | SVL female (mm) | HL/SVL male (%) | HL/SVL female (%) | | Dorsum | Lateral | Alive | Preservative |
| *B. ferruginus* | **11.6–12.5** (N = 9) | 13.0–14.5 (N = 4) | 35–40 (N = 9) | 36–39 (N = 4) | Black (N = 14) | **Densely rough** (N = 14) | Densely rough (N = 14) | Orange or **orange with brownish spots or stripe on middle dorsum** (N = 14) | Pale cream or **pale cream with dark spots or stripe on middle dorsum** (N = 14) |
| *B. fusco-lineatus* | 9.7–11.0 (N = 9) | 12.0–12.3 (N = 3) | 30–34 (N = 9) | 32–34 (N = 3) | Black (N = 12) | **Densely rough** (N = 12) | Densely rough (N = 12) | **Orange with greenish stripe on middle dorsum or dorsum green-ish with orange sides of the body and belly** (N = 12) | **Pale cream with dark stripe on mid-dle dorsum or dor-sum dark with pale cream on sides of the body and belly** (N = 12) |
| *B. izeck-sohni* | 10.3–12.1 (N = 11) | 12.5–13.1 (N = 4) | 33–40 (N = 11) | 34–36 (N = 4) | Black (N = 17) | Smooth (N = 17) | **Mod-erately rough** (N = 17) | Orange with small green rounded dots on sides of the body (N = 17) | Pale cream (N = 17) |
| *B. leopardus* | 9.7–12.6 (N = 18) | 10.9–12.2 (N = 9) | 29–34 (N = 18) | 31–34 (N = 9) | Black (N = 40) | Smooth (N = 40) | **Mod-erately rough** (N = 40) | **Orange with green rounded dots on sides of the body and belly** (N = 40) | **Pale cream with small rounded dark dots on sides of the body and belly** (N = 40) |
| *B. mari-aetere-zae* | 10.4–11.2 (N = 4) | 10.7–13.4 (N = 3) | 32–36 (N = 4) | 29–33 (N = 3) | Black (N = 10) | **Mod-erately rough** (N = 10) | Densely rough (N = 10) | **Orange with blue stripe on mid-dle dorsum, brown patches on dorsum and sides of the body, and green rounded dots on belly** (N = 10) | **Pale cream with dark patches on dorsum and sides of the body and dark rounded dots on belly** (N = 10) |
| *B. mirissi-mus* | 9.9–11.7 (N = 10) | 10.0–12.9 (N = 3) | 32–34 (N = 10) | 32–36 (N = 3) | Black (N = 14) | Smooth (N = 14) | Densely rough (N = 14) | **Orange with white stripe on mid-dle dorsum** (N = 14) | **Pale cream with whitish stripe on middle dorsum** (N = 14) |
| *B. olivaceus* | 8.9–10.0 (N = 15) | 11.0–12.6 (N = 4) | 30–35 (N = 15) | 31–34 (N = 4) | Black (N = 19) | **Densely rough** (N = 19) | Densely rough (N = 19) | **Green with orange on chin and center of the belly or with ventral orange in scattered small dots** (N = 19) | **Dark with pale cream on chin and center of the belly or with ventral pale cream in scattered small dots** (N = 19) |
| *B. pernix* | 9.6–11.7 (N = 12) | 12.6–14.0 (N = 10) | 30–36 (N = 12) | 31–33 (N = 10) | Black (N = 22) | Smooth (N = 22) | **Mod-erately rough** (N = 22) | **Orange on central dorsum with dark sides of the body and orange and dark belly** (N = 22) | **Pale cream on central dorsum with dark sides of the body and pale cream and dark belly** (N = 22) |
| *B. pombali* | 9.9–13.3 (N = 15) | 12.7–15.2 (N = 12) | 33–36 (N = 15) | 31–35 (N = 12) | Black (N = 27) | Smooth (N = 27) | **Mod-erately rough** (N = 27) | Orange or orange with brown spots on sides of the body and belly (N = 27) | Pale cream or pale cream with dark spots on sides of the body and belly (N = 27) |

*(Continued)*

| | Measurement/ proportion | | | | | Skin texture of the body | | Color | |
|---|---|---|---|---|---|---|---|---|---|
| Species | SVL male (mm) | SVL female (mm) | HL/SVL male (%) | HL/SVL female (%) | Iris | Dorsum | Lateral | Alive | Preservative |
| *B. quiririensis* | 9.0–11.9 (N = 11) | 9.4–13.2 (N = 9) | 29–33 (N = 11) | 27–33 (N = 9) | Black (N = 20) | Smooth (N = 20) | **Moderately rough** (N = 20) | **Dorsum dark with orange on head and along the vertebrae; ventral orange with variable amount of dark or greenish in posterior belly or in central and posterior belly** (N = 20) | **Dorsum dark with pale cream on head an along the vertebrae; ventral pale cream with variable amount of dark in posterior or in posterior and central belly** (N = 20) |
| *B. tabuleiro* | 9.7–11.6 (N = 3) | **10.7** (N = 1) | 33–37 (N = 3) | 33 (N = 1) | Black (N = 2) | Smooth (N = 4) | Densely rough (N = 4) | **Dorsum olive green with brown spots and a white stripe along the vertebrae; head orange with green, white, and brown spots; sides of the body entirely olive green; belly orange with olive green spots on the throat and cloacal region** (N = 2) | **Dorsum dark brown with light grey head and stripe along the vertebrae; sides of the body pale cream; belly, pale cream with brown spots in the anterior and posterior regions** (N = 4) |
| *B. tridactylus* | 10.7–14.2 (N = 37) | **14.4–15.7** (N = 5) | 31–36 (N = 37) | 31–34 (N = 5) | Black (N = 42) | Smooth (N = 42) | **Moderately rough** (N = 42) | **Orange with small rounded green dots on the sides of the body or dark orange on the dorsum with the sides of the body and belly blackish or dark brown with the middle of the dorsum light brown; in all these patterns, there are small whitish rounded dots or a large whitish patch on the sides of the body** (N = 42) | **Pale cream with dark rounded dots on sides of the body or pale cream dorsum with dark sides of the body and belly or entirely dark** (N = 42) |
| *B. verrucosus* | 9.3–11.7 (N = 20) | 10.4–13.2 (N = 9) | 29–34 (N = 20) | 30–34 (N = 9) | Black (N = 29) | **Densely rough** (N = 29) | Densely rough (N = 29) | **Light green with orange or yellowish head, middle dorsum, and belly** (N = 29) | **Dark with pale cream head, middle of the dorsum, and belly** (N = 29) |

| | Advertisement call | | | | Tips of fingers | | | | | |
|---|---|---|---|---|---|---|---|---|---|---|
| Species | Note groups | Attenuated notes | Maximum number of notes in a note group | Maximum number of pulses per note | I | II | III | Toe V (externally) | Outer metatarsal tubercle | Snout shape (dorsal view) |
| *B. lulai* sp. nov. | Yes (N = 13) | Yes (N = 13) | 2 (N = 13) | 4 (N = 13) | Rounded (N = 32) | Rounded (N = 32) | Pointed (N = 32) | Absent (N = 32) | Present (N = 32) | Rounded (N = 32) |
| *B. actaeus* | Yes (N = 15) | Yes (N = 15) | 2 (N = 15) | **3** (N = 15) | Rounded (N = 12) | Rounded (N = 12) | Pointed (N = 12) | Absent (N = 12) | **Absent** (N = 12) | Rounded (N = 12) |
| *B. albolineatus* | Yes (N = 20) | **No** (N = 20) | 2 (N = 20) | **3** (N = 20) | Rounded (N = 29) | Rounded (N = 29) | Pointed (N = 29) | Absent (N = 29) | Present (N = 29) | Rounded (N = 29) |

*(Continued)*

| Species | Advertisement call | | | | Tips of fingers | | | Toe V (externally) | Outer meta-tarsal tuber-cle | Snout shape (dor-sal view) |
|---|---|---|---|---|---|---|---|---|---|---|
| | Note groups | Atten-uated notes | Maximum number of notes in a note group | Maximum number of pulses per note | I | II | III | | | |
| *B. auro-guttatus* | Yes (N = 6) | Yes (N = 6) | 2 (N = 6) | 4 (N = 6) | Rounded (N = 17) | Rounded (N = 17) | Pointed (N = 17) | Absent (N = 17) | Present (N = 17) | Rounded (N = 17) |
| *B. boticario* | Yes (N = 8) | Yes (N = 8) | 2 (N = 8) | **3** (N = 8) | Rounded (N = 11) | Rounded (N = 11) | Pointed (N = 11) | Absent (N = 11) | Present (N = 11) | Rounded (N = 11) |
| *B. brunneus* | Yes (N = 20) | Yes (N = 20) | **3** (N = 20) | 4 (N = 20) | Rounded (N = 32) | **Pointed** (N = 32) | Pointed (N = 32) | **Vestigial** (N = 32) | Present (N = 32) | **Mucronate** (N = 32) |
| *B. coloratus* | Yes (N = 5) | Yes (N = 5) | 2 (N = 5) | **3** (N = 5) | Rounded (N = 20) | Rounded (N = 20) | Pointed (N = 20) | Absent (N = 20) | Present (N = 20) | **Semicircu-lar** (N = 20) |
| *B. curupira* | Yes (N = 26) | Yes (N = 26) | **5** (N = 26) | **3** (N = 26) | Rounded (N = 17) | Rounded (N = 17) | Pointed (N = 17) | Absent (N = 17) | Present (N = 17) | Rounded (N = 17) |
| *B. ferruginus* | Yes (N = 7) | Yes (N = 7) | 2 (N = 7) | **3** (N = 7) | Rounded (N = 14) | Rounded (N = 14) | Pointed (N = 14) | Absent (N = 14) | Present (N = 14) | Rounded (N = 14) |
| *B. fusco-lineatus* | Yes (N = 7) | **No** (N = 7) | **3** (N = 7) | **3** (N = 7) | Rounded (N = 12) | Rounded (N = 12) | Pointed (N = 12) | Absent (N = 12) | Present (N = 12) | Rounded (N = 12) |
| *B. izeck-sohni* | Yes (N = 1) | Yes (N = 1) | **3** (N = 1) | **2** (N = 1) | Rounded (N = 17) | **Pointed** (N = 17) | Pointed (N = 17) | Absent (N = 17) | Present (N = 17) | Rounded (N = 17) |
| *B. leopardus* | Yes (N = 10) | Yes (N = 10) | **6** (N = 10) | **3** (N = 10) | Rounded (N = 40) | **Pointed** (N = 40) | Pointed (N = 40) | **Vestigial** (N = 40) | Present (N = 40) | **Truncate** (N = 40) |
| *B. mari-aetere-zae* | ? | ? | ? | ? | Rounded (N = 10) | Rounded (N = 10) | Pointed (N = 10) | Absent (N = 10) | Present (N = 10) | Rounded (N = 10) |
| *B. mirissi-mus* | Yes (N = 12) | **No** (N = 12) | 2 (N = 12) | **3** (N = 12) | Rounded (N = 14) | Rounded (N = 14) | **Rounded** (N = 14) | Absent (N = 14) | Present (N = 14) | **Semicircu-lar** (N = 14) |
| *B. olivaceus* | Yes (N = 19) | Yes (N = 19) | 2 (N = 19) | **3** (N = 19) | Rounded (N = 19) | Rounded (N = 19) | Pointed (N = 19) | Absent (N = 19) | Present (N = 19) | Rounded (N = 19) |
| *B. pernix* | Yes (N = 8) | Yes (N = 8) | 2 (N = 8) | 4 (N = 8) | Rounded (N = 22) | Rounded (N = 22) | **Rounded** (N = 22) | Absent (N = 22) | **Absent** (N = 22) | **Semicircu-lar** (N = 22) |
| *B. pombali* | Yes (N = 8) | Yes (N = 8) | **4** (N = 8) | **3** (N = 8) | Rounded (N = 27) | Rounded (N = 27) | Pointed (N = 27) | **Vestigial** (N = 27) | Present (N = 27) | Rounded (N = 27) |
| *B. quiririen-sis* | Yes (N = 9) | Yes (N = 9) | 2 (N = 9) | 4 (N = 9) | Rounded (N = 20) | Rounded (N = 20) | Pointed (N = 20) | **Vestigial** (N = 20) | Present (N = 20) | **Mucronate** (N = 20) |
| *B. tabuleiro* | Yes (N = 5) | Yes (N = 5) | 2 (N = 5) | 4 (N = 5) | Rounded (N = 4) | Rounded (N = 4) | **Rounded** (N = 4) | Absent (N = 4) | **Absent** (N = 4) | Rounded (N = 4) |

*(Continued)*

**Table 2.** (Continued)

| Species | Advertisement call | | | | Tips of fingers | | | Toe V (externally) | Outer meta-tarsal tuber-cle | Snout shape (dor-sal view) |
|---|---|---|---|---|---|---|---|---|---|---|
| | Note groups | Atten-uated notes | Maximum number of notes in a note group | Maximum number of pulses per note | I | II | III | | | |
| *B. tridactylus* | **No** (N = 15) | **No** (N = 15) | --- | **3** (N = 15) | Rounded (N = 42) | Rounded (N = 42 | Pointed (N = 42) | Absent (N = 42) | **Absent** (N = 42) | Rounded (N = 42) |
| *B. verru-cosus* | Yes (N = 11) | Yes (N = 11) | 2 (N = 11) | **8** (N = 11) | Rounded (N = 29) | Rounded (N = 29) | Pointed (N = 29) | Absent (N = 29) | Present (N = 29) | Rounded (N = 29) |

lateral views; nostrils not protuberant, directed anterolaterally; canthus rostralis not distinct; mouth approximately sigmoid in shape; loreal region slightly concave; eye slightly protuberant in dorsal and lateral views; ED 32% of HL; tympanum indistinct; vocal sac not expanded externally; vocal slits present; tongue longer than wide, with posterior half not attached to floor of mouth; choanae small and ovoid, anterior to eyes; vomerine odontophores absent. Arm and forearm relatively slender; arm approximately as long as forearm; finger IV not visible externally; tip of fingers I and II rounded, tip of finger III pointed; relative lengths of fingers (in external view) I < II < III; subarticular tubercles absent; inner and outer metacarpal tubercles absent. Legs short, moderately robust; THL 36% of SVL; shank length 87% of thigh length; toes II–III short and distinct; toes I and V not visible externally; relative lengths of toes (in external view) II < III < IV; subarticular tubercles and inner metatarsal tubercles absent; outer metatarsal tubercle present and discreet. Dorsum with smooth texture, without visible co-ossifications; head, chin and arms with smooth texture; dorsal region of legs and sides of the body with densely rough texture; ventral surface (except chin) with densely rough texture.

## Coloration of the holotype

In life (Fig 2), overall dorsal coloration uniform bright orange from head to pelvic region; granular warts on lateral regions of body greenish yellow, with orange between warts; arms and legs orange, with discrete irregular greenish or brown spots. Chin and chest orange; belly and ventral region of thighs greenish yellow; iris completely black. In preservative after four months (Fig 3), dorsum of head cream; orange regions on the dorsum becomes pale cream, greenish yellow region of the flanks becomes gray, and discreet irregular greenish spots becomes brown.

## Measurements of holotype (in mm)

SVL = 10.6; HL = 3.6; HW = 4.3; ED = 1.2; IOD = 2.1; IND = 1.2; EN = 0.6; SL = 1.1; UEW = 0.7; FLL = 2.2; HAL = 1.6; THL = 4.0; TL = 3.4; TSL = 1.8; FL = 2.5.

## Variation in the type series

Morphometric variation is given in Table 1. Species sexually dimorphic, with the females generally bigger (SVL = 11.7–13.4 mm) than males (SVL = 8.9–11.3 mm). Although the finger IV of most specimens is absent on an external examination, some specimens have a small vestigial trace of the finger IV (e.g., Fig 4A4, E3, F3). There are slight differences in coloration among specimens (Fig 4), particularly in the presence of irregular, green spots in some specimens on the arms, legs, side of the body, and near the head (e.g., Fig 4E1, F1). Though most of the bright orange and yellow coloration fades in preservation, the green spots remain, becoming dark gray (e.g., Fig 4D3, E3, F3).

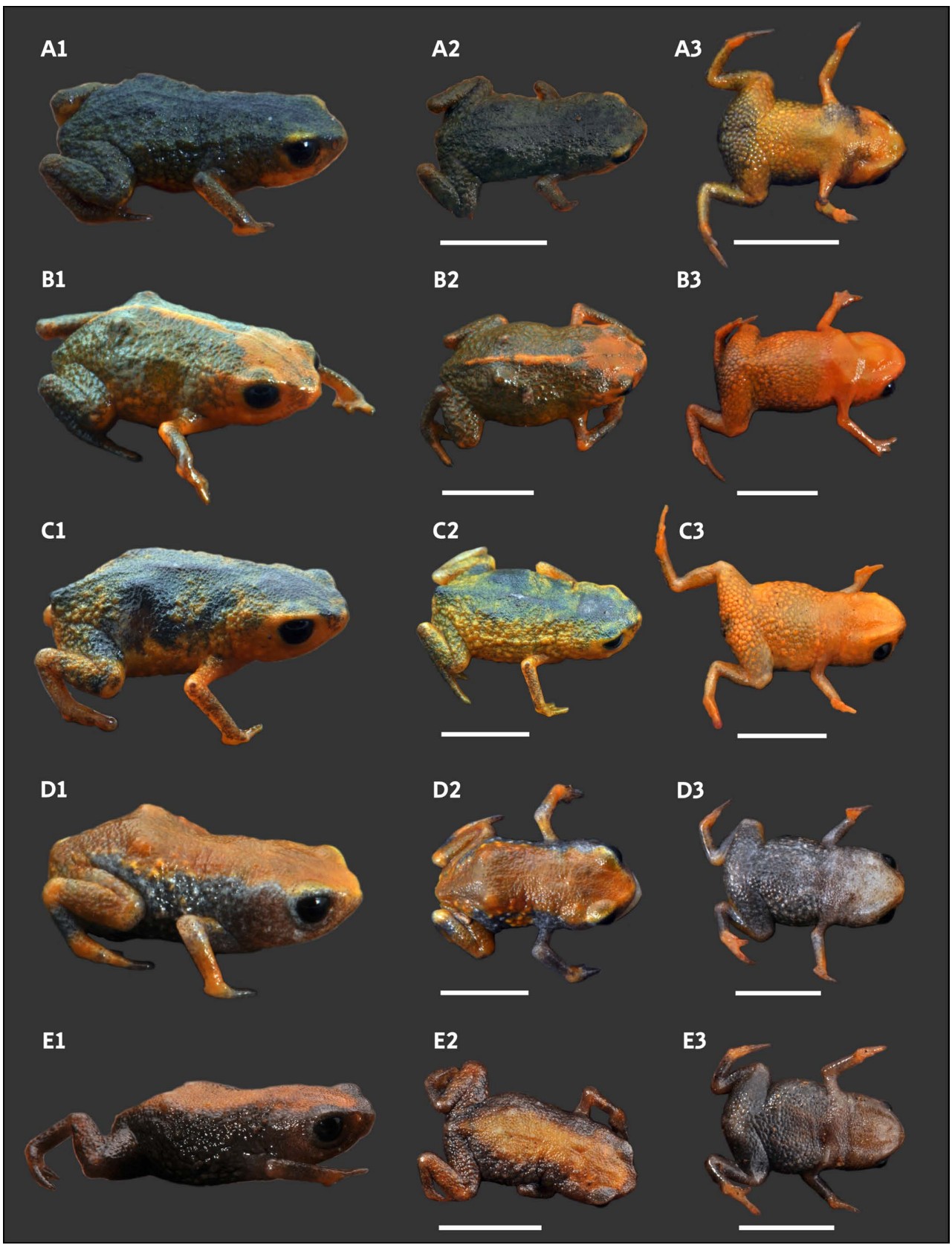

**Fig 5. Divergent color variation of *Brachycephalus* species.** A1–A3 = *B. actaeus* MHNCI 11625 (Forte Marechal Luz, Ilha de São Francisco, municipality of São Francisco do Sul, Santa Catarina). B1–B3 = *B. auroguttatus* MHNCI 11768 (trail to Pedra da Tartaruga, municipality of Garuva, Santa Catarina). C1–C3 = *B. fuscolineatus* MHNCI 11599 (Morro do Baú, municipality of Ilhota, Santa Catarina [type locality]). D1–D3 and E1–E3 = *B. tridactylus* MHNCI 10852 and MHNCI 11767, respectively (Torre Embratel, Parque Estadual do Rio Turvo, municipality of Cajati, São Paulo). Abbreviation: MHNCI = Museu de História Natural Capão da Imbuia, Curitiba, Paraná. Scale bars equal 5 mm. Photographs: A1–D3 = Luiz F. Ribeiro; E1–E3 = Marcos R. Bornschein.

## Description of general osteology

Based on microCT scan of MHNCI 11601 (paratype; male; Fig 6). The skull is short and slightly broader than long. Vomers and nasals are distinct and not synostosed with other bones (Fig 6C). Neopalatines are distinct and minute. Frontoparietals, prootics, exoccipitals, sphenethmoids, and parasphenoid are synostosed, resulting in a skull shape similar to other species in the genus (e.g., *B. coloratus* and *B. curupira*). The frontoparietals are partially fused across the midline. The premaxillae are broad, widely separated, and have weakly developed odontoids; each has a robust pars dentalis, and a robust alary process that is taller than wide and widely separated from adjacent nasal. The quadratojugal has a broad articulation with the maxilla, which is thin and nearly straight. The pterygoids are slender, each with a long anterior ramus that approaches but does not articulate with the adjacent maxilla. The squamosal are robust, and each bears a large rectangular zygomatic ramus which is directed towards the maxillae; they also have a long, slender, and somewhat flattened posterior ramus that has a broad connection to the prootic (Fig 6C). Small, curved sphenethmoids are present at the anterior margin of nasal capsule. The parasphenoid is broad and robust. The vertebral column of *B. lulai* sp. nov. has seven presacral, procoelous, non-imbricate vertebrae, with the eighth presacral fused to the sacral vertebra. First presacral vertebra (atlas) lacks transverse process and has widely separated cotyles. Lengths of the transverse process of presacral vertebrae along with that of the sacral diapophyses: III > IV ≅ SD > V ≅ VI > II ≅ VII (SD = sacral diapophyses). Transverse processes of presacral vertebrae II and III perpendicular to the notochordal axis, IV and V oriented posteriorly and those of VI and VII oriented slightly anteriorly in relation to the notochordal axis. The sacrum (a composite including the fused presacral VIII) has robust transverse processes and a large sesamoid is found near the articulation with each ilium. The urostyle is robust and has a high dorsal ridge that decreases along the length of the bone (Fig 6A). The pectoral girdle is arciferal, robust, and lacks a sternum and omosternum (Fig 6F). The scapula is robust, featuring a prominent anterior process. The suprascapula is weakly ossified with a well-ossified cleithrum. The ilium is robust, with a fully ossified acetabulum with a ventral acetabular expansion composed of ilium, pubis, and ischium. The ilium has a well-developed and prominent dorsal crest (Fig 6E). The radioulna is slightly shorter than the humerus. The femur and tibiofibula are similar in length. The distal carpals (Element Y and II–V) are fused. The radiale and ulnare are large and subequal in size. The phalangeal formula for the manus is 1–2–3–1 and there is both a single ossified prepollex and a small palmar sesamoid. The tips of the terminal manual phalanges are blunt (Fig 6H). There are two large distal tarsals. The phalangeal formula for the pes is 1–2–3–4–1 and there are two plantar sesamoids but no obvious distinct prehallux. The tips of the terminal pedal phalanges are blunt (Fig 6G).

## Description of skin histology

A conspicuous and homogeneously distributed MDL was detected in the skin of the dorsal region of the body of *B. lulai* sp. nov. (Fig 7). In specimens from Pico Garuva, the MDL represented, on average, 22% and 30% of the dermis thickness in the medial and caudal portions of the dorsal region, respectively, while in specimens from Monte Crista, the MDL represented 26% and 27% of the dermis thickness in the medial and caudal of the dorsal region, respectively (Table 3).

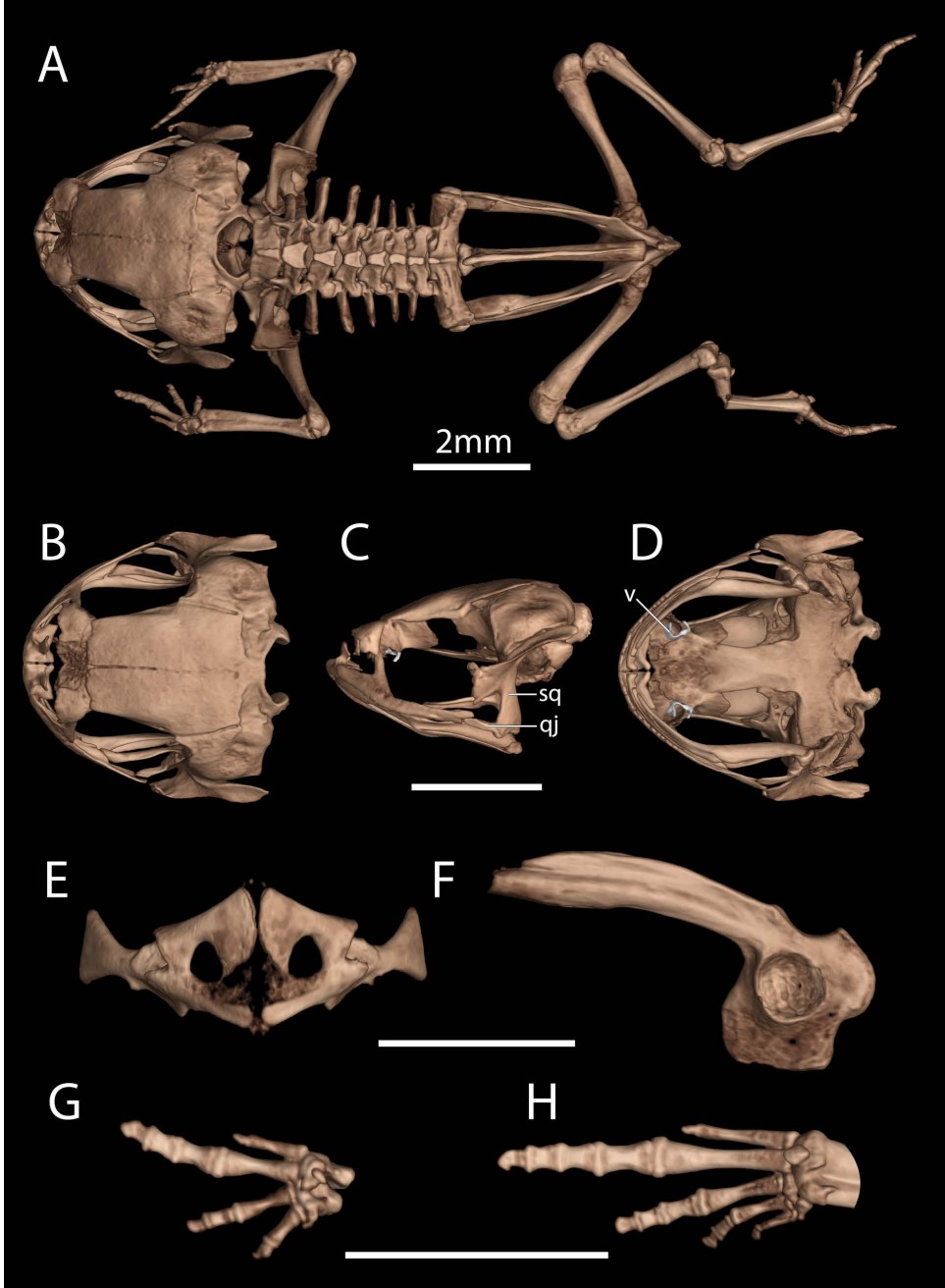

**Fig 6. High-resolution computed tomography (CT) scans of a paratype of *Brachycephalus lulai* sp. nov. (MHNCI 11601) showing key osteo-logical features.** (A) Dorsal view of the skeleton; (B) dorsal, (C) lateral (without the lower jaw), and (D) ventral views of the skull; (E) pectoral girdle in ventral view; (F) ilium in lateral view; (G) left hand in palmar view; (H) left foot in plantar view. While the vomer is fused to surrounding elements, we have highlighted its approximate boundaries. Abbreviations: qj = quadratojugal; sq = squamosal; v = vomer. Scale bars equal 2 mm.

## Bioacoustics

We analyzed 13 advertisement calls (Table 4), each from a distinct individual. Recordings were made at Pico Garuva (MHNCI 224–32, vouchers MHNCI 11597–8, MHNCI 11600) and at Monte Crista (MHNCI 233–6; Table 4; S2 Appendix).

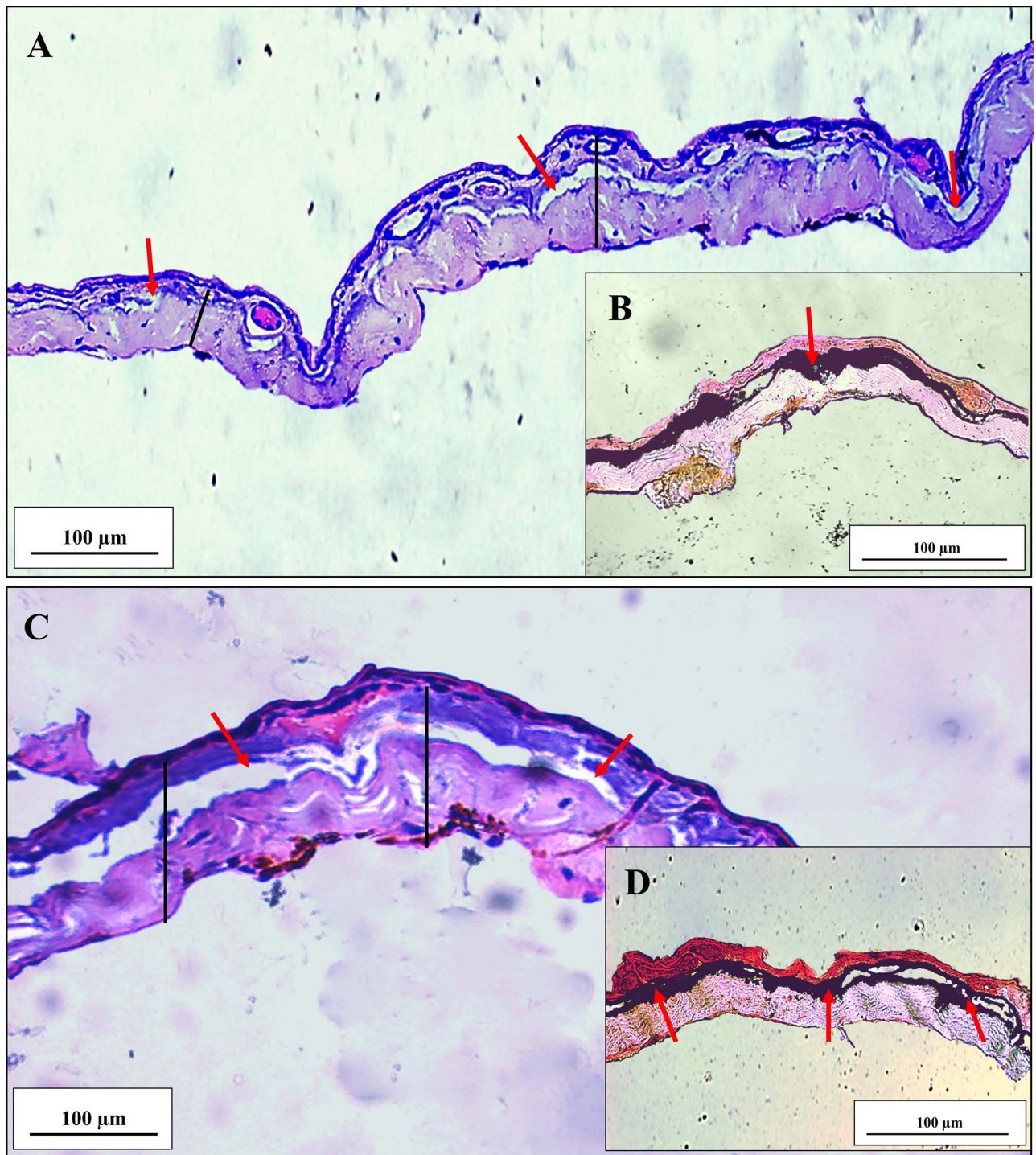

**Fig 7. Photomicrography of the dorsal skin of *Brachycephalus lulai* sp. nov.** (A and B) Specimens from Pico Garuva. (C and D) Specimens from Monte Crista. The red arrows indicate the mineralized dermal layer in the dermis. The black line indicates the extent of the dermis. In A and C, Hematoxylin and Eosin staining; in B and D, von Kossa staining.

**Table 3. Measurement (μm) of the dermis thickness and mineralized dermal layer (MDL) of the skin of the medial and caudal regions of the dorsal body of *Brachycephalus lulai* sp. nov.**

| Body region | Pico Garuva (N=3 individuals) | | Monte Crista (N=3 individuals) | |
|---|---|---|---|---|
| | Dermis | MDL | Dermis | MDL |
| Medial | 58.17±15.89 | 12.83±4.01 | 50.36±7.03 | 13.16±5.12 |
| Caudal | 39.01±16.65 | 11.81±2.49 | 67.12±16.80 | 18.26±5.04 |

Values are provided as mean±standard deviation.

Values represent the range and, if relevant, mean±SD between parentheses (see Table 6 for results of all parameters). The advertisement calls of the species include isolated notes and note groups (Table 4, Fig 8).

### Entire call

We did not record an advertisement call from the beginning, because calls were already in progress when recording started (Table 4). This prevented us from measuring the complete duration of the call, but the call duration from recorded calls varies from 61.7–201.8 s (116.5±45.8 s; N=13 calls; Fig 8A). Note rate was 5.2–10.6 notes per minute (7.5±1.7 note per minute; N=13 calls). The number of notes per call (excluding attenuated notes) varies from 7–30 notes (15.8±7.0 notes; N=13 calls). Note duration (of isolated notes and notes of note groups) was 0.010–0.061 s (0.033±0.009 s; N=205 notes). Note dominant frequency was 5.2–6.6 kHz (6.2±0.2 kHz; N=205 notes).

### Isolated notes

Eleven calls present only isolated notes and the total number of notes are not possible to determine since these calls were not recorded from the beginning. However, the duration of the call including only isolated notes based on the recorded part of these call lasts from 48.9–201.8 s (106.8±50.6 s; N=11 calls) and they included 7–30 isolated notes (14.9±7.2 notes; N=11 calls). Note rate of the call including only isolated notes was 5.2–16.1 notes per minute (8.0±2.8 notes per minute; N=13 calls). Isolated notes present 1–4 pulses (2.5±0.6 pulses; N=189 isolated notes; Fig 8). The inter-note interval in isolated notes was 4.6–15.0 s (8.2±2.2 s; N=173 inter-note intervals).

### Note groups

Two calls present note group, one with three note groups and the other with five note groups. The duration of the call including only note groups was 18.4–33.7 s (26.0±10.8 s; N=2 calls). Note rate of the call including only note groups was 16.3–17.8 notes per minute (17.1±1.0 notes per minute; N=2 calls). All note groups present two notes composing the group (N=8 note groups; Fig 8). Duration of note groups was 0.361–0.416 s (0.396±0.020 s; N=7 note groups). The inter-note interval within note groups was 0.290–0.346 s (0.325±0.021 s; N=8 inter-note interval within note groups). Note groups present 2–3 pulses per note (2.3±0.5 pulses; N=16 notes; Fig 8).

### Attenuated notes

All advertisement calls recorded presented attenuated notes (N=13 calls; Fig 8E and 9), in the number of 7–21 notes per call associated with attenuated notes (14.1±5.1 notes). There were 181 notes preceded by a single attenuated note and three notes preceded by two attenuated notes. All note groups were preceded by attenuated notes (N=8 note groups). However, only the first note comprising the note group was preceded by the attenuated note (Table 4). Most of the isolated notes were preceded by attenuated notes (N=176 notes), with only 13 that were not (Table 4). Attenuated notes present 1–2 pulses (1.0±0.2 pulse; N=186 attenuated notes). Note dominant frequency for attenuated notes was 5.1–7.7 kHz (5.7±0.4 kHz; N=186 attenuated notes).

**Table 4. Structure of the advertisements calls (AC) of *Brachycephalus lulai* sp. nov. and other members of the *B. pernix* species group.**

| Individuals (ind) and call deposit number | Structure | A | B |
|---|---|---|---|
| *B. actaeus* (Forte Marechal Luz, Ilha de São Francisco, municipality of São Francisco do Sul, Santa Catarina) | | | |
| Ind 1 (MHNCI 365) | $_{1-}$1, $_{1-}$1, $_{1-}$1, $_{1-}$1, $_{1-}$1, $_{1-}$2, $_{1-}$2, $_{1-}$2, $_{1-}$2, $_{1-}$2, $_{1-}$2, $_{1-}$2, $_{1-}$2, $_{1-}$2, $_{1-}$2, $_{1-}$2, $_{1-}$2, $_{1-}$2, $_{1-}$2, $_{1-}$2, $_{1-1-}$2, $_{1-1-}$2, $_{1-1-}$2, $_{1-}$2, 2, 2, $_{1-1-}$2, $_{1-1-}$2, $_{1-1-}$2, $_{1-1-}$2, $_{1-}$2, $_{1-}$2, $_{1-}$2, $_{1-}$2, 1 | 2 | |
| Ind 2 (MHNCI 366) | 1, 2, 2, 2, 2, 2, 2, 2, 2, 2, 3, 3, 3, 3, 3, 3, 2, 2, 3, 3, 2, 2, 2, 2, 2, 2, 2, 2, 1, 1, 1, 1 | 2 | |
| Ind 3 (MHNCI 367) | 1, 1, 1, 1, 1, 1, 2, 2, 2, 2, 2, 2, 2, 2, 2, 2, $_{1-1-}$2, 2, 2, 2, 2, 2, 2, 2, 2, 2, 2, 2, 2 | 2 | |
| Ind 4 (MHNCI 368) | 1, 1, 1, 1, 1, 1, 1, 2, 2, 2, 1, 2, 2, 2, 2, 2, 2, 2, 2, 2, 2, 2, 2, 2, 2, 2, 2, 2, 2, 2 | 1 | |
| Ind 5 (MHNCI 369) | $_{1-}$1, $_{1-}$1, 1, 1, 1, 1, 2, 2, 2, 2, 2, 2, 2, 2, 2, 2, 2, $_{1-}$2, 2, 2, 2, 2, $_{1-}$2, 2, 2, 2, $_{1-}$2, 2, 2, 2, 2, 2, 2, 2, $_{1-}$2, $_{1-}$2, 2, 2, 2, 2, 3, 2 | 1 | |
| *B. actaeus* (Serra da Palha, Laranjeiras, Ilha de São Francisco, municipality of São Francisco do Sul, Santa Catarina) | | | |
| Ind 1 (MHNCI 289) | 2, 2, 2, 2, 2, 2, 2, 2, 2, 2, 2 | 4 | |
| *B. actaeus* (Serra da Tiririca, municipality of Itapoá, Santa Catarina) | | | |
| Ind 1 (MHNCI 280) | 2, 2, 2, 2, 2, 2, 2, 2, 2, 2, 2, 2, 2, $_{1-}$2, 2 | ? | |
| Ind 2 (MHNCI 281) | 3, 3, 3, 3 | ? | x |
| Ind 3 (MHNCI 282) | 2, 2, 3, 3, 3, 3, 3, 3 | 6 | x |
| Ind 4 (MHNCI 283) | 1, 1, 1,?,?,?, 1, 1, 2, 2, 2, 2, 2, 2, 2, 2, 2, 2, 2 | 3 | |
| Ind 5 (MHNCI 284) | 1, 1, 2, 2, 2, 2, 2, 2, 2, 2, 2, 2, 2, 2, 2, 2, 2, 2, 2 | 1 | |
| Ind 6 (MHNCI 285) | 2, 2, $_{1-}$2, 2, 2, 2, $_{1-}$2, 2, $_{1-}$2, 2, $_{1-}$2, 2, 2, (2–2), (2–2), 2 | 3 | |
| Ind 7 (MHNCI 286) | 2, 2, 2, 2, 3, 3, 3, 3, 3, 3, 3, 3, 3, 3, 3, 3, 3, 3, 3, 3, 3, 3, 3, 3, 3 | ? | |
| Ind 8 (MHNCI 287) | 2, 2, 2, 2, 2, 2,2, 2, 2, 2, 2, 1 | 2 | |
| Ind 9 (MHNCI 288) | 2, 2, 2, $_{1-}$2, 2, 2, $_{1-}$2, $_{1-}$2, 2, 2, 2, 2, 2, 2, $_{1-}$2, 2, 2, 2, 2, 2, 2, $_{1-}$2, 2, 2, $_{1-}$2, 2, $_{1-}$2, $_{1-}$2, 2, $_{1-}$2, $_{1-}$2, $_{1-}$2, $_{1-}$2 | 2 | |
| Ind? (MHNCI 302) | 2, 3, 2, $_{1-}$3, $_{1-}$3 | 2 | |
| Ind? (MHNCI 303) | 2, 2, 2, 2, 2, 3, 3, 3, 3, 3, 3, 3, 3, 3, 3, 3, 3, 3, 3, 3, 3, 3, 3, 2 | ? | |
| Ind? (MHNCI 304) | 2, 2, 2, 2, 3, 3, 3, 3, 3, 3, 3, 3, 3, 3, 3, 3, 3, 3, 3, 3, 2 | ? | |
| Ind? (MHNCI 305) | 1, 2, 2, 2, 2, 3, 2, $_{1-}$3, $_{1-}$3, $_{1-}$3, 3, $_{1-}$3, $_{1-}$3, $_{1-}$3, $_{1-1-}$3, $_{1-}$3, $_{1-}$3, $_{1-}$3, $_{1-}$3, $_{1-}$3, $_{1-}$3, 3, 3, $_{1-}$2, 2 | 1 | |
| Ind? (MHNCI 306) | 2, 2, 2, 2, 3, $_{1-}$3, 3, 3, 3, $_{1-}$3, $_{1-}$3, 3, $_{1-}$3, $_{1-}$3, $_{1-}$3, 3, $_{1-}$3, $_{1-}$3, $_{1-}$3, $_{1-}$3, $_{1-}$3, 3, 2, $_{1-}$2, 1 | 1 | |
| Ind? (MHNCI 307) | 2, 1, $_{1-}$2, 2, $_{1-}$2, 2, 2, 2, 3, 2, $_{1-}$3, $_{1-}$3, $_{1-}$3, $_{1-}$3, $_{1-}$3, $_{1-}$3, $_{1-}$3, $_{1-}$3, $_{1-}$3, $_{1-}$3, $_{1-}$3, $_{1-}$3, $_{1-}$3, $_{1-1-}$3, $_{1-}$3, $_{1-1-}$3, $_{1-}$3, $_{1-}$3, $_{1-}$2, $_{1-}$2 | 2 | |
| *B. auroguttatus* (Trail to Pedra da Tartaruga, municipality of Garuva, Santa Catarina) | | | |
| Ind 1 (MHNCI 381) | $_{1-}$2, $_{1-}$2, $_{1-}$2, $_{1-}$2, $_{1-}$2, ($_{1-}$2–2), ($_{1-}$3–2), ($_{1-}$2–2), ($_{1-}$2–3) | 8 | x |
| Ind 1 (MHNCI 387) | $_{1-}$2, $_{1-}$2, $_{1-}$3, $_{1-}$3, $_{1-}$2, $_{1-}$3, $_{1-}$3, ($_{1-}$2–2), ($_{1-}$3–2), ($_{1-}$2–3) | 4 | |
| Ind 2 (MHNCI 382) | $_{1-}$2, $_{1-}$2, $_{1-}$2, $_{1-}$3, $_{1-}$3, $_{1-}$3, $_{1-}$3, $_{1-}$3, $_{1-}$3, $_{1-}$3, $_{1-}$4, $_{1-}$3, $_{1-}$3, $_{1-}$4, $_{1-}$3, $_{1-}$3, 2, $_{1-}$3, $_{1-}$3, $_{1-}$3, $_{1-}$3, $_{1-}$3, $_{1-}$3, $_{1-}$3, $_{1-1-}$3, $_{1-}$3, $_{1-}$3, $_{1-1-}$4 | 1 | |
| Ind 2 (MHNCI 384) | $_{1-}$1, $_{1-}$1, $_{1-}$2, $_{1-1-}$2, $_{1-1-}$2, $_{1-1-}$3, $_{1-1-}$3, $_{1-1-}$3, $_{1-1-}$3, $_{1-1-}$4, $_{1-1-}$4 | 1 | |
| Ind 3 (MHNCI 383) | 1, $_{1-}$1, $_{1-}$1, ($_{1-}$1–1), ($_{1-}$1–1) | 3 | |
| Ind 4 (MHNCI 385) | $_{1-}$3, 2, 2, 2, 2, 2, 3, 3, 3, 3, 3, $_{1-}$3, $_{1-}$3 | > 8 | x |
| Ind 5 (MHNCI 386) | $_{1-}$2, $_{1-}$2, $_{1-}$2, $_{1-}$2, $_{1-}$2, $_{1-}$2, $_{1-}$2, $_{1-1-}$2, $_{1-1-}$3, $_{1-}$2, $_{1-}$2 | ? | x |
| Ind 6 (MHNCI 388) | 2, 2, $_{1-}$2, $_{1-}$2, $_{1-}$2, $_{1-}$3, $_{1-}$3 | 4 | x |
| *B. boticario* (Morro do Cachorro, boundary of the municipalities of Blumenau, Gaspar, and Luiz Alves, Santa Catarina) | | | |
| Ind 1 (MHNCI 138) | 3, 3, 3, 3, 3, (3–3), (3–3), (3–3), (3–3), (3–3), (3–3), (3–3), (3–3), (3–3), (3–3) | ? | |
| Ind 1 (MHNCI 141) | 1, 1, 2, 2, 2, 2, 3, 3, 3, 3, 3, 3, 3, 3, 3, 3, (3–3), 3, (3–3), (3–3), (3–3), (3–3), (3–3), (3–3) | 1 | |
| Ind 2 (MHNCI 139) | 2, 2, 2, 2, 2, 2, (2–2), (2–2), 2, 1, 2, 1, 1 | ? | |
| Ind 2 (MHNCI 141) | 1, 1, 1, 2, 2, 2, 2, 2, 2, 2, 2, 2, 2, 2, 2, 2, (2–2), (2–2), (2–2), (2–2), (2–2), (2–2), (2–2) | 0 | |
| Ind 3 (MHNCI 140) | 1, 2, 2, 2, 2, 2, 2, 2, 2, 3, 3 | 3 | |
| Ind 3 (MHNCI 142) | 1, 1, 1, 2, 2, 2, 2, 2, 2, 2, 2, 2, 2, 2, (2–2), (2–2), (2–2) | 2 | |
| Ind 3 (MHNCI 143) | 1, 1, 2, 2, 2 | 2 | x |

*(Continued)*

| Individuals (ind) and call deposit number | Structure | A | B |
|---|---|---|---|
| Ind 4 (MHNCI 144) | 1, 1, 2, 2, 2, 2, 2, (2–2), (2–2), (2–2), (2–2), (2–2) | 3 | |
| Ind 4 (MHNCI 145) | 1, 2, 2, 1, 2, 1, 2, (1–1), (1–1) | ? | |
| Ind 5 (MHNCI 146) | 1, 1, 2, 2, 2, 2, 2, 2, 2, 2, 2, 2, (2–2), (2–2), (2–2), (2–2), 2, 2 | 1 | |
| Ind 5 (MHNCI 148) | 1, 1, 1, 1, 2, 2, 2, 2, 2, 2, (2–2), (2–2), (2–2), (2–2), (2–2), (2–2) | 1 | |
| Ind 6 (MHNCI 160) | $_1$1, 2, $_1$2, $_1$2, $_1$2, $_1$2, $_1$2, $_1$2, $_1$2, $_1$2, $_1$2, $_1$2, $_1$2, $_1$2, ($_1$2–2), $_1$2, ($_1$2–2), ($_1$2–2) | ? | |
| Ind 7 (MHNCI 161) | 2, 2, 2, 2, 2, 2, 2, 2, (2–2), (2–2), (2–2), (2–2), (2–2), (2–2), (2–2) | ? | |
| Ind 8 (MHNCI 162) | 2, 1, 2, 2, 2, 2, 2, 2, 2, 2, 2, 2, 2, 2, 2, (2–2), 2 | ? | |
| Ind? (MHNCI 163) | 1, 1, 2, $_1$2, $_1$2, $_1$2, $_1$2, $_1$2, $_1$2, $_1$2, $_1$2, 2, 2, (2–2), 2, (2–2), 2, (2–2), 2, 2, 2 | 0 | |
| Ind? (MHNCI 164) | 1, 1, 1, 1, 1, 1, 2, 2, 2, 2, 2, 2, 2, 2, 2, 2, 2, 2, 2, 2, (2–2), 2, 2, 1 | 0 | |
| *B. brunneus* (Caratuva, Serra dos Órgãos, municipality of Campina Grande do Sul, Paraná) | | | |
| Ind 1 (MHNCI 083) | $_1$2, $_1$2, $_1$2, $_1$2, $_1$2, $_1$2, $_1$2, $_1$2, $_1$3, $_1$3, $_1$3, $_1$3, $_1$3, $_1$3, $_1$3, $_1$3, $_1$3, $_1$3, $_1$3, $_1$3, ($_1$3–2), ($_1$3–2), ($_1$3–2), ($_1$3–3), ($_1$3–3), ($_1$3–2) | ? | |
| Ind 2 (MHNCI 084) | $_1$2, $_1$3, $_1$2, $_1$2, $_1$3, ($_1$3–2), ($_1$3–2), $_1$3, $_1$3, ($_1$3–2), ($_1$3–3), ($_1$3–2), ($_1$3–3), ($_1$3–3) | ? | |
| Ind 3 (MHNCI 085) | $_1$2, $_1$3, $_1$2, $_1$3, $_1$3, $_1$3, $_1$3, $_1$3, $_1$3, ($_1$3–2), ($_1$3–3), ($_1$3–3), ($_1$3–3), ($_1$3–3), ($_1$3–3), ($_1$3–3), ($_1$3–3), ($_1$3–2), ($_1$3–3–3) | ? | |
| Ind 4 (MHNCI 086) | $_1$2, $_1$2, $_1$2, $_1$2, $_1$3, $_1$3, $_1$3, $_1$2, $_1$3, $_1$2, ($_1$2–2), ($_1$3–2), ($_1$2–2), ($_1$3–2), ($_1$3–2), ($_1$2–2–2), ($_1$3–2–2), ($_1$3–2–2), ($_1$3–2–2) | ? | |
| Ind 5 (MHNCI 087) | ($_1$2–2), ($_1$2–2), ($_1$3–2), ($_1$2–2), ($_1$3–2), ($_1$3–2), $_1$2, $_1$2, $_1$2, $_1$2, $_1$1, $_1$1 | ? | |
| Ind 6 (MHNCI 088) | $_1$2, $_1$3, $_1$2, ($_1$2–2), ($_1$2–2), ($_1$3–2), ($_1$3–3), ($_1$3–3–2), ($_1$3–2) | ? | |
| Ind 7 (MHNCI 089) | ($_1$3–2), ($_1$3–2), ($_1$3–3), ($_1$3–3–2), ($_1$3–2–2), ($_1$2–2–2), ($_1$2–2–2), ($_1$2–2–2), ($_1$2–2–2), ($_1$2–2) | ? | |
| Ind 8 (MHNCI 090) | (3–3), ($_1$3–3), ($_1$3–3), ($_1$3–3) | ? | |
| Ind 9 (MHNCI 091) | $_1$2, $_1$2, $_1$3, $_1$3, $_1$2, $_1$3, $_1$3, $_1$2, ($_1$2–2), ($_1$2–2), ($_1$3–2), ($_1$3–2), ($_1$2–2), ($_1$2–2), ($_1$2–2) | ? | |
| Ind 10 (MHNCI 092) | $_1$4, $_1$3, $_1$4, $_1$4, ($_1$4–4), ($_1$4–4), ($_1$4–3), ($_1$4–3), ($_1$4–3–3), ($_1$4–4–3), ($_1$4–4–3), ($_1$3–3–3), ($_1$3–4–3) | ? | |
| Ind 11 (MHNCI 093) | $_1$1, 1, $_1$1, $_1$1, $_1$1, $_1$1, $_1$2, $_1$2, $_1$2, $_1$2, $_1$2, $_1$2, $_1$3, $_1$2, $_1$3, $_1$3, $_1$3, $_1$3, $_1$2, ($_1$3–2), ($_1$2–2), ($_1$2–2), ($_1$2–2), ($_1$2–2), ($_1$2–2), ($_1$2–2) | ? | |
| Ind 12 (MHNCI 094) | $_1$3, $_1$3, $_1$3, $_1$3, $_1$3, ($_1$3–3), ($_1$3–3), ($_1$3–3), ($_1$3–3), ($_1$3–3), ($_1$3–3), ($_1$3–3–3), ($_1$3–3–2), ($_1$3–3–2) | ? | |
| Ind 13 (MHNCI 095) | $_1$1, $_1$1, $_1$2, $_1$2, $_1$2, $_1$2, $_1$2, $_1$3, $_1$2, $_1$2, $_1$3, $_1$3, $_1$3, $_1$3, $_1$3 | ? | |
| Ind 14 (MHNCI 096) | $_1$1, $_1$1, $_1$1, $_1$1, $_1$2, $_1$2, $_1$2, $_1$2, $_1$2, $_1$3, $_1$3, $_1$3, $_1$3, $_1$3, ($_1$3–2), ($_1$3–3), ($_1$3–3), ($_1$3–3), ($_1$2–2–2), ($_1$2–2–2), ($_1$3–2–2), ($_1$2–2–2), ($_1$2–2–2) | ? | |
| *B. brunneus* (Caranguejeira, Serra da Graciosa, municipality of Quatro Barras, Paraná) | | | |
| Ind 1 (MHNCI 349) | $_1$2, $_1$3, $_1$2, $_1$2, $_1$2, $_1$4, $_1$2, $_1$3, $_1$2, $_1$3, $_1$2, $_1$2, $_1$2, $_1$2, $_1$2 | ? | ? |
| Ind 2 (MHNCI 350) | ($_1$3–3), $_1$3, $_1$3 | ? | ? |
| Ind 3 (MHNCI 351) | $_1$3, $_1$3, $_1$3, $_1$2, (2–2), $_1$2, $_1$2, $_1$2, $_1$2, $_1$2, $_1$2, ($_1$2–2), ($_1$2–2), ($_1$3–2), ($_1$3–3), ($_1$2–2), $_{1–1}$2, 3 | ? | ? |
| Ind 4 (MHNCI 352) | ($_1$3–3) ($_1$3–3), ($_1$3–2), ($_1$3–1) | ? | ? |
| Ind 5 (MHNCI 353) | ($_1$3–2), ($_1$3–3), ($_1$3–3), ($_1$3–3), ($_1$3–3), ($_1$2–3–2) | ? | ? |
| Ind 6 (MHNCI 353) | $_1$3, $_1$3, $_1$3, 3, $_1$4, 4 | ? | ? |
| *B. coloratus* (Estância Hidroclimática Recreio da Serra, Serra da Baitaca, municipality of Piraquara, Paraná) | | | |
| Ind 1 (MHNCI 245) | $_1$2, $_1$2, $_1$3, $_1$3, $_1$3, $_1$3, $_1$3 | 2 | |
| Ind 2 (MHNCI 246) | 1, 2, 2, 2, 2, 2, 2, 2, 2, 3, 3, 3, 3, 3 | 2 | |
| Ind 2 (MHNCI 250) | 1, $_1$1, $_1$2, $_1$3, $_1$3, $_1$3, $_1$3, $_1$3, $_1$3, $_1$3, $_1$3 | 1 | |
| Ind 3 (MHNCI 247) | 1, 1, 1, 1, $_1$1, $_1$2, $_1$2, $_1$2, $_1$2, $_1$2, $_1$2, $_1$2, $_1$2, $_1$2, $_1$2, $_1$2, $_1$2, $_1$2, $_1$2, $_1$2, $_1$3, $_1$2, $_1$3, $_1$3, $_1$3 | 1 | |
| Ind 3 (MHNCI 248) | 1, $_1$2, $_1$2, $_1$2, $_1$2, $_1$2, $_1$2, $_1$2, $_1$2, $_1$3, $_1$2, $_1$3, $_1$2, 2, $_1$2, $_1$3, $_{1–1}$3, $_{1–1}$3, $_{1–1}$3, $_{1–1}$3, $_{1–1}$3, ($_{1–1}$2–2), $_{1–1}$2, $_{1–1}$2, $_{1–1}$2, $_{1–1}$3, $_{1–1}$2, ($_{1–1}$2–2) | 2 | |
| Ind 3 (MHNCI 249) | $_1$1, $_1$1, $_1$1, $_1$1, $_{1–1}$2, $_{1–1}$2, $_{1–1}$2, $_{1–1}$2, $_{1–1}$2, $_{1–1}$2, $_{1–1}$2, $_{1–1}$2, $_{1–1}$2, $_{1–1}$2, $_{1–1}$2, $_{1–1}$2, $_{1–1}$2, $_{1–1}$3, $_{1–1}$2, $_{1–1–1}$2, $_{1–1}$2, $_{1–1–1}$2, $_{1–1}$2, $_{1–1}$2, ($_{1–1}$2–2) | 1 | |
| Ind 3 (MHNCI 354) | $_1$1, $_1$1, $_1$1, $_1$1, $_1$1, $_1$2, $_1$2, $_1$2, $_1$2, $_1$2, 2, $_1$2, $_1$2, $_1$2, $_1$2, $_1$2, $_1$3, $_1$2, $_1$3, $_1$3, $_1$2, $_1$3, 2, $_1$2, $_1$2, $_1$2 | 1 | |

*(Continued)*

| Individuals (ind) and call deposit number | Structure | A | B |
|---|---|---|---|
| Ind 4 (MHNCI 355) | $_12$, $_13$, $_13$, $_12$, $_12$, $_12$, $_13$, $_13$, $_13$, $_13$, $_13$ | 1 | |
| Ind 5 (MHNCI 356) | $_12$, $_12$, $_12$, $_13$, $_13$, $_13$, $_13$, $_13$, $_13$, $_13$, $_13$, $_23$ | 1 | |
| *B. curupira* (Morro do Canal, municipality of Piraquara, Paraná) | | | |
| Ind 1 (MHNCI 097) | $_11$, $_11$, $_11$, $_11$, $_11$, $_11$, $_11$, $_11$, $_11$, $_11$, $_11$, $_11$, $_11$, $_11$, ($_12$–1), ($_12$–1), ($_12$–2), ($_12$–2), ($_12$–2), ($_12$–2–2), ($_12$–2–2), ($_11$–2–2), ($_11$–2–2), ($_11$–2–1), ($_11$–2–2), ($_11$–2–1), ($_11$–1–1), ($_11$–1–1), ($_11$–1–1) | ? | |
| Ind 2 (MHNCI 098) | 1, $_11$, 1, 1, $_11$, $_11$, $_11$, $_11$, $_11$, $_11$, $_11$, $_11$, $_11$, $_11$, ($_11$–1), ($_12$–1), ($_11$–1), ($_12$–2), ($_11$–1), ($_12$–2), ($_11$–1–1), ($_11$–2–1), ($_12$–2–1), ($_12$–2–1), ($_12$–2–1), ($_12$–1–1), ($_12$–1–1), ($_11$–1–1) | ? | |
| Ind 3 (MHNCI 099) | $_12$, $_12$, $_12$, $_12$, $_12$, $_12$, $_12$, $_12$, $_12$, $_12$, ($_12$–2), ($_12$–2), ($_12$–2), ($_12$–2), ($_12$–2), ($_12$–2), ($_12$–2–2), ($_12$–2–2), ($_12$–2–2), ($_12$–2–2), ($_12$–2–2), ($_12$–2) | ? | |
| Ind 4 (MHNCI 100) | $_11$, $_11$, $_11$, $_11$, $_11$, $_11$, $_11$, $_12$, $_11$, $_11$, $_11$, $_11$, $_11$, $_11$, ($_12$–2), ($_12$–2), ($_12$–2), ($_12$–2), ($_11$–1), ($_12$–2–2), ($_12$–2–2), ($_12$–1–2), ($_12$–2–2), ($_12$–2–2), ($_12$–2–2), ($_12$–2–2–2), ($_12$–2–2–2), ($_12$–2–2–2), ($_11$–2–1–1), ($_11$–1–1) | ? | |
| Ind 5 (MHNCI 101) | $_11$, $_11$, $_11$, $_11$, $_11$, 1, $_11$, $_11$, $_11$, $_11$, $_11$, ($_11$–1), ($_11$–1), ($_11$–2), ($_12$–2), ($_12$–1), ($_11$–2), ($_11$–2–1), ($_11$–2–1), ($_11$–2–1), ($_11$–1–1), ($_11$–1–1), ($_11$–2–1), ($_11$–1–1) | ? | |
| Ind 6 (MHNCI 102) | 1, $_11$, $_11$, $_11$, $_11$, ($_11$–1), $_11$, 1, ($_11$–1), ($_11$–1–1), ($_13$–1–1), ($_11$–1–1), ($_11$–1–1), (1–1–1–1), ($_11$–1–1–1), ($_11$–1–1), ($_11$–1–1–1) | ? | |
| Ind 7 (MHNCI 103) | 1, $_11$, $_11$, $_11$, $_11$, 1, $_11$, $_11$, $_11$, $_11$, $_11$, ($_{1-1}$1–1), ($_11$–1), ($_11$–1), ($_11$–1), ($_11$–2), ($_11$–2), ($_11$–2–1), ($_12$–1), ($_11$–2–1), ($_11$–2–1), ($_11$–2–1), ($_11$–1–1), ($_11$–1–1), ($_11$–1–1) | ? | |
| Ind 8 (MHNCI 104) | 1, 1, 1, 1, 1, 1, 2, 2, 2, ($_12$–2), ($_12$–2), ($_12$–2), ($_12$–2–2), ($_12$–2–2), ($_12$–2–2), (2–2–2–2), ($_12$–2–2) | ? | |
| Ind 9 (MHNCI 105) | $_12$, $_12$, ($_11$–1), ($_11$–1), ($_11$–1), ($_11$–1), ($_11$–1), ($_12$–1–1), ($_12$–1–1), ($_11$–1–1), ($_11$–1–1), ($_11$–1–1), ($_11$–1–1), ($_11$–1–1), ($_11$–1–1) | ? | |
| Ind 10 (MHNCI 106) | $_11$, $_11$, ($_11$–1), ($_11$–1), ($_11$–1), ($_11$–1), ($_11$–1–1), ($_11$–1–1), ($_11$–1–1), (1–1–1), (1–1–1), (1–1–1–1), (1–1–1–1), (1–1–1) | ? | |
| *B. curupira* (Serra do Salto, Malhada District, municipality of São José dos Pinhais, Paraná) | | | |
| Ind 1 (MHNCI 107) | $_11$, $_11$, $_11$, $_11$, $_11$, $_12$, $_11$, $_11$, $_12$, $_12$, $_{1-1}2$, $_{1-1}2$, ($_{1-1}2$–2), ($_12$–1), ($_12$–2), ($_12$–2), ($_12$–2), ($_12$–2), ($_11$–2), ($_11$–2), ($_11$–2) | ? | |
| Ind 2 (MHNCI 108) | $_11$, $_11$, $_11$, $_11$, $_11$, $_11$, $_11$, $_11$, $_11$, $_11$, $_11$, $_11$, $_12$, $_12$, $_12$, $_12$, $_12$, $_12$, $_12$, ($_12$–1), $_11$, $_12$, ($_12$–2), ($_12$–2), ($_12$–2), ($_12$–2), ($_12$–1) | ? | |
| Ind 3 (MHNCI 109) | 1, 1, $_11$, $_11$, 1, $_11$, $_21$, $_11$, $_12$, $_12$, $_12$, $_12$, $_12$, ($_12$–1), ($_12$–1), ($_12$–1), ($_11$–1), ($_11$–1), ($_11$–1), ($_11$–1) | ? | |
| Ind 4 (MHNCI 110) | $_12$, 2, 1, 2, 1, 1, 2, 2, 1, 2, $_12$, $_12$, $_12$, $_12$, $_12$, $_12$, $_12$, $_12$, $_12$, ($_12$–2), ($_12$–2), ($_12$–2), ($_12$–2), ($_12$–2), ($_12$–2), (2–2), (2–2), ($_11$–2), ($_11$–2), ($_12$–1), 1 | ? | |
| Ind 5 (MHNCI 111) | $_12$, ($_11$–1), ($_12$–1), ($_11$–2), ($_11$–1), ($_11$–1), ($_11$–1), ($_11$–1), ($_11$–1), $_11$, $_11$ | ? | |
| Ind 6 (MHNCI 112) | ($_12$–2), ($_12$–2), ($_12$–2), ($_12$–2–2), ($_12$–2–2), ($_12$–2–2), ($_12$–2–2–2), ($_11$–2–2–2), ($_12$–2–2–2), ($_12$–2–2–2), ($_11$–2–2–2), ($_11$–2–2–2), (1–2–2–1–2), (1–2–2–1) | ? | |
| Ind 7 (MHNCI 113) | $_12$, $_12$, $_12$, $_12$, $_12$, ($_12$–2), ($_12$–2), ($_12$–2), ($_12$–2), ($_12$–2), ($_12$–2), ($_12$–2), ($_12$–2–2), ($_12$–2–2), ($_12$–2–2), ($_12$–2–2), ($_12$–2–2), ($_12$–2–2) | ? | |
| Ind 8 (MHNCI 114) | ($_12$–2–2), ($_12$–2–2), ($_12$–2–2), ($_12$–2–2), (2–2–2), (2–2–2), (2–2–2), (2–2–2), (2–2–2), (1–2–2) | ? | |
| Ind 9 (MHNCI 115) | (2–2), ($_12$–2), ($_12$–2), ($_12$–2), ($_12$–2), ($_12$–2), ($_12$–2), ($_12$–2), ($_12$–2), ($_12$–2), ($_12$–2), ($_12$–2), ($_12$–2) ($_11$–2), $_11$, $_11$ | ? | |
| Ind 10 (MHNCI 116) | (2–3–3), ($_12$–3–3), ($_12$–3–3–2), ($_12$–3–2–2), (2–2–2–2), (2–2–2–2), (2–2–2–2), (2–2–2–2), (2–2–2), (2–2–1) | ? | |
| Ind 11 (MHNCI 117) | ($_12$–2–2), ($_12$–2–2–2), ($_12$–2–2), ($_12$–2–2), ($_12$–2–2–2), ($_12$–2–2–2), ($_12$–2–2–2), ($_12$–2–2–2) | ? | |
| Ind 12 (MHNCI 118) | ($_12$–2), ($_12$–2–2), ($_12$–2–2), ($_12$–2–2), ($_12$–2–2), ($_12$–2–2), ($_12$–2–2), ($_12$–2–2), ($_12$–2–2), ($_12$–2–2), ($_12$–2–2–2) | ? | |
| Ind 13 (MHNCI 119) | $_13$, ($_13$–2), ($_12$–2), ($_12$–2), ($_13$–3), ($_13$–3), ($_13$–3), ($_13$–3), ($_23$–3), ($_23$–3–3), ($_13$–3–3), ($_12$–2–2), ($_23$–3–3), ($_13$–3–3), ($_13$–3–3), ($_13$–3–3) | ? | |
| Ind 14 (MHNCI 120) | ($_12$–2), ($_12$–2–2), ($_12$–2–2), ($_12$–2–2), ($_12$–2–2), ($_12$–2–2), ($_12$–2–2), ($_12$–2–2), ($_12$–2–2), ($_12$–2–2–2), ($_12$–2–2–2) | ? | |
| Ind 15 (MHNCI 121) | ($_12$–2–2–2), ($_12$–2–2–2), ($_12$–2–2–2), ($_12$–2–2–2), ($_12$–2–2–2), ($_12$–2–2–2), ($_12$–2–2–2), ($_12$–2–2–2) | ? | |
| Ind 16 (MHNCI 122) | ($_12$–2), ($_12$–2), ($_12$–2), ($_12$–2), ($_12$–2), ($_12$–2), ($_12$–2–2), ($_12$–2–2), ($_12$–2–2), ($_12$–2–2), (2–2–2–2), ($_12$–2–2–2), ($_12$–2–2), ($_12$–2–2), ($_12$–2–2–2), ($_12$–2–2–2), ($_12$–2–2) | ? | |

*(Continued)*

| Individuals (ind) and call deposit number | Structure | A | B |
|---|---|---|---|
| *B. ferruginus* (Olimpo, Serra do Marumbi, municipality of Morretes, Paraná) | | | |
| Ind 1 (MHNCI 290) | $_1$2, $_1$2, $_1$3, $_1$3, (3–3), (3–3), (3–3), (3–3), ($_1$3–3), (3–3), (3–3) | 1 | |
| Ind 1 (MHNCI 291) | 1, 2, 2, 2, 3, 3, (3–3), 3 | 1 | |
| Ind 2 (MHNCI 290) | 1, 1, $_1$2, 2, 2, 2, 3, 3, $_1$3, $_1$2, $_1$3, 2, $_1$3, $_{1-1}$2, ($_1$2–3), $_{1-1}$2, $_1$2 | 0 | |
| Ind 3 (MHNCI 292) | 2, 2, 2, 3, 2, 2, (3–3), (2–3), 3 | 3 | |
| Ind 4 (MHNCI 293) | 2, 3, 3, 3, 3, 3, 3, 3, 3 | ? | |
| Ind 5 (MHNCI 294) | 2, 2, 2, 2, 3, 3, 3, (3–3), 3, (3–3), (3–3), (2–3), (2–3), (2–3) | 4 | |
| Ind 6 (MHNCI 295) | 1, 1, 2, $_1$3, 3, 2, 2, 2, $_{1-1}$2, 2, $_1$3, $_1$2, ($_1$2–2), ($_1$3–3), $_1$2 | 1 | |
| Ind 6 (MHNCI 297) | $_1$2, $_1$2, $_1$2, $_1$2, 2, 2, $_1$2, $_1$2, $_1$2, $_1$2, $_1$2, $_1$2 | 2 | |
| Ind 6 (MHNCI 298) | $_1$1, 1, $_1$1, $_1$2, $_1$2, $_1$2, $_1$2, $_1$2, $_1$2, $_1$2, $_1$2, $_1$2, $_1$2, $_1$2, $_1$2, $_1$2, $_{1-1}$2, $_{1-1}$2 | 1 | |
| Ind 6 (MHNCI 299 and 300) | 1, 1, $_{1-1}$1, $_1$2, $_1$2, $_1$2, $_1$2, $_1$2, $_1$2, $_1$2, $_1$2, $_1$2, $_{1-1}$2, $_1$2, $_1$2, $_{1-1-1}$2, $_1$2, $_{1-1}$2, $_{1-1}$2 | 0 | |
| Ind 6 (MHNCI 301) | $_1$1, $_1$2, $_1$2, $_1$2, ($_1$2–2), ($_1$2–2), ($_1$2–2), $_1$2, $_1$2, ($_{1-1}$2–2), 2, $_1$2 | 2 | |
| Ind 7 (MHNCI 296) | 1, 1, 2, $_1$2, $_1$3, $_1$2, $_1$3, $_1$3, $_1$3, $_1$3, $_1$3, $_1$3, $_1$3, $_1$3, $_1$3, $_1$3, $_1$3, $_1$3, $_1$3 | 1 | |
| *B. fuscolineatus* (Morro Braço da Onça, municipality of Luiz Alves, Santa Catarina) | | | |
| Ind 1 (MHNCI 345) | 1, 1, 1, 1, 2, 2, 2, 2, 2, 2, 2, 2, (2–2), (2–2), (2–2), (?–?), (2–2), (2–2), (2–2), (2–2) | 0 | |
| Ind 1 (MHNCI 357) | 2, 1,1 1, 1, 1, 1 | 1 | |
| Ind 2 (MHNCI 345) | ?, 1, 2, 1, 2 | 1 | x |
| Ind 2 (MHNCI 346) | 1, 1, 2, 2, 2, 2, 2, 2, 2, 2, (2–2), (3–2), (3–3), (3–2), (3–3), (2–3), (2–2), (2–3), (2–2), (2–2), (2–3), (3–2), (2–2) | 0 | |
| Ind 2 (MHNCI 358) | 1, 1, 1, 1, 2, 2, 2, 2, 2, 2, 2, 2, 2, (2–3), (3–3), (3–3), (3–3), (3–3) | 0 | |
| Ind 2 (MHNCI 359 | 1, 1, 1, 1, 2, 2, 2, 2, 2, (2–2), (2–2), (3–2), (3–2), (2–2) | 0 | |
| Ind 3 (MHNCI 360) | 2, 2, 2, 2, 2, (2–2), (2–2), (2–2), (2–2), (2–2), (2–2), (2–2), (2–2), (2–2), (2–2), (2–2) | 2 | |
| Ind 3 (MHNCI 361) | 1, 1, 1, 1, 1, 1, 1, 2, 2, 1, 1, 1, 1, (2–2), (2–2), (2–2), (2–2), (1–1) | 1 | |
| Ind 4 (MHNCI 362) | 1, 1, 1, 1, 1, 2, 2, 2, 2, 2, 2, 2, (2–2), (2–2), (2–2), (2–2), (2–2), (2–2), (2–2), (2–2), (2–2), (2–2), (3–2), (2–2), (2–2), (2–2) | 0 | |
| Ind 5 (MHNCI 363) | 1, 1, 2, 2, 2, 2, 2, (2–2), (2–2), (2–2), (2–2), (2–2), (2–2), (2–2–2), (3–2–2) | 0 | |
| *B. fuscolineatus* (Morro do Baú, municipality of Ilhota, Santa Catarina) | | | |
| Ind 1 (MHNI 370) | 1, 1, 2, 2, 2, 2, 2, 2, 2, 2, 2, 2, 2 | ? | |
| Ind 2 (MHNI 371) | 2, 2 | ? | ? |
| *B. izecksohni* (Torre da Prata, Serra da Prata, boundary of the municipalities of Morretes, Paranaguá, and Guaratuba, Paraná) | | | |
| Ind 1 (MHNCI 256) | ($_1$?–?), ($_1$2–2), ($_1$2–2) | ? | |
| Ind 1 (MHNCI 257) | $_1$2, $_1$2, $_1$2, $_1$2, $_1$2, ($_1$2–2), ($_1$2–2), ($_1$2–2), ($_1$2–2–2), ($_1$2–2–2), ($_1$2–2–2), ($_2$2–2–2), ($_1$2–2–2), ($_1$2–2–2), ($_1$2–2–2), ($_1$2–2–2), ($_{1-1}$2–2), ($_1$1–2), ($_1$1–2), ($_1$1–2) | 0 | |
| Ind 1 (MHNCI 258) | $_1$1, 1, 1, 2, $_1$2, $_1$2, $_1$2, $_1$2, $_{1-1}$2, $_{1-1}$2, $_{1-1}$2, $_{1-1}$2, ($_1$2–2), ($_{1-1}$2–2), ($_{1-1}$2–2), ($_{1-1}$2–2), ($_{1-1}$2–2), ($_{1-1}$2–2), ($_1$2–2–1), ($_1$2–2–2), ($_1$2–2–2), ($_1$2–2), ($_{1-1}$2–2), ($_{2-2}$2–2), ($_{1-1}$2–2), ($_{1-2}$2–2–2), ($_{1-1}$2–2–2), ($_{1-1}$2–2–2), ($_{1-1}$2–2–2) | 0 | |
| *B. leopardus* (Morro dos Perdidos, municipality of Guaratuba, Paraná) | | | |
| Ind 1 (MHNCI 343) | 1, 1, 1, 2, 2, 2, 2, 2, $_1$2, $_1$2, 2, $_1$2, $_1$3, $_1$3, $_1$3, $_1$2, $_1$2, 2, 2, $_1$2, 2, $_1$2, $_1$2, ($_1$2–1), ($_1$2–2), (2–2–2), ($_1$2–2–2), ($_1$2–2–2), ($_1$2–2–2), ($_1$2–2–2), ($_1$2–2–2), ($_1$2–2–2–1), ($_1$2–2–2), ($_1$2–2–2), ($_1$2–2–2), ($_1$2–2–2), ($_1$2–2–2$_{-1}$), ($_1$2–2–1) | 0 | |
| Ind 2 (MHNCI 344) | ($_1$2–3), ($_1$2–2–1), (3–3–2), (3–3–2), ($_1$3–2–2), (3–2–2), ($_1$2–2–2$_{-1}$), (3–3–2), (2–2–2), ($_1$2–2) | ? | |
| Ind 3 (MHNCI 372) | (2–2), (2–2–1), (2–2), (2–2–1), (2–2–2), (2–2), (2–2) | ? | |
| Ind 4 (MHNCI 373) | $_1$1, $_1$1, $_1$1, $_1$1, 1, 1, $_1$1, 1, $_1$1, $_1$1, $_1$1, $_1$2, $_1$2 | ? | |
| Ind 5 (MHNCI 374) | 2, (2–2), (2–2), (3–2), (3–3), (3–3), (3–2), (3–3), (3–2), (3–3), (2–3), (3–3–2), (3–3–1) | ? | |
| *B. leopardus* (Serra do Araçatuba, municipality of Tijucas do Sul, Paraná) | | | |

*(Continued)*

**Table 4.** (Continued)

| Individuals (ind) and call deposit number | Structure | A | B |
|---|---|---|---|
| Ind 1 (MHNCI 337) | 1, 1, 1, 1, 1, $_1$1, $_1$1, $_1$1, ($_1$1–1), ($_1$1–$_1$1), ($_1$2–1), (2–2), (2–2), ($_1$2–2), ($_1$2–2), ($_1$2–2–1), ($_1$2–2–1), ($_1$2–2–2), ($_1$2–2–2), ($_1$2–2–2–1), ($_1$2–2–2–1), ($_1$2–$_1$2–2–1), ($_1$2–2–2–1), ($_1$2–$_1$2–2–1), ($_1$2–2–2–1), ($_1$2–2–2–2) | 2 | |
| Ind 2 (MHNCI 338) | ($_1$2–2–2), ($_1$2–2–2), ($_1$2–2–2–2), ($_1$2–2–2), ($_1$2–2–2) | ? | |
| Ind 2 (MHNCI 341) | 1, 1, (1–1), (1–1), (1–2), (2–2), (2–2), (1–2–1), (2–2–1), (2–2–2), (2–2–1), ($_1$2–2–2), (2–2–2–1), (2–2–2–1), (2–2–2–2), (2–2–2–1), ($_1$2–2–2–2), ($_1$2–2–2–2), ($_1$2–2–2–2), (2–2–2–2–1), ($_1$1–2–2–2–1), ($_1$2–2–2–1–1), (1–2–2–1) | 2 | |
| Ind 3 (MHNCI 339) | 1, 2, $_1$2, 2, 2, $_1$2, 2, $_1$2, 2, 2, 2, 2, 2, 2, (2–2), ($_1$2–2), ($_1$2–2), ($_1$2–2), ($_1$2–2), ($_1$2–2), ($_1$3–2–2), ($_1$2–2–2), ($_1$2–2–2), ($_1$2–2–2), ($_1$2–2–2), ($_1$2–3–2), (2–2–2), ($_1$2–2–2), ($_1$2–2–2), ($_1$2–2), ($_1$2–2), ($_1$2–2–2), ($_1$2–2–2) | 2 | |
| Ind 3 (MHNCI 340) | $_1$1, $_1$2, $_1$2, 2, $_1$2, $_1$2, $_1$2, ($_1$2–2), ($_1$2–2), ($_1$2–2), ($_1$2–2), ($_1$2–2), ($_1$2–2), ($_1$2–2–2), ($_1$2–2–2), ($_1$2–2–2), ($_1$2–2–2), ($_1$2–2–2), ($_1$2–2–2), ($_1$2–2–2) | 2 | |
| Ind 4 (MHNCI 342) | 2, 2, $_1$2, 2, (2–2), 2, (2–2), (2–2), (2–2), (2–2), ($_1$2–2), (2–2–2), (2–2–2), ($_1$2–2–2–$_1$), ($_1$2–2–2), (2–2–2), (2–2–2–2), ($_1$2–2–2–2), ($_1$$_1$2–2–2–2), ($_1$2–2–2–2), ($_1$2–2–2–2–$_1$), (2–2–2–2–1), ($_1$1–2–2–2–2), $_1$1, 1 | 2 | |
| Ind 5 (MHNCI 364) | $_1$1, $_1$1, $_1$1, $_1$1, $_1$1, $_1$1, $_1$1, $_1$1, $_1$2, $_1$1, $_1$1, ($_1$2–1), ($_1$2–2), ($_1$2–2), ($_1$2–2), ($_1$2–2), ($_1$2–2), ($_1$2–2–1), ($_1$2–2–2), ($_1$2–2–2), ($_1$2–2–2), ($_1$2–2–2), ($_1$2–2–2), ($_1$2–2–2), ($_1$1–2–1), ($_1$1–2–1) | 0 | |
| *B. lulai* sp. nov. (Pico Garuva, municipality of Garuva, Santa Catarina) | | | |
| Ind 1 (MHNCI 224) | $_1$3, $_1$3, $_1$3, $_1$3, $_1$3, $_1$2, $_1$3 | ? | |
| Ind 2 (MHNCI 225) | $_1$1, $_1$1, $_1$2, $_1$2, $_1$2, $_1$2, $_1$2, $_1$3, $_1$2, $_1$2, $_1$2, $_1$2, $_1$2, $_1$2, $_1$2, $_1$2, $_1$2, $_1$2, $_1$2, $_1$2 | ? | |
| Ind 3 (MHNCI 226) | $_1$2, $_1$3, $_1$3, $_1$3, $_1$3, $_1$3, $_1$3, $_1$3, $_1$3, $_1$3, $_1$3, $_1$3, $_1$3, $_1$3, $_1$3 | ? | |
| Ind 4 (MHNCI 227) | $_1$3, $_1$3, $_1$3, $_1$3, $_1$3, $_1$3, $_1$3, $_1$3, $_1$3, $_1$3, $_1$3, $_1$3, $_1$3, $_1$2, $_1$2 | ? | |
| Ind 5 (MHNCI 228) | $_1$3, $_1$3, $_1$3, $_1$3, $_1$$_1$3, $_1$$_1$3, $_1$$_1$3 | ? | x |
| Ind 6 (MHNCI 229) | $_1$1, $_1$1, $_1$1, $_1$1, $_1$2, $_1$3, $_1$3, $_1$2, $_1$2, $_1$3, $_1$3, $_1$3, $_1$3 | ? | |
| Ind 7 (MHNCI 230) | $_1$2, $_1$2, $_1$2, $_1$3, $_1$2, $_1$3, 2, $_1$2, $_1$2 | ? | |
| Ind 8 (MHNCI 231) | $_1$3, $_1$3, $_1$3, $_1$4, $_2$4, $_1$3, $_2$4, $_1$3 | ? | |
| Ind 9 (MHNCI 232) | $_1$2, $_1$2, $_1$2, $_1$2, $_1$2, $_1$3, $_1$3, $_1$3, $_1$2, ($_1$2–2), ($_1$3–3), ($_1$3–3), ($_1$3–3), ($_1$3–2), $_1$2, 2, $_1$2, $_1$2, $_1$2 | ? | |
| *B. lulai* sp. nov. (Monte Crista, municipality of Garuva, Santa Catarina) | | | |
| Ind 1 (MHNCI 233) | 2, 2, 2, 2, 2, 2, 2, 2, 2, 2, $_1$1, $_1$1, $_1$1, $_1$2, $_1$2, $_1$2, $_1$2, $_1$2, $_1$2, $_1$2, $_1$2, $_1$2, $_1$2, $_1$2, $_1$2, $_1$2, $_1$2, $_1$2 | ? | |
| Ind 2 (MHNCI 234) | $_1$2, $_1$2, $_1$2, $_1$2, $_1$2, $_1$3, $_1$3, $_1$3, $_1$3, $_1$3, $_1$3, $_1$3, $_1$3, $_1$3, $_1$3, $_1$3, $_1$3, $_1$2 | ? | |
| Ind 3 (MHNCI 235) | $_1$3, $_1$3, $_1$3, $_2$3, $_2$3, $_2$3, $_2$3, $_2$3, $_1$3, $_1$3, $_1$3, $_1$3, $_1$3, $_1$3, $_1$3, $_1$3, $_1$3, $_1$3, $_1$3 | ? | |
| Ind 4 (MHNCI 236) | $_1$2, $_1$2, $_1$2, $_1$2, $_1$2, $_1$2, $_1$2, $_1$2, $_1$2, $_1$3, ($_1$2–2), ($_1$2–2), ($_1$2–2), $_1$3 | ? | |
| *B. olivaceus* (Castelo dos Bugres, municipality of Joinville, Santa Catarina) | | | |
| Ind 1 (MHNCI 324) | 3, 3, 3, 3, 3, $_1$3, $_1$$_1$2, $_1$3, $_1$3, $_1$3, $_1$3, $_1$3, ($_1$3–3), ($_1$3–2), ($_1$3–2), (2–2) | 0 | |
| Ind 2 (MHNCI 325) | 3, 3, 3, 3, 3, 3, 3, (3–3), (3–3), (3–3), (3–3), (3–3), (3–3) | ? | |
| Ind 3 (MHNCI 326) | $_1$2, $_1$2, $_1$2, $_1$2, $_1$2, $_1$2, $_1$2, $_1$2, $_1$2, $_1$2, $_1$2, ($_1$2–2), $_1$2, ($_1$2–2), $_1$2 | 0 | |
| *B. olivaceus* (Morro do Boi, municipality of Corupá, Santa Catarina) | | | |
| Ind 1 (MHNCI 308) | $_1$1, $_2$1, $_1$1, $_1$1, $_3$1, $_1$1, $_1$2, $_1$2, $_1$2, $_1$2, $_1$2, $_1$$_1$2, $_2$$_1$2, $_3$2, $_3$1, $_3$2, $_1$1, $_1$1 | 0 | |
| Ind 2 (MHNCI 309) | 1, 1, $_1$1, 1, 2, $_1$2, 2, 2, $_1$2, $_1$2, $_1$2, $_1$2, $_1$2, $_1$2, $_1$2, 3, $_1$2, $_1$2, $_1$2 | 0 | |
| Ind 2 (MHNCI 316) | $_1$$_1$1, $_1$2, $_1$2, $_1$$_1$1, $_1$2, $_1$2, 1, 1, $_1$1, $_1$2, $_1$2, $_1$$_1$1, $_1$$_1$2, $_1$$_1$2 | 0 | |
| Ind 3 (MHNCI 310) | $_1$2, $_2$2, $_1$2, $_1$2 | ? | |
| Ind 4 (MHNCI 311) | $_1$2, $_1$2, $_1$2, 2, (2–2) | 3 | |
| Ind 5 (MHNCI 311) | 2, $_1$2, 2, 2, 2, (2–1), 2 | ? | |
| Ind 5 (MHNCI 312) | 2, 2, 2, 2, 2, 2, 2, (2–1), (2–2), 2 | ? | |
| Ind 6 (MHNCI 313) | 2, 2, $_1$2, 2, 2, 2 | ? | |
| Ind 7 (MHNCI 314) | $_1$2, 2, 2, 2, 2, 2, 2, 2, $_1$2, $_1$2, 2, $_1$2, $_1$2, 2, 2, 2 | ? | |
| Ind 8 (MHNCI 315) | 1, $_1$1, $_1$1, $_1$1, $_1$1, $_1$2, $_1$2, $_1$2, $_1$2, $_2$2, $_2$2, $_1$2, $_1$2, $_2$2, $_1$2 | 0 | |
| Ind 9 (MHNCI 316) | 1, $_1$1, $_1$$_1$1, $_1$1, $_1$$_1$1, $_2$$_1$1, $_1$$_1$2, $_2$$_1$2, $_1$$_1$2, $_1$2, $_3$1, $_1$2 | 0 | |
| Ind 9 (MHNCI 316) | (2–3), ($_1$2–3), (2–2) | ? | |

*(Continued)*

| Individuals (ind) and call deposit number | Structure | A | B |
|---|---|---|---|
| *B. olivaceus* (Pico Jurapê, municipality of Joinville, Santa Catarina) | | | |
| Ind 1 (MHNCI 317) | $_12$, 2, 2, 2, 2, 2, 2, $_12$, $_12$ | ? | |
| Ind 2 (MHNCI 318) | 1, 2, 1, 2, 2, 2, 2, 2, 2, 3, $_13$, 3, 3, 3, 3, 3, 3, 1 | 1 | |
| Ind 3 (MHNCI 319) | 1, 1, 1, 1, 2, 1, 2, 2, 2, 2, 2, 2, 2, 2, 2, 2, 2, 2, 2, 2 | 2 | |
| Ind 4 (MHNCI 320) | $_12$, $_12$, $_12$, 2, $_12$, $_12$, 2, $_12$ | 1 | |
| Ind 4 (MHNCI 321) | $_11$, 2, $_12$, 2, $_12$, $_12$, $_12$, $_12$, $_12$, $_13$, $_12$, $_12$, $_13$, $_13$, $_13$, $_12$, $_12$, $_13$, $_13$, $_13$, $_13$ | 1 | |
| Ind 5 (MHNCI 322) | $_12$, $_12$, $_12$, $_12$, $_12$, $_12$, $_12$, $_12$, $_12$, $_12$, $_13$, $_13$, $_{1-1}3$, $_{1-1}2$, $_13_{-1}$, $_13_{-1}$, $_{1-1}3$, $_13$, $_13$, $_13$, $_{2-1}3$, $_13$, $_13$, $_13$, $(_{1-1}3–2)$, $_12$, $_12$, $_12$ | 1 | |
| Ind 6 (MHNCI 323) | 2, $_11$, $_12$, $_12$, $_12$, $_12$, $_12$, $_12$, $_12$, $_13$, $_12$, $_13$, $_13$, $_13$, $(_13–3)$, $(_13–2)$, $(_12–3)$ | 3 | |
| Ind 7 (MHNCI 375) | 1, 2, $_12$, 2 | 1 | x |
| *B. pernix* (Anhangava, Serra da Baitaca, municipality of Quatro Barras, Paraná) | | | |
| Ind 1 (MHNCI 251) | $_14$, $_13$, $_14$, $_13$, $_13$, $_13$, $_13$, $_13$, $_13$, $_13$, $_14$, $_13$, $_13$, $_14$, $_13$, $_13$, $_13$ | ? | |
| Ind 2 (MHNCI 252) | $_{1-1}3$, $_{1-1}3$, $_{1-1}3$, $_{1-1}3$, $_{1-1}3$, $_{1-1}3$, $_{1-1}3$, $_{1-1}3$, $_{1-1}3$, $_{1-1}3$, $_{1-1}3$ | ? | |
| Ind 3 (MHNCI 253) | $_12$, $_13$, $_13$, $_13$, $_13$, $_13$, $_13$, $_13$, $_13$, $_{1-1}3$, $_13$, $_13$, $_13$, $_13$, $_13$, $_13$, $_33$ | ? | |
| Ind 4 (MHNCI 254) | $_23$ $(_13–3)$, $(_13–3)$, $(_13–3)$, $_13$, $(_13–3)$, $(_13–3)$, $(_13–3)$, $_13$, $(_13–3)$ | ? | |
| Ind 5 (MHNCI 255) | $_13$, 3, 3, 3, 3, 3, 3, 3, 3, 3, 3, 3, 3, 3, 3, 3, $_13$ | ? | |
| Ind 6 (MHNCI 376) | (2–2), 2, (2–2), 2, $_12$, $_12$, $_12$ | ? | |
| Ind 7 (MHNCI 377) | $_13$, 3, $_13$, $_13$, $_13$, $_13$, $_13$, $_13$, $_13$, $_13$, $_13$, $_13$, $_13$ | ? | |
| Ind 8 (MHNCI 378) | $_13$ $_13$, $_13$, $_13$, $_13$, $_13$, 3, $_13$, $_13$, $_13$, $_13$, $_13$ | ? | |
| *B. pombali* (Morro dos Padres, Serra da Igreja, municipality of Morretes, Paraná) | | | |
| Ind 1 (MHNCI 259) | $_12$, $_12$, $_12$, $_22$, $(_12–2)$, $(_12–2)$, $(_12–2)$, $(_12–2)$, $(_13–3)$, $(_12–3)$, $(_13–2)$, $(_13–3)$, $(_13–2–3)$, $(_12–2–2)$, $(_22–2–2)$, $(_12–2–2)$, $(_12–2–2)$, $(_12–2–2)$, $(_12–2–2)$, $(_12–2–2)$, (2–2–2–2), $(_12–2–2)$, $(_12–2–2)$ | ? | |
| Ind 2 (MHNCI 260) | $_12$, $(_12–2)$, $(_12–2)$, $(_12–2)$, $(_12–2)$, $(_12–2)$, $(_12–2)$, $(_12–2–_12)$, $(_12–2–2)$, $(_12–2–2)$, $(_12–2–2)$, $(_12–_12–2)$, $(_12–2–_1\,2)$, $(_12–2–2)$, $(_{2-1}2–2–2)$, $(_{1-1}2–1–1)$, 1, $_{1-1}1$ | ? | |
| Ind 3 (MHNCI 261) | $_12$, $_12$, $_12$, $_12$, $_12$, $_12$, $(_12–2)$, $(_12–2)$, $(_12–2)$, $(_12–2)$, $(_12–2)$, $(_12–2–2)$, $(_12–2–2)$, $(_12–2–2)$, $(_12–2–2)$, $(_12–2–2)$, $(_12–2–2)$, $(_12–2–2)$, $(_12–2–2)$, $(_23–3–3)$, $(_12–2)$ | 3 | |
| Ind 4 (MHNCI 262 | 1, 1, 2, 2, 2, 2, 2, 2, 2, (2–2), (2–2), (3–2), (2–2), (2–2), (2–2), (2–2), (2–2), (2–2–1), (2–2–1), (2–2), $(_12–2)$, $(_12–2)$, $(_12–1)$, $(_12–2)$, $(_12–2–1)$ | 2 | |
| Ind 5 (MHNCI 262) | $_12$, $(_12–2)$, $_12$, $(_12–2)$, $(_12–2)$, $(_12–2)$, $(_12–2)$, $(_12–2–2)$, $(_12–2–2)$, $(_12–2–2)$, $(_12–2–2)$, $(_12–2–2)$, $(_12–2–2)$, $(_12–2–2)$, $(_12–2–2)$, $(_12–2–2)$ | 0 | |
| Ind 6 (MHNCI 263) | $(_12–2–2–2)$, $(_12–2–2–2)$, $(_12–2–2–2)$ | ? | |
| Ind 7 (MHNCI 263) | 2, 3, 3, 3, 3, 3, 3, 3, 3, 3 | ? | ? |
| Ind 8 (MHNCI 264) | $(_12–2–2)$, $(_12–2–2)$ | ? | |
| *B. quiririensis* (Campos do Quiriri, Serra do Quiriri, on the border between the municipalities of Campo Alegre and Garuva, Santa Catarina) | | | |
| Ind 1 (FNJV 0040992) | $_14$, $_14$, $_14$, $_{1-1}4$, $_14$, $_14$ | ? | ? |
| *B. quiririensis* (Serra do Quiriri, municipality of Campo Alegre, Santa Catarina) | | | |
| Ind 1 (MHNCI 327) | $_12$, $_12$, $_12$, $_13$, $_12$, $_12$, $_12$, $_12$, $_12$, $_12$, $_12$, $_23$, $_12$, $_12$, $_12$, $_12$, $_12$ | ? | x |
| Ind 1 (MHNCI 328) | $_12$, $_12$, $_12$, $_12$, $_12$, $_12$ | ? | |
| Ind 2 (MHNCI 327) | 1, $_{1-1}1$, $_{1-1}1$, $_12$, $_12$, $_12$, $_12$, $_12$, $_12$, $_12$ | 0 | x |
| Ind 3 (MHNCI 333) | 2, 2, 2, 2, 2, 3, 3, 3, 3, 3, 3, 3, 3, $_13$, 3, $_22$, $_32$ | ? | x |
| Ind 4 (MHNCI 334) | $_12$, $_12$, $_12$, $_22$, 2, 2, $(_23–3)$, $(_12–2)$, $(_22–2)$, (2–3), 2 | 2 | |
| Ind 4 (MHNCI 335) | 1, $_{1-1}1$, $_{1-1}1$, $_{1-1}1$, $_{1-1}1$, $_12$, $_12$, 2, $_12$, 2, 2, 2, $_12$, 2, $_22$, 2, 2, $_12$, $_12$, 2, 2, 2, $_12$, 2, $_12$, 2, 2 | 0 | |
| Ind 4 (MHNCI 336) | 1, 1, 1, 1, 1, 2, 2, 2, $_12$, $_12$, $_{1-1}2$, $_{1-1}2$, $_13$, $_{1-1}3$, $(_{1-1}2–3)$, (2–3), $(_12–2)$, $(_12–3)$, $(_12–3)$, $(_12–3)$, $(_12–3)$ | 0 | |
| *B. quiririensis* (Bradador, Serra do Quiriri, municipality of Garuva, Santa Catarina) | | | |
| Ind 1 (MHNCI 329) | $_12$, $_12$, $_13$, $_13$, $_{1-1}3$, $_13$, 2, $_13$, $_12$, 2, 2, 3, 2, 2 | ? | x |

*(Continued)*

**Table 4.** (Continued)

| Individuals (ind) and call deposit number | Structure | A | B |
|---|---|---|---|
| Ind 2 (MHNCI 330) | $_1$3, $_1$3, $_1$2, $_1$3, $_1$3, 3, 3, $_1$3, $_1$3, $_1$3 | ? | |
| Ind 3 (MHNCI 331) | $_1$3, $_1$3, $_1$3, $_1$3, 3, $_1$3, $_1$3, $_1$3, $_1$3, $_1$3, $_1$3, $_1$3, $_1$3, $_{1\text{-}1}$3, $_{1\text{-}1}$3 | ? | |
| Ind 4 (MHNCI 332) | $_1$3, $_{1\text{-}1}$3, $_1$3, 3, $_1$3, 3, $_1$3, 3, $_1$3, $_1$3, 3, $_1$3 | ? | x |
| *B. tabuleiro* (Afluente da margem direita do rio do Ponche, Serra do Tabuleiro, municipality of São Bonifácio, Santa Cataria) | | | |
| Ind 1 (CASA 154) | 3, 3, 3, 3, 3, 3, 3, 3, 3 | ? | ? |
| Ind 2 (CASA 155) | $_1$3, $_1$3, $_1$3, 3, $_1$3, $_1$3, 2, $_1$3 | ? | ? |
| Ind 3 (CASA 156) | 3, 3, 3, 3, 3, 3, 3, 3, 3, $_1$3, 2, $_1$3, $_1$3, $_1$4 | ? | ? |
| Ind 3 (CASA 157) | $_1$4, $_1$4, ($_1$4–3), ($_1$4–3), (4–3), ($_1$4–4), ($_1$4–3), ($_1$4–3), ($_1$4–3), ($_1$3–3) | ? | ? |
| Ind 4 (MHNCI uncatalogued) | $_1$1, 1, $_1$1, $_1$2, $_1$2, $_1$2, $_1$2, $_1$2, $_1$2, $_1$2, $_1$2, $_1$2, $_1$2, $_1$2, $_1$2, $_1$2, $_1$2, $_{1\text{-}1}$2, $_1$2 | 3 | |
| Ind 5 (MHNCI uncatalogued) | 1, $_1$1, 1, $_1$1, $_1$1, $_1$1, $_1$2, $_1$2, $_1$2, $_1$2, $_1$2, $_1$2, $_1$2, $_1$2, $_1$2, $_1$2, $_1$2, $_{1\text{-}1}$2, $_1$2, $_1$2, $_1$2, $_{1\text{-}1}$2, $_1$2, $_{1\text{-}1}$3, $_{1\text{-}1}$2, $_{1\text{-}1}$2, $_{1\text{-}1}$3 | 1 | |
| Ind 5 (MHNCI uncatalogued) | 1, $_{1\text{-}1}$1, $_1$1, 2, $_1$2, $_1$2, $_1$2, $_1$2, $_1$2, $_1$2, $_1$2, $_1$2, $_1$2, $_1$2, $_1$2, $_1$2, $_1$2, $_1$3, $_1$3, $_1$2, $_1$2 | 1 | |
| *B. verrucosus* (Morro da Tromba, municipality of Joinville, Santa Catarina) | | | |
| Ind 1 (MHNCI 237) | 1, 1, 1, 1, 1, 1, 2, 2, 2, 1, 2, 2, 2, $_1$2, $_1$2, 2, 2, 2, 2, $_1$2, ($_1$2–2), ($_1$2–2), ($_1$2–2), ($_1$2–2) | ? | |
| Ind 2 (MHNCI 238) | 1, 1, 1, 1, 1, 1, 2, 2, 2, 2, 2, 2, 2, 2, 2, 2, 2, 2, 2, 2, 2, 2, 2, 2, 2, 2, 2, 2, 2, 2, 2, 2, 2, 2, 2, 2, 2, 2, 3 | ? | |
| Ind 3 (MHNCI 239) | 3, 4, 4, 3, $_1$3, 3,?, 3, 3, 3, $_2$3 | ? | |
| Ind 4 (MHNCI 240) | ?,?,?,?, 2, ($_1$2–2),?, $_1$2 | ? | |
| Ind 5 (MHNCI 241) | 1, 1, 1, 1, 1, 1, 2, $_1$2, 2, 2, 2, 2, 2, $_1$2, 2, $_1$2, $_1$2, $_1$2, ($_1$3–2), ($_1$2–2), ($_1$2–2), ($_1$2–2), ($_1$2–2), ($_1$2–2), ($_1$2–2), ($_1$2–2) | ? | |
| Ind 6 (MHNCI 242) | 2, 3, 2, 3, 3, 3, 3, $_1$3, 3, 3, $_1$3, 3, $_1$3, 3 | ? | |
| Ind 7 (MHNCI 243) | 1, 1, 1, $_1$2, $_1$2, 2, $_1$2, 2, $_1$2, $_1$2, $_1$2, 2, 2, $_1$2, $_1$4, $_1$3, $_1$3, $_1$3 | ? | |
| Ind 8 (MHNCI 244) | ($_1$5–7), ($_1$4–7), ($_1$6–7), ($_1$7–6), ($_1$8–7), ($_1$7–7), ($_1$6–7) | ? | |
| Ind 9 (MHNCI 379) | 2,?, 2, 3, 2, 3, 2, 2, 2, (4–4), 2 | 3 | |
| Ind 10 (MHNCI 380) | 1, 2, 2, 2, 3, 3, 3, 3, 3, 3, 3 | 0 | |
| Ind 11 (MHNCI 380) | 3, 2, 2, 3, 2, 2, 3, $_1$3, 2, 2, 3, 3, $_1$3, 3, 4, 4 | 0 | |
| *Brachycephalus* sp. (Serra Canasvieiras, boundary of the municipalities of Guaratuba and Morretes, Paraná | | | |
| Ind 1 (MHNCI 275) | $_1$2, $_1$2, $_1$2, ($_1$2–2), ($_1$2–2), ($_1$2–2), ($_1$2–2), ($_1$2–2), ($_1$2–2), ($_1$2–2), ($_1$2–2), ($_1$2–2), ($_1$2–2), ($_1$2–1) | ? | |
| Ind 2 (MHNCI 276) | $_1$2, $_1$2, $_1$2, ($_1$2–1), ($_1$2–1), ($_1$2–1), ($_1$2–1), ($_1$2–1), ($_1$2–2), ($_1$2–2), ($_1$2–2) | ? | |
| Ind 3 (MHNCI 277) | $_1$2, $_1$2, $_1$2, $_1$2, $_1$2, $_1$2, $_1$2, $_1$2, $_1$2, $_1$2, $_1$2, $_1$2, $_1$2, $_1$2, $_1$2, $_1$2, $_1$2, $_1$2, $_1$2, $_1$2, $_1$2, $_1$2, $_1$2, $_1$2, $_1$2, $_1$2, $_1$2, $_1$2, $_1$2, ($_1$2–2), ($_1$2–2), ($_1$2–2), ($_1$2), ($_1$2–2), ($_1$2), ($_1$2), ($_1$2), ($_1$2), ($_1$2), ($_1$2), ($_1$2), ($_1$2) | ? | |
| Ind 4 (MHNCI 277) | 1, 1, 1, 1, 1, 1, 1, 1, 1, 1, 1, ($_1$2), ($_1$2–2), ($_1$2–2), ($_1$2–2), ($_1$2–2), ($_1$2–2), ($_1$2–2), ($_1$2–2–2), ($_1$2–2–2), ($_1$2–2–2), ($_1$2–2–2), ($_1$2–2–2), ($_1$2–2–2), ($_1$1–2–2) | 0 | |
| Ind 5 (MHNCI 278) | ($_1$1–2), $_1$2, $_1$?, $_1$2, ($_1$2–2–2), ($_1$2–2–2), ($_1$1–2), $_1$1 | ? | |
| Ind 6 (MHNCI 279) | ($_2$2–2), ($_1$2–2), ($_1$2–2–2), ($_1$2–2–2), ($_1$2–2–2), ($_1$2–2–2), ($_1$2–2–2), ($_1$2–2–2) | ? | |
| *Brachycephalus* sp. (Serra do Pico, municipality of Joinville, Santa Catarina) | | | |
| Ind 1 (MHNCI 347) | 1, 1, 1, 2, 2, 2, 2, $_1$2, $_1$3, $_1$3, $_1$3, $_1$3, $_1$4, $_{1\text{-}1}$4, $_1$4, $_{1\text{-}1}$4, $_1$4, 4, $_1$4, ($_1$4–3), ($_1$4–4), $_{1\text{-}1}$4, ($_1$3–3), (3–3), ($_1$3–3), $_1$3 | 0 | |
| Ind 2 (MHNCI 348) | 1, 2, 2, 2, 3, 3, 3, 3, $_1$3, 3, 3, 3, (3–3), (3–3), (3–3) | 2 | |
| *Brachycephalus* sp. (Tupipiá, Serra dos Órgãos, municipality of Antonina, Paraná) | | | |
| Ind 1 (MHNCI 265) | ($_1$?–?), ($_1$?–?), ($_1$?–?–?), ($_1$?–?–?), (?–?–?), (?–?–?) | ? | |
| Ind 2 (MHNCI 265) | ?,?,?,?,?,?,?, $_1$?, $_1$?,?, $_1$?,?, $_1$?, $_1$? | 0 | |
| Ind 3 (MHNCI 266) | ($_1$3–3), ($_{1\text{-}1}$3–3) | ? | |
| Ind 4 (MHNCI 267) | $_1$1, $_1$2, $_1$2 | ? | |
| Ind 4 (MHNCI 268) | $_1$1, $_1$1, $_1$1, $_1$1, $_1$1, $_1$2, $_1$2, $_1$2, $_1$2, $_1$2, $_1$2, $_1$2, $_1$2, $_1$2, $_1$2, ($_1$2–2), ($_1$2–2), ($_1$2–2), $_1$2 | 0 | |

*(Continued)*

**Table 4.** (Continued)

| Individuals (ind) and call deposit number | Structure | A | B |
|---|---|---|---|
| Ind 5 (MHNCI 269) | $_1$1, 1, 1, 1, $_1$1 | ? | |
| Ind 6 (MHNCI 270) | $_1$1, $_1$1, $_1$1, $_1$1, $_1$1, $_1$1, $_1$1, $_1$2, $_1$2, $_1$2, $_1$2, $_1$2, $_1$2, $_1$2, $_1$2, ($_1$2–1), ($_1$2–1), ($_1$2–1), $_1$2, $_1$1, $_1$1 | 0 | |
| Ind 7 (MHNCI 271) | 1, $_1$1, $_1$1, $_1$2, $_1$2, $_1$2, $_1$2, $_1$2, $_1$2, $_1$2, $_1$2, $_1$2, 2, $_1$2, 2, $_1$2, 2, $_1$2, $_1$2, $_1$2, $_1$2, $_1$2, $_1$2, $_2$2, 2, ($_1$2–2), ($_1$2–2), $_1$2, $_1$2 | 0 | |
| Ind 8 (MHNCI 272) | $_1$2, $_1$2, $_1$2, $_1$2, $_1$2, $_1$2, $_1$2, $_1$2, $_1$2, $_1$2, $_1$2, $_1$2, $_1$2, $_1$1 | ? | |
| Ind 9 (MHNCI 273) | $_1$1, $_{1-1}$1, $_{1-1}$1, $_{1-1}$2, $_2$1, $_{1-1}$2, $_1$2, $_{1-1}$2, $_{1-1}$2, $_{1-1}$2, $_{1-1}$2, $_{1-1}$2, $_{1-1}$2, $_{1-1}$2 | 2 | |
| Ind 10 (MHNCI 274) | $_1$2, $_1$2, $_1$2, $_1$2, $_1$2, $_1$2, $_1$2, $_1$3, $_1$2, $_1$2, $_1$3, $_1$3, $_1$2, $_1$3, $_1$2, $_1$3, $_1$3, $_1$2, $_1$2, $_1$3, ($_1$2–3), $_1$3, $_1$3, $_1$3, $_1$3, $_1$3, $_1$3, $_1$3, $_1$3, $_1$3, $_1$3, $_1$3, $_1$3, $_1$3, ($_1$1–3), $_1$2, $_1$2, $_1$3, $_1$3, $_1$3, $_1$3 | 3 | |

Each number represents a note, while the numerical value indicates the number of pulses for each note. Occasionally, it was difficult to count the number of pulses within a note; in such instances, the note was denoted by the use of a question mark (?). Numbers in normal font outside parentheses represent isolated notes, in normal font between parentheses represents note groups, and in subscript represents attenuated notes, that we do not consider forming note groups. Abbreviations: A = number of isolated notes heard being emitted before recording the AC; B = AC possibly interrupted in consequence of an involuntary movement of the researcher; MHNCI = Museu de História Natural Capão da Imbuia, Curitiba, Paraná; CASA = Coleção Audiovisual do Semiárido, Mossoró, Rio Grande do Norte.

## Phylogenetic relationships

The inferred phylogenetic relationships between *B. lulai* sp. nov. and other species of the *B. pernix* species group are shown Fig 10 and the inferred phylogenetic distances between these species are shown in Table 6. Minimum and maximum 16S distances between the tested species are shown in S3 Table. All specimens of *B. lulai* sp. nov. form a clade, with strong support, which itself forms a larger clade together with *B. auroguttatus* and *B. quiririensis*, also with strong support. The relationships between these three species was not resolved.

## Habitat, geographical distribution, and natural history

*Brachycephalus lulai* sp. nov. was recorded in montane Atlantic Forest (*Floresta Ombrófila Densa Montana*; Fig 11), where it lives in the leaf litter. It seems to be locally abundant in the two localities with records, on the southeastern hillside from Serra do Quiriri, municipality of Garuva, southern Brazil: Pico Garuva and Monte Crista, at altitudes between 435–990 m a.s.l. As these localities are 6.3 km apart, it is likely that *B. lulai* sp. nov. populations occur across the intervening forested hillside (Fig 12). We recorded individuals calling throughout the day, but vocal activity decreased during the hottest hours (*c.* 11:00 h – 15:00 h).

We found two parasites located subcutaneously at the left thigh and the gular region of one individual of *B. lulai* sp. nov. (MHNCI 11593) (Fig 13). We tentatively identified these as *Ophiotaenia* sp., a proteocephalid tapeworm, due to the morphological similarity with previously recorded parasites found under similar conditions [58], without apparent diagnostic morphological structures and at a young stage. The known phylogenetic proximity between species of the genus *Brachycephalus*, evidenced in this study and in other previous studies [33,34,59], presumably makes it possible for parasitism by *Ophiotaenia* to occur in all species of the genus. Indeed, *Ophiotaenia* spp. are generalist parasites that use not only amphibians, but also snakes and turtles as hosts [58]. Finally, *Ophiotaenia* parasites have been found in other species of the genus, such as *B. brunneus* and *B. izechsohni*, under the same conditions as those reported in the present study (L.F. Ribeiro, pers. obs).

## Conservation status and Green Status

Although we have altitudinal data for *B. lulai* sp. nov., the polygons of mapped habitat based on isolines and suitable habitat are unrealistic, as they extends for many kilometers beyond the two current records in areas that are highly unlikely

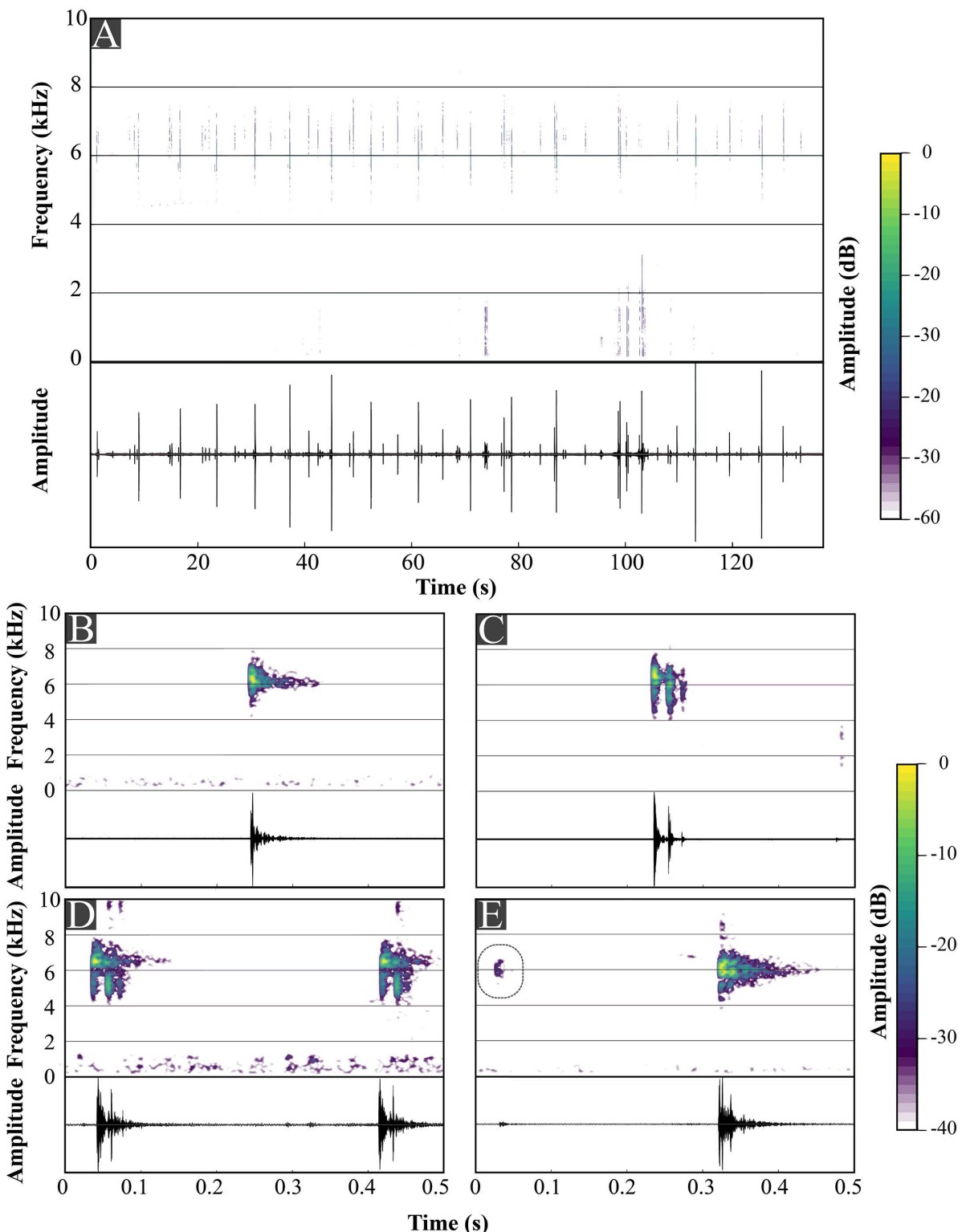

**Fig 8. Example of advertisement call parameters of *Brachycephalus lulai* sp. nov.** (A) Entire advertisement call (MHNCI 226). (B) Isolated note with one pulse (MHNCI 225). (C) Isolated note with three pulses (MHNCI 224). (D) Note group with three and two pulses (MHNCI 232). (E) Attenuated note (black circle) preceding an isolated note with three pulses (MHNCI 226). Spectrograms are produced with a FFT size of 8192 points, Hann window, and overlap of 90% in A and FFT 512 points, Hann window, and overlap of 90% in B–E.

**Table 5. Call parameters described in the advertisement call of *Brachycephalus lulai* sp. nov.**

| Parameter | Values |
|---|---|
| Entire call | |
| (1) [Call duration (s)] | 116.459±45.836 (61.692–201.765) [13/13] |
| (2) Note rate (notes per minute) (excluding attenuated notes) | 7.525±1.659 (5.252–10.557) [13/13] |
| (3) [Number of notes per call (including attenuated notes)] | 30.077±11.485 (14–48) [13/13] |
| (4) [Number of notes per call (excluding attenuated notes)] | 15.769±7.049 (7–30) [13/13] |
| (5) Note duration (s) (of isolated notes and notes of note groups) | 0.033±0.009 (0.010–0.061) [205/13] |
| (6) Number of pulses per notes (of isolated notes and notes of note groups) | 2.449±0.605 (1–4) [205/13] |
| (7) Note dominant frequency (kHz) | 6.194±0.242 (5.168–6.632) [205/13] |
| (8) Highest frequency (kHz) | 6.582±0.239 (6.115–7.235) [205/13] |
| (9) Lowest frequency (kHz) | 5.758±0.295 (4.823–6.288) [205/13] |
| Isolated notes | |
| (10) [Duration of the call including only isolated notes (s)] (when the calls present note groups among isolated notes, only the longest part with isolated notes was counted) | 106.771±50.613 (48.916–201.764) [13/13] |
| (11) Note rate of the call including only isolated notes (notes per minute) | 7.999±2.845 (5.252–16.058) [13/13] |
| (12) [Number of isolated notes per call (excluding attenuated notes)] | 14.538±6.666 (7–30) [13/13] |
| (13) Number of pulses per isolated notes | 2.465±0.616 (1–4) [189/13] |
| (14) Inter-note interval in isolated notes (s) (time from the end of one isolated note to the beginning of the next isolated note) | 8.191±2.252 (4.650–15.005) [173/13] |
| Note groups | |
| (15) Duration of the call including only note groups (s) (only the longest part was counted, when the calls present isolated notes among note groups) | 26.039±10.810 (18.395–33.683) [2/2] |
| (16) Note rate of the call including only note groups (notes per minute) (calculated only when at least two note groups were present) | 17.083±1.049 (16.341–17.825) [2/2] |
| (17) Number of note groups per call | 4.000±1.414 (3–5) [2/2] |
| (18) Number of notes in each note group | 2.000±0.000 (2–2) [8/2] |
| (19) Duration of note group (s) | 0.396±0.020 (0.361–0.416) [7/2] |
| (20) Inter-note group interval (s) (time from the end of one note group to the beginning of the next note group) | 8.215±0.622 (7.532–9.373) [7/2] |
| (21) Inter-note interval within note groups (s) (time from the end of the first note to the beginning of the next note of the same note group) | 0.325±0.021 (0.290–0.346) [8/2] |
| (22) Number of pulses per note in note groups | 2.313±0.473 (2–3) [16/2] |
| Attenuated notes | |
| (23) Number of notes per call associated with attenuated notes | 14.077±5.123 (7–21) [13/13] |
| (24) Number of attenuated notes associated with each note of the call | 1.032±0.177 (1–2) [205/13] |
| (25) Number of pulses in attenuated notes | 1.032±0.177 (1–2) [186/13] |
| (26) Shortest interval between an attenuated note and its associated note (s) | 0.289±0.02 (0.121–0.325) [183/13] |
| (27) Attenuated note dominant frequency (kHz) | 5.713±0.372 (5.167–7.667) [186/13] |
| (28) Attenuated note highest frequency (kHz) | 6.323±0.490 (5.500–8.269) [186/13] |
| (29) Attenuated note lowest frequency (kHz) | 5.209±0.366 (4.392–6.890) [186/13] |

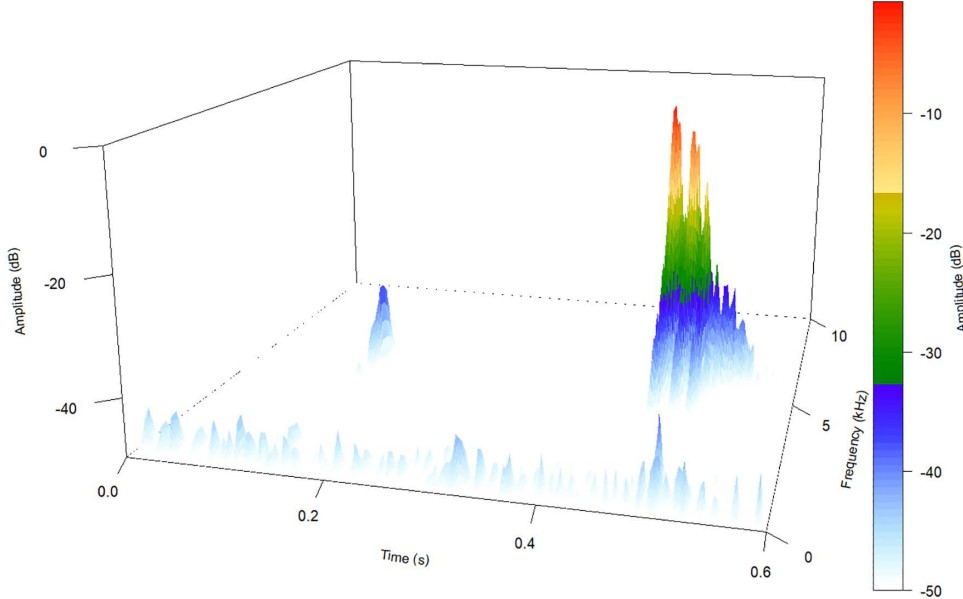

**Fig 9. Example of an attenuated note (smaller, on the left) of *Brachycephalus lulai* sp. nov. followed by an isolated note with three pulses (larger, on the right).** Sound taken from MHNCI 227 and generated with 90 overlap and 256 FFT size by R package Soundshape.

to be occupied. Therefore, its geographical distribution was estimated by overlapping the grid and current records (lower bound and best estimate), resulting in two occupied cells (S13 Fig) and a total area of occupancy of 8 km² (Table 7).

Considering the current environmental quality of its habitat and absence of significant threats in the region of the records, as well as the absence of plausible threats that could affect the species and qualify it for a threatened category in the near future, we propose for it to be considered as Least Concern (LC) (Table 7), even considering its small distribution area. Of the 20 remaining species of *B. pernix* group, six are also assessed as LC, four as Vulnerable (VU), five as Endangered (EN), and five as Critically Endangered (CR) (Table 7; S2–S22 Figs).

Based on field observations of threats and ecology and on its conservation status (see above), we consider that the species is performing its ecological functions at baseline levels, even being naturally small. Therefore, the Current and Long-term Potential Green Scores of *B. lulai* sp. nov. are 100% (classified as Non-Depleted; Table 8). Of the 20 remaining species of *B. pernix* group, nine are also Non-Depleted, two are Moderately Depleted, two are Largely Depleted, and seven are Critically Depleted (Table 8).

A conservation unit could be proposed for the protection of *B. lulai* sp. nov., together with *B. auroguttatus* and *B. qu ririiensis*, as they are distributed close to each other (Fig 14; see S1 File). This conservation unit would have a Weight of 107.5 points (Table 8), a value higher than any other conservation unit that could be created to protect other species of the *B. pernix* group in Paraná and Santa Catarina states (Table 8).

## Etymology

The specific epithet honors Luiz Inácio Lula da Silva, who has been elected President of Brazil on three occasions. Through this tribute, we seek to encourage the expansion of conservation initiatives focused on the Atlantic Forest as a whole, and on Brazil's highly endemic miniaturized frogs in particular.

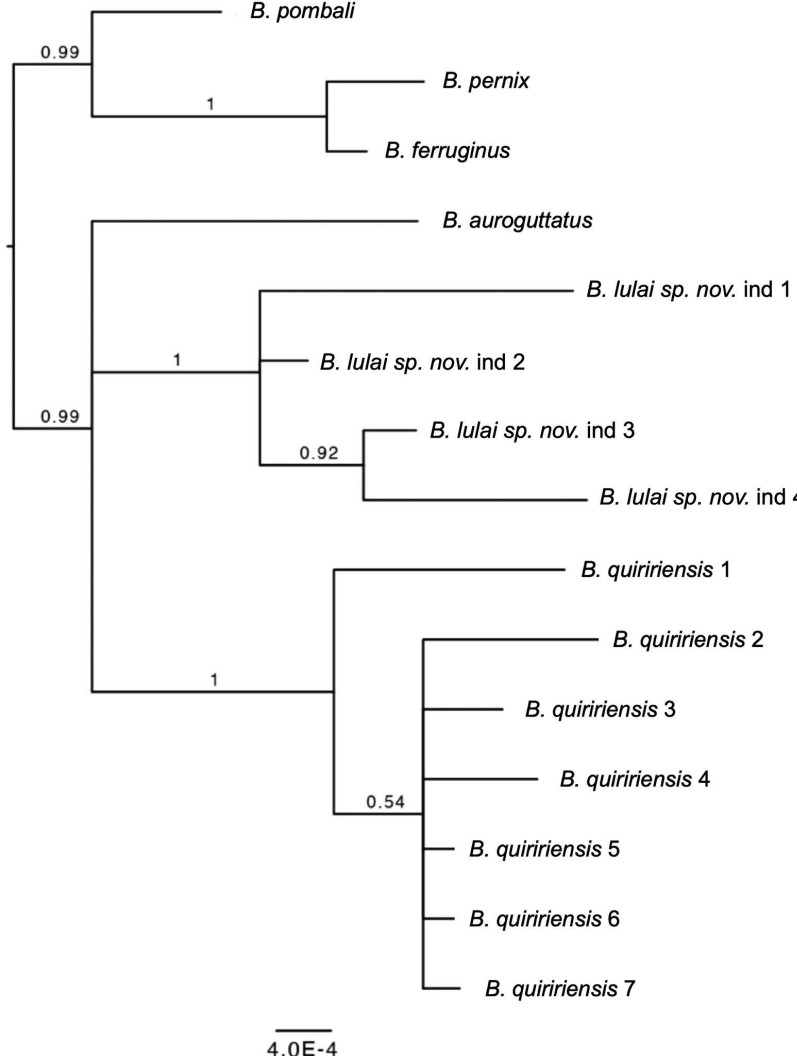

**Fig 10. Relationships between *Brachycephalus lulai* sp. nov. and some species of the *B. pernix* group based on three loci.** Phylogenetic analysis was carried out using Bayesian inference and values above branches correspond to node posterior probabilities. Nodes with posterior probabilities lower than 50% were collapsed.

**Table 6. Pairwise genetic distance (%) between the studied *Brachycephalus* species for all loci (upper diagonal) and 16S alone (lower diagonal).**

| Species | B. auroguttatus | B. ferruginus | B. lulai sp. nov. | B. pernix | B. pombali | B. quiririensis |
|---|---|---|---|---|---|---|
| B. auroguttatus | – | 1.20 | 1.29 | 1.42 | 1.09 | 1.13 |
| B. ferruginus | 0.33 | – | 1.23 | 0.11 | 0.55 | 1.18 |
| B. lulai sp. nov. | 0.44 | 0.33 | – | 1.73 | 1.40 | 1.43 |
| B. pernix | 0.44 | 0.00 | 0.44 | – | 0.77 | 1.53 |
| B. pombali | 0.44 | 0.11 | 0.44 | 0.22 | – | 1.16 |
| B. quiririensis | 0.55 | 0.66 | 0.55 | 0.77 | 0.77 | – |

Recordings were made while calls were in progress (see Table 4). Thus, the measurements of some parameters reflect only the values of the recorded section of the calls. These parameters are indicated in square brackets. Values are expressed, in general, by mean ± SD (range) [sample size/individuals analyzed for each parameter]. For some parameters the mean, SD, and range are not applicable. Abbreviation: SD = standard deviation.

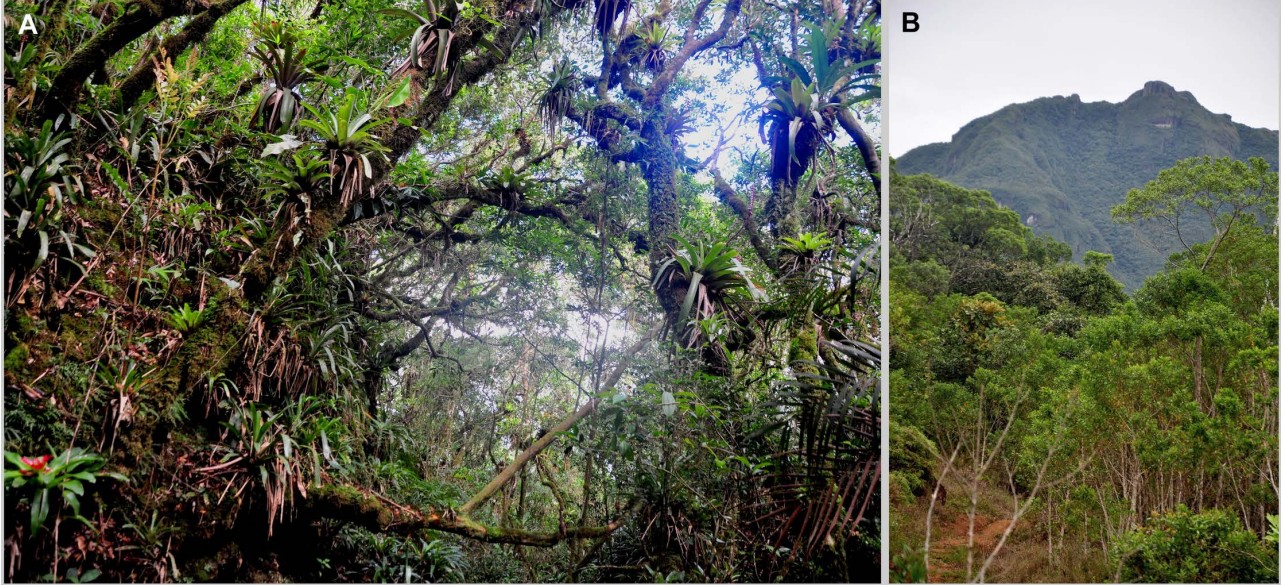

**Fig 11. Habitat at the type locality of *Brachycephalus lulai* sp. nov., Pico Garuva, municipality of Garuva, Santa Catarina, southern Brazil.** (A) Middle and low strata of the forest (*Floresta Ombrófila Densa Montana*) at 750 m above sea level. (B) East slope of Pico Garuva, 1,260 m above sea level. Photographs by Luiz F. Ribeiro.

## Discussion

### Phylogenetic relationships

Although the new species is clearly a member of the *B. pernix* species group, the available phylogenetic evidence indicated in Fig 2 does not allow for inferring whether *B. lulai* sp. nov. is more closely-related to *B. quiririensis* or *B. auroguttatus* and it is likely that a more extensive analysis will be necessary to address this issue. Nevertheless, the phylogeny conclusively shows three important findings: (1) all sequenced individuals within each species are monophyletic; (2) *B. lulai* sp. nov., *B. quiririensis* and *B. auroguttatus* are reciprocally monophyletic (even though only one sample of *B. auroguttatus* was used); and (3) the level of intraspecific genetic variability within *B. lulai* sp. nov. is comparable of that found in *B. quiririensis*. These findings reinforce our interpretation that genetic evidence supports the recognition of *B. lulai* sp. nov. as a new taxon.

### Osteology

All species of the *B. ephippium* group have eight presacral vertebrae, in general with presence of ornamented spinal plates (lacking ornamentation in *B. alipioi*), with presence of ornamented paravertebral plates and with presacrals IV–V and VI–VII fused in *B. ephippium* [52,53]. Most species in the *B. pernix* group with their osteology described have eight presacral vertebrae, *viz B. izecksohni* and *B. brunneus* [60], *B. pombali* and *B. ferruginus* [61], *B. coloratus* and *B. curupira* [4], *B. albolineatus* [62], and *B. actaeus* [63]. Like *B. lulai* sp. nov., the VIII presacral vertebra is fused in *B. albolineatus*, and it is possible that *B. brunneus* and *B. pombali* also presents this fusion [60]. However, more specimens should be analyzed, since there may be variation in the number of vertebrae and malformations. Manuella Folly (personal communication to LF Ribeiro, 2022) reported that among 13 analyzed species of *B. pernix* group none showed this degree of fusion and all have eight distinct presacral vertebrae. In addition, she said also that some specimens showed malformations, which could be fusion or an extra fused vertebra.

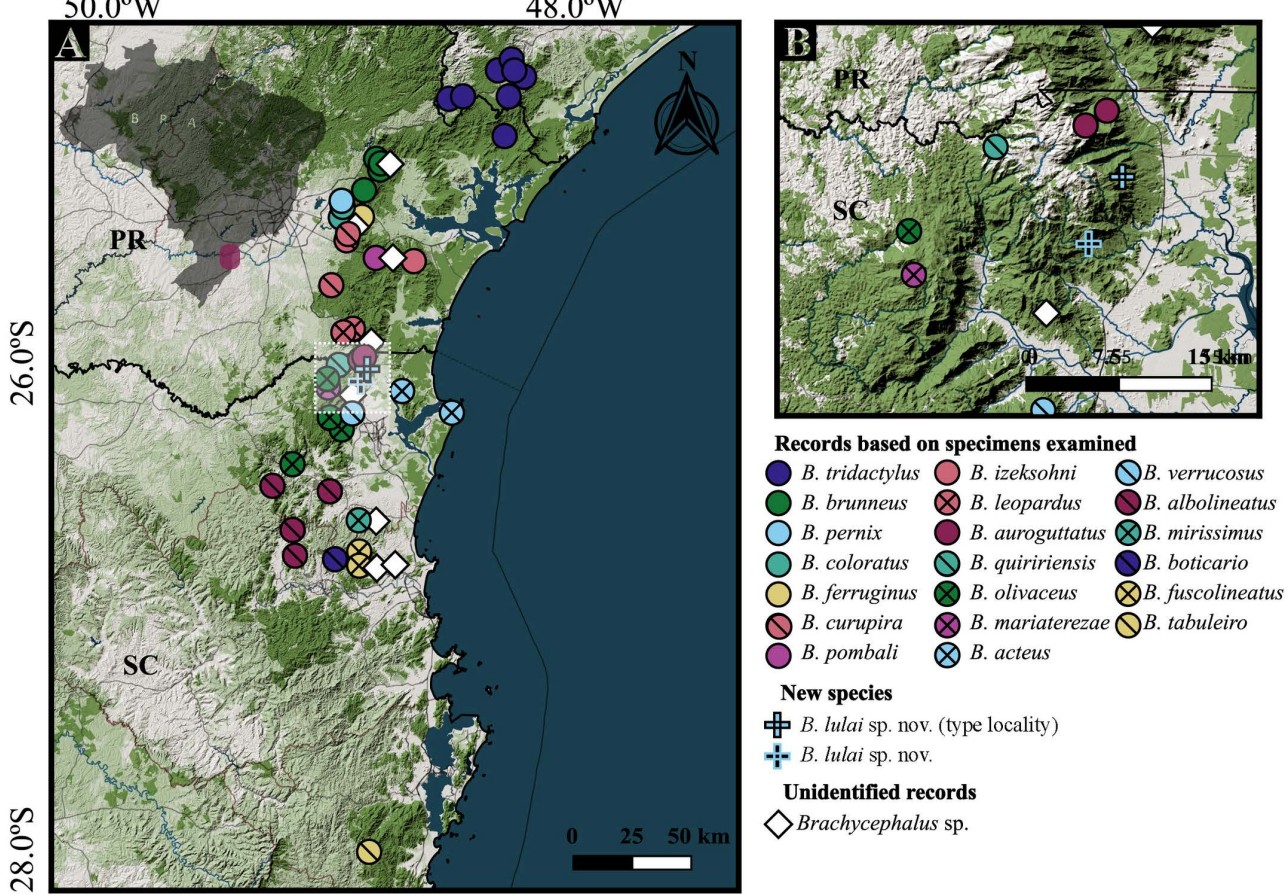

**Fig 12. Location of specimens examined of species from *Brachycephalus pernix* group, in southeastern and southern Brazil.** (A) All specimens. (B) Detail of the distribution of *B. lulai* sp. nov. Abbreviations: SP = São Paulo; PR = Paraná; SC = Santa Catarina. Urban areas are displayed in gray, open vegetation areas in light green, and dense vegetation in dark green. Basemaps: OpenStreetMap, under Open Database License (ODbL), available at https://www.openstreetmap.org/.

## Bioacoustics

In addition to *B. lulai* sp. nov., we describe aspects of the advertisement call of 16 other species (Table 4). These descriptions, all under a note-centered approach, reveal important aspects of the call structure that are overlooked in a call-centered approach [28], for example presence of note groups, attenuated notes, and warming notes (see below). These descriptions reinforce that calls increase in complexity along note emissions during a given call [26], with the incorporation of higher number of pulses and, in some cases, note groups during the call emission. In the call-centered approach (see below), these differences in call emissions would be perceived as variation in the number of notes per advertisement call, in the number of pulses per notes, and in the rate of call emissions [28]. This last parameter is exclusive of the advertisement call description under call-centered approach because the intervals between calls under the note-centered approach, as [26], represent a long time of silence of about 20–40 min [6,26,27] which, usually, is not measured.

The note-centered approach to advertisement call description in *Brachycephalus* [6, 26, 27, 28, this study] had not been adopted by other research groups working with the genus until recently [47,54]. However, in both studies, the authors treated the advertisement call as a phenomenon distinct from that considered in other works [6,26,27,28], this

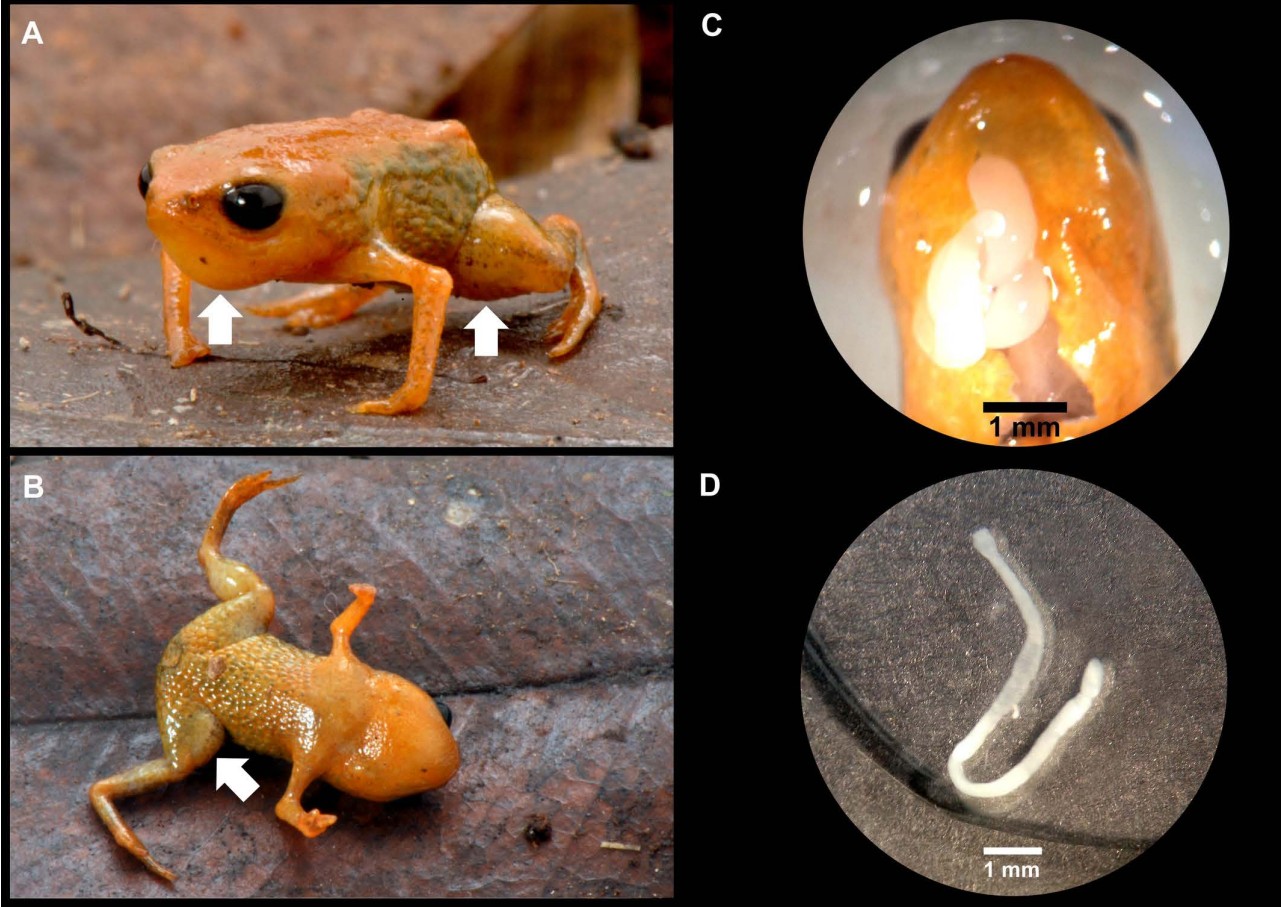

**Fig 13. *Brachycephalus lulai* sp. nov. (MHNCI 11593) parasitized by tapeworms (Ophiotaenia sp).** (A) The arrows point to the locations of the parasites at the gular region and the left thigh. (B) The arrow points to the location of the parasite on the left thigh. Edema regions can also be seen in these images, such as the left leg and right thigh. (C) *Ophiotaenia* sp. soon after removal from the gular region of the collected frog. (D) *Ophiotaenia* sp. isolated from the *B. lulai* sp. nov. Photographs by Luiz F. Ribeiro.

study]. Mângia *et al.* [47] interpret the call of *B. tabuleiro* as what other words have described as "note groups" (see Table 4), whereas Toledo *et al.* [54] interpret the call of *B. dacnis* as corresponding to what other works have described as "note groups" (the "complex" call) and "isolated notes" (the "simple" call). Mângia *et al.* [47] stated that they used the call-centered approach to describe the advertisement call of *B. tabuleiro*, but, considering two notes as its call they have worked, in fact, with the note-centered approach. In addition to this methodological issue (see below), the authors also used two different terms ("call duration" and "note duration") to refer to the same parameter, and missed the opportunity to clarify what the isolated notes of *B. tabuleiro* actually represent (see Table 4). Toledo *et al.* [54] adopted the note-centered approach in the description of *B. dacnis*, following Bornschein *et al.* [28], who demonstrated that it was impossible to distinguish the call of that, at the time undescribed, species from that of *B. hermogenesi* using the call-centered approach. These authors also treated "call duration" and "note duration" as equivalent parameters. However, by doing so, either "call duration" would represent a longer time span than "note duration," or "note duration" would necessarily include the interval between the two notes of the so-called "complex call" described by Toledo *et al.* [54].

**Table 7. Conservation status of species of the *Brachycephalus pernix* group, southeastern and southern Brazil.**

| Species | EOO (km²) | Upper bound AOO (km²) | Lower bound AOO (km²) | Mapped habitat (km²) | Altitudinal range records (m above sea level) | Altitudinal range mapping (m above sea level) | Number of locations | Conservation status |
|---|---|---|---|---|---|---|---|---|
| *B. actaeus* | 488.3 | 352 | 28 | 164.295 | 20–530 | 20–698 | 3 | EN B1ab(iii)+B2ab(iii) |
| *B. albolineatus* | 450.3 | 220 | 16 | 35.878 | 500–835 | 500–917 | 6 | VU B1ab(iii)+B2ab(iii) |
| *B. auroguttatus* | **4.0** | --- | 4 | --- | 385–1,100 | --- | 1 | LC |
| *B. boticario* | 4.0 | 4 | 4 | 0.501 | 680–795 | 680–832 | 1 | CR B1ab(iii)+2ab(iii) |
| *B. brunneus* | **148.0** | 148 | 24 | 59.325 | 1,074–1,770 | 1,074–1,830 | 2 | LC |
| *B. coloratus* | **12.0** | 12 | 8 | 1.475 | 1,145–1,250 | 1,145–1,260 | 3 | EN B2ab(iii) |
| *B. curupira* | 280.3 | 248 | 8 | 110.223 | 980–1,320 | 980–1,537 | 2 | LC |
| *B. ferruginus* | **100.0** | 100 | 4 | 60.405 | 965–1,537 | 965–1,537 | 1 | LC |
| *B. fuscolineatus* | **8.0** | 8 | 8 | 0.980 | 525–790 | 525–820 | 2 | EN B1ab(iii)+2ab(iii) |
| *B. izecksohni* | **16.0** | 16 | 4 | 3.781 | 980–1,340 | 980–1,454 | 1 | VU D2 |
| *B. leopardus* | **24.0** | 24 | 8 | 3.633 | 1,340–1,645 | 1,340–1,645 | 2 | VU D2 |
| *B. lulai* sp. nov. | **8.0** | --- | 8 | --- | 435–990 | --- | 1 | LC |
| *B. mari-aeterezae* | **4.0** | --- | 4 | --- | 1,265–1,270 | --- | 1 | CR B1ab(iii)+2ab(iii) |
| *B. mirissimus* | **8.0** | 8 | 4 | 0.725 | 470–540 | 470–560 | 1 | CR B1ab(iii)+2ab(iii) |
| *B. olivaceus* | 276.5 | --- | 16 | --- | 648–985 | | 3 | EN B1ab(iii)+2ab(iii) |
| *B. pernix* | **16** | 16 | 4 | 3.885 | 1,135–1,405 | 1,135–1,405 | 1 | VU D2 |
| *B. pombali* | **4** | --- | 4 | --- | 845–1,300 | --- | 1 | LC |
| *B. quiririensis* | **40.0** | 40 | 12 | 6.261 | 1,240–1,380 | 1,240–1,468 | 1 | CR B1ab(iii) |
| *B. tabuleiro* | **4.0** | --- | 4 | --- | 880–897 | --- | 1 | CR B1ab(iii)+2ab(iii) |
| *B. tridactylus* | 1,677 | 1,240 | 32 | 520.274 | 715–1,140 | 700–1,489 | 2 | EN B1ab(iii) |
| *B. verrucosus* | 4 | --- | 4 | --- | 455–945 | --- | 1 | LC |

Dashes in the upper bound of AOO indicate that we did not map the estimated occupied habitat for that species, either due to the absence of an associated altitudinal range or because mapping was unrealistic given the vegetation and altitude. EOO values shown in bold represent species for which we adjusted this metric to match the AOO, in order to ensure consistency in the definition.

For some species in the *B. pernix* group, the first notes emitted during an advertisement call, under a note-centered approach, are difficult to hear in both the field and in recordings, which is why we rarely record the first emissions. We named these weak starting notes of an advertisement call as "warming notes" [26], assuming that these reflect the individual's preparation process building up to the "typical" strongest notes. Nevertheless, for other species that we analyzed, we recorded some complete advertisement calls by individuals and for which the first notes were similar to others in structure and intensity. It is possible for individuals to begin calling with warming notes if they are "cold" and without them if they are "warmed up" by the previous emission of one or more calls.

Like warming notes, attenuated notes, also under a note-centered approach, could prepare the individual to emit the immediately subsequent notes with relatively higher sound energy/amplitude. Of those species that show attenuated

**Table 8. Green Status of species of the *Brachycephalus pernix* group, southeastern and southern Brazil.**

| Species | Green Scores (percentages) | | Recovery Potential (%) | Strategic Weight (points) | Potential Weight of a CU[1] (points) |
|---|---|---|---|---|---|
| | Current (Weight of each SU) | Long-term potential (Weight of each SU) | | | |
| *B. actaeus* | 46.7 ($10^2$, 1.5, 1.5) | 100.0 ($10^2$, $10^2$, $10^2$) | 53.3 | 68.3 | 68.3 |
| *B. albolineatus* | 15 (1.5, 1.5, 1.5, 1.5, 1.5, 1.5) | 100.0 ($10^2$, $10^2$, $10^2$, $10^2$, $10^2$, $10^2$) | 85.0 | 90.0 | 90.0 |
| *B. auroguttatus* | 100.0 ($10^2$) | 100.0 ($10^2$) | 0.0 | 1.0 | 107.5[3] |
| *B. boticario* | 15.0 (1.5) | 100.0 ($10^2$) | 85.0 | 105.0 | 105.0 |
| *B. brunneus* | 100.0 ($10^2$, $10^2$) | 100.0 ($10^2$; $10^2$) | 0.0 | 1.0 | ---[4] |
| *B. coloratus* | 43.3 (1.5, $10^2$, 1.5) | 100.0 ($10^2$, $10^2$, $10^2$) | 56.5 | 71.5 | ---[4] |
| *B. curupira* | 100.0 ($10^2$, $10^2$) | 100.0 ($10^2$, $10^2$) | 0.0 | 1.0 | ---[4] |
| *B. ferruginus* | 100.0 ($10^2$) | 100.0 ($10^2$) | 0.0 | 1.0 | ---[4] |
| *B. fuscolineatus* | 57.5 (1.5, $10^2$) | 100.0 ($10^2$, $10^2$) | 42.5 | 57.5 | 57.5 |
| *B. izecksohni* | 100.0 ($10^2$) | 100.0 ($10^2$) | 0.0 | 5.0 | ---[4] |
| *B. leopardus* | 15 (1.5, 1.5) | 100.0 ($10^2$) | 85 | 90 | 90 |
| *B. lulai sp. nov.* | 100.0 ($10^2$) | 100.0 ($10^2$) | 0.0 | 1.0 | 107.5[3] |
| *B. mariaeterezae* | 15.0 (1.5) | 15.0 (1.5) | 0.0 | 20.0 | 35.0[5] |
| *B. mirissimus* | 15.0 (1.5) | 15.0 (1.5) | 0.0 | 20.0 | 20.0 |
| *B. olivaceus* | 100.0 ($10^2$, $10^2$, $10^2$) | 100.0 ($10^2$, $10^2$, $10^2$) | 0.0 | 15.0 | 35.0[5] |
| *B. pernix* | 100.0 ($10^2$) | 100.0 ($10^2$) | 0.0 | 5.0 | ---[4] |
| *B. pombali* | 100.0 ($10^2$) | 100.0 ($10^2$) | 0.0 | 1.0 | ---[4] |
| *B. quiririensis* | 15.0 (1.5) | 100.0 ($10^2$) | 85.0 | 105.5 | 107.5[3] |
| *B. tabuleiro* | 15.0 (1.5) | 100.0 ($10^2$) | 85.0 | 105.5 | 105.5[6] |
| *B. tridactylus* | 62.5 (2.5, $10^2$) | 100.0 ($10^2$, $10^2$) | 37.5 | 52.5 | ---[4] |
| *B. verrucosus* | 100.0 ($10^2$) | 100.0 ($10^2$) | 0.0 | 1.0 | 1.0 |

Abbreviations: SU = spatial unit, CU = conservation unit.

[1]Sum of the strategic weight of the higher number of species that could be included in a CU that incorporated the least possible amount of human occupation.

[2]SU naturally small, but with ecological functions at baseline levels and without a specific threat (functional).

[3]Conservation unit to protect a group of species number 1 (proposed as Refúgio de Vida Silvestre Serra do Quiriri).

[4]Species already recorded inside CU.

[5]Conservation unit to protect a group of species number 2.

[6]Our assessment suggests that the records of *Brachycephalus tabuleiro* fall outside the limits of Parque Estadual da Serra do Tabuleiro (see Discussion).

notes, we demonstrate that not all notes are preceded by attenuated notes (see Table 4). Attenuated notes are perceived more clearly in spectrograms than in oscillograms. The quality of the recordings may influence the perception of these notes. "Typical" notes—when recorded far away from the emitter—have a spectral quality that does not allow for the detection of pulses and, in these conditions, attenuated notes might have been diluted by background noise (see [27]). The accuracy of the researchers can also influence the perception of the attenuated notes. For example, there are attenuated notes in the advertisement calls of two species, *B. olivaceus* and *B. quiririensis* (Table 4), for which previous descriptions of their calls did not mention this feature [64]. However, the oscillogram of Fig 2A of Monteiro *et al.* [64] does reveal

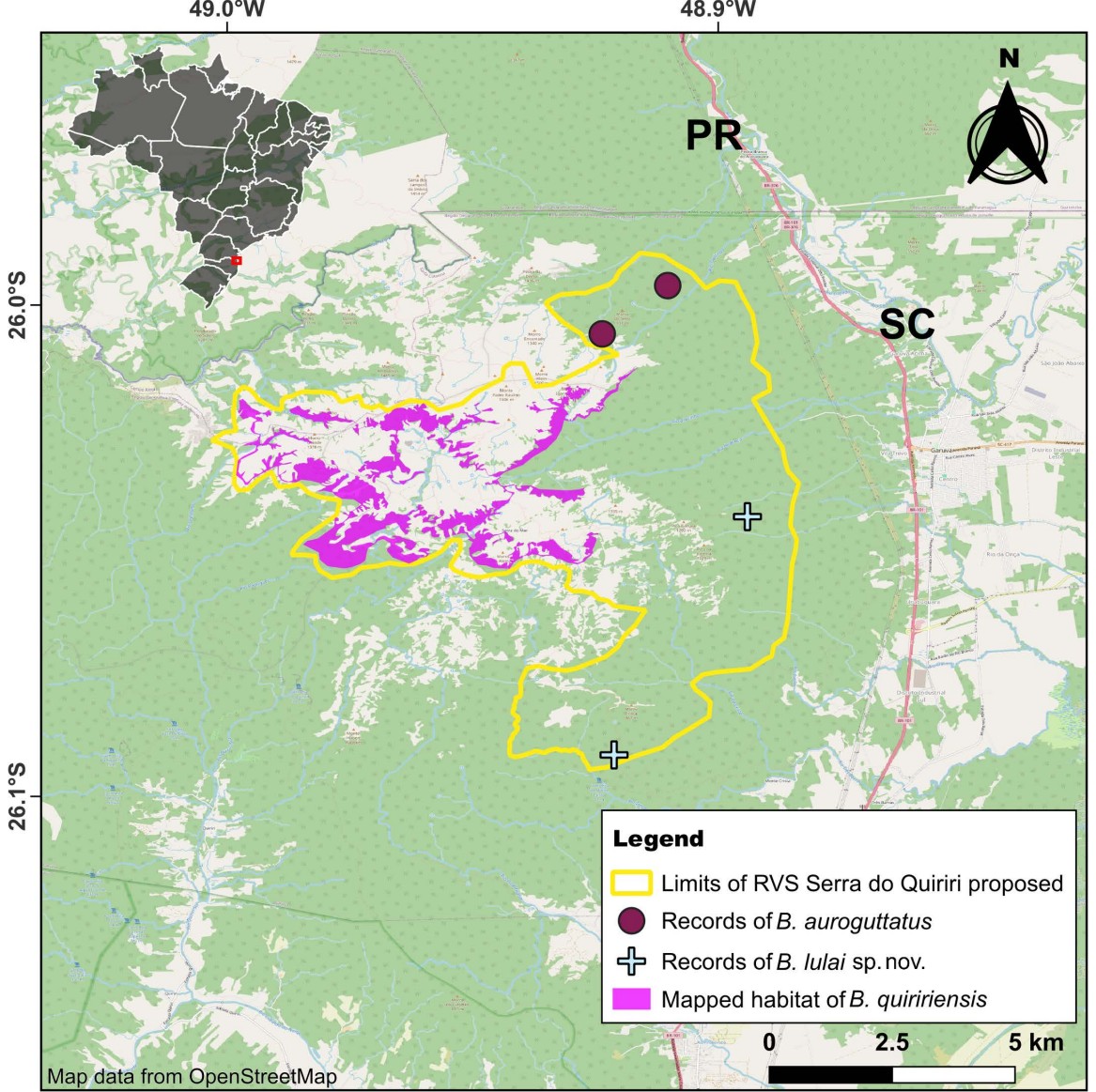

**Fig 14. Limits of the Refúgio de Vida Silvestre Serra do Quiriri, integral protection conservation unit proposed for *Brachycephalus lulai* sp. nov., *B. auroguttatus*, and *B. quiririensis*.** Abbreviations: PR = Paraná; SC = Santa Catarina; RVS = Refúgio de Vida Silvestre. Basemaps: Natural Earth, under Public Domain, available at https://www.naturalearthdata.com, and OpenStreetMap, under Open Database License (ODbL), available at https://www.openstreetmap.org/.

that attenuated notes were emitted in the advertisement call of *B. quiririensis*; prior to the notes shown in that figure, there is a discrete intensity of energy related to the attenuated notes.

Intraspecific variation in the level of complexity in advertisement call, under a note-centered approach, could reflect different functions to the different parts of the calls (with or without note groups), i.e., defense vs. mating [26]. However, later it was demonstrated that not all *Brachycephalus* advertisement calls present note groups [27]. In fact, we still know little about the daily, seasonal, and behavioral variation of *Brachycephalus* calls and their functions.

Finally, Goutte *et al.* [9] demonstrated that some species in the *B. ephippium* group are insensitive to their own advertisement call. However, Monteiro *et al.* [63] described responses by *B. actaeus* to playback experiments, perhaps perceiving call emissions through vibrations in other body receptors [28].

## Taxonomy

The first described species of the *B. pernix* group stood out for its unusual morphology (see [65]), including the absence of dermal co-ossification that is common to the *B. ephippium* group and a body shape that differs from the leptodactyliform shape common to the flea toads (see [13]). Considering all 21 species in the *B. pernix* group described to date, there are few diagnostic features distinguishing them—sometimes limited to coloration in life—though genetic evidence supports these as distinct lineages [33,59]. Relatively few additional characters contribute to distinguishing species within the *B. pernix* group, in part due to the scarcity of osteological and vocal descriptions. For example, the presence of the neopalatine ( [60]; see also [61]) and presence of note groups in the advertisement calls [6,26,27]. Only eight species of the *B. pernix* group had their advertisement call completely described so far [6, 26, 27, 47, 63, 64, this study] and only four were also diagnosed by call features [6, 47, 63, this study].

There are two additional difficulties in delimiting species of the *B. pernix* group: the procedures used in call descriptions and the use of color characters based on the literature alone. Two approaches are recognized for describing anuran calls: the call-centered approach and the note-centered approach [25]. *Brachycephalus actaeus* [63], *B. olivaceus* [64], *B. quiririensis* [64], and *B. tabuleiro* (see below; [47]) had their advertisement calls initially described in a call-centered approach (but see Table 4 for partial description of these species under note-centered approach), whereas *B. albolineatus* [26], *B. mirissimus* [6], *B. tridactylus* [27], *B. lulai* sp. nov. and all remaining species of the *B. pernix* group, except *B. mariaeterezae*, were described in a note-centered approach (this study). The primary difference between these approaches is that a set of notes issued in sequence is considered the advertisement call in the note-centered approach, whereas the call-centered approach treats a single note as the entire call [25]. Mângia *et al.* [47] introduced an unprecedented variation, likely not supported by the reference literature (see above). While based on Köhler *et al.*'s [25] call-centered approach, they suggest that the call of *B. tabuleiro* consists of two notes [47]. For a simple comparison, under the note-centered approach, as [26], the call of *B. tabuleiro* of Mângia *et al.* [47] with one note would be an example of an isolated note, and the call of this species composed of two notes would be an example of a note group (see Table 4). As demonstrated previously [28], the call-centered approach prevents the detection of some diagnostic call parameters because different parameters and substantial differences across them along the duration of a call are treated as simple variation of the call of the species. In conclusion, the use of the call-centered approach for species of the *B. pernix* group masks heterogeneity by treating distinct elements, perhaps with distinct functions, as mere variations of a single entity. We recommend that researchers employing the call-centered approach to describe *Brachycephalus* advertisement calls prioritize obtaining long-duration recordings of individuals, made at close range (< 2 m from the caller; see [27]). Moreover, we encourage the investigation of the potential biological significance of any heterogeneity observed in the calls.

As for the coloration in life, it has typically relied on qualitative interpretation of colors when descriptions were not made using a standardized color guide, as is the case of all described *Brachycephalus* species. This prevents robust comparisons in the descriptions of new species. However, the effect of personal interpretation can be minimized in the future literature if works limited the inclusion of live colors comparisons of the species that have been described by a single researcher.

The scarcity of information on features to distinguish species and the extent of intraspecific variation has led to debate about the validity of species in the *B. pernix* group [54,66,67]. No species, however, has been proposed as a synonym thus far, despite the accumulation of new specimens and expanding the knowledge on intraspecific variation. We emphasize that a species diagnosis can be revised through redescription, as was recently done for the flea toads *B. sulfuratus* in relation to *B. hermogenesi* [68]. A reassessment indicated that the diagnostic characters of these species varied across individuals, but also revealed other differences [28].

We have summarized the commonly used diagnostic traits and how these vary across the 21 species of the *B. pernix* group (Table 2). This is based on our observations of both living and preserved specimens in each species. All species can be distinguished by one or a combination of traits. There are also many similarities across these species, including in their advertisement calls and in coloration, such as the several orange or predominantly green species. Our summary shows that the green species such as *B. olivaceus* and *B. albolineatus* present distinct advertisement calls, the first with attenuated notes and *B. albolineatus* without them (Table 2). The advertisement calls also help to distinguish certain orange or brown species, such as *B. tridactylus* (see Table 2 and Fig 5D1–D3, E1–E3) without note groups and without attenuated notes, in comparison to the orange *B. lulai* sp. nov. that has both. The brown species *B. brunneus* and *B. curupira* can be distinguished by the maximum number of notes in a note group and maximum number of pulses per note, 3 and 4 *versus* 5 and 3, respectively, in addition to the iris having golden spots in *B. curupira* when alive (Table 2).

We have taken care to present the sample size in our summary of comparisons. Unfortunately, there are some species with less than 10 individuals collected, which limits our understanding of intraspecific variation in coloration and anatomical traits. Further sampling of *Brachycephalus* species is expected to significantly enhance our understanding of intraspecific variation, while also contributing to the revision of species boundaries. Both the lack of funds supporting field research and the difficulty of accessing some montane locations remain a problem to increasing sampling, which is sometimes achieved only after opening many kilometers of trails in dense forests [15].

## Biogeography

Geomorphological and sedimentological studies in Serra do Quiriri have shown that there were at least three semi-arid periods in the region, interspersed with wet periods. The most recent of these semi-arid periods dates from the beginning of the Pleistocene [69,70]. We hypothesize that the biogeography of *Brachycephalus* reflects a history in which their forested habitat occurred continuously at the base of the mountain complex in semi-arid periods and that these forests extended to higher altitudes during more humid periods [14,33,34]. For the *B. lulai* sp. nov. clade to diverge into three species that occur in Serra do Quiriri (Fig 10 and 12; see Fig 1 of Bornschein *et al.* [21]), these forests likely occurred as isolated patches (microrefugia) among the dominant grasslands, which could have led to speciation by vicariance. In contrast, vicariance among other species in the *B. pernix* group likely occurred through isolation on mountains separated by forested valleys.

The southern arboreal vegetations of the Atlantic Forest are richer in "forest types" than the arboreal vegetations of the regions in the northern portion of this domain [71], but little is known, and even modeled, about the relationship and interaction between fauna and vegetation during the hypothesized dispersals of vegetation over time. The expansion of arboreal vegetation over mountaintop grasslands is documented by palaeoecological studies (e.g., [72,73]) and still occurs today, potentially in all mountains where grasslands are still present. This expansion typically begins with the establishment of shrubs and small trees (M.R. Bornschein, pers. obs.). This early-stage "arboreal" vegetation is less than 1 m tall and is characterized by an herbaceous-shrubby stratum. It gradually develops in structure and increases in height, transitioning into more complex arboreal formations. In areas where grasslands persist, the advance of this arboreal stratum continues, leading to the formation of a mosaic of vegetation types in various stages of ecological succession. When shrubs and trees reach approximately 3–4 m in height, the vegetation becomes stratified into two strata: an herbaceous-shrub stratum and an arboreal stratum. From this point up to around 10 m in height, the vegetation structure typically includes three strata (herbaceous, shrub, and arboreal). Despite its relatively low stature and simplified vertical structure, this vegetation is classified as highland dense ombrophylous forest (*Floresta Ombrófila Densa Altomontana*), also referred to as cloud forest [74–81], based on an adapted version of the nomenclatural criteria proposed by Veloso *et al.* [23]. According to these criteria, true forests are expected to feature taller vegetation structured into four distinct strata ( [23]; see also [82]), a condition not met by these formations. Indeed, the recognition that highland dense ombrophylous forest is a structurally simplified vegetation type is well established. When composed of three strata, this environment is

treated as a transitional formation toward montane dense ombrophylous forest (*Floresta Ombrófila Densa Montana*; [79]), which typically occurs at lower elevations. In contrast, any arboreal formation exhibiting four strata in the study region, above 400 m a.s.l., is consistently classified as montane dense ombrophylous forest [79].

It is crucial to recognize that the colder-climate arboreal vegetation occupying mountaintops within the geographic range of *Brachycephalus* does not exhibit four strata. This indicates that such vegetation functions similarly to pioneer formations in ecological succession (*sensu* [23]) or to regenerating vegetation following disturbances such as logging or fire (see [83]). Pioneer and regenerating formations undergo structural changes over time [23], often evolving substantially within just a few decades [84,85], unlike mature forests, which tend to remain "static" in their structural configuration over many decades or even centuries.

The growth rate of highland dense ombrophylous forest following its initial colonization of mountaintop grasslands remains unknown. However, since 1988, M.R. Bornschein has documented the advance of this forest over grasslands in the Serra dos Órgãos, Paraná (not to be confused with the homonymous region in Rio de Janeiro), reaching higher elevations and subsequently being colonized by *B. brunneus* and an undescribed *Brachycephalus* species [2]. These observations demonstrate the upward expansion of the geographic range of species in the genus *Brachycephalus*, reaching elevations of up to 1,878 m a.s.l. These species were recorded in 2018 in areas where they had not previously been observed, occupying vegetation estimated to be less than 30 years old, based on surveys conducted in the 1980s, 1990s, and 2000s. In contrast, forest advancement has been interrupted in some mountains where grassland fires are frequent, such as in Serra do Araçatuba, Paraná ( [72]; M.R. Bornschein, pers. obs.), where *B. leopardus* is found [2].

We hypothesize that highland dense ombrophylous forest served as the initial environment that enabled the historical upward expansion of forest ecosystems, subsequently facilitating the dispersal of *Brachycephalus* species and, later, their allopatric speciation via vicariance [14,33,34]. Furthermore, our findings indicate that *Brachycephalus* species continue to disperse into recently formed environments, demonstrating a degree of ecological plasticity. At least *B. brunneus*, *B. leopardus*, *B. izecksohni*, and *Brachycephalus* sp. currently occupy 1-meter-tall highland dense ombrophylous forest (M.R. Bornschein, pers. obs.), while *B. leopardus* and *B. izecksohni* may also occur in grassland habitats [21].

The same speciation processes that likely acted within *Brachycephalus* probably also characterize other organisms with limited dispersal capacity (see [86]). In addition to the 21 species of the *B. pernix* group that are endemic to the mountains of southern Brazil, this region is also home to four endemic phytotelm-breeding anurans in the genus *Melanophryniscus* [87,88] and two small bird species related to *Scytalopus speluncae* that are awaiting description ( [89]; taxonomy according to Maurício *et al.* [90]).

## Conservation

The many recently described species of anurans endemic to the state of Santa Catarina [6,13,62,63,88,91] motivated technical meetings between the Environmental Institute of Santa Catarina (Instituto de Meio Ambiente – IMA), the Fundação Grupo Boticário de Proteção à Natureza, and researchers from the Mater Natura – Instituto de Estudos Ambientais. As a result, IMA instituted the program called "*Conservação dos anfíbios microendêmicos e de distribuição restrita de Santa Catarina*", which aims to support and develop research on the distribution and conservation status of these species, and to develop conservation strategies to them (Portaria IMA nº 283/2018). The present study is an effort aimed at all of the objectives of that program.

Our Green Status assessments follow the recommendation to perform this analysis for other endangered species of *Brachycephalus* [92]. This third Brazilian study evaluating Green Status again demonstrated the efficiency of this tool in assessing the species' current situation and the impacts of conservation strategies to be developed [85,86]. Additionally, the general assessment of the species made it possible to rank priorities and to define target species and key areas for conservation.

None of the 12 species of the *B. pernix* group from Santa Catarina state were recorded in Integral Protection Conservation Units (IPCUs; [2, this study]), a type of Brazilian conservation unit that necessitates the acquisition of land by the government. Although *B. tabuleiro* has been recorded as occurring in the IPCU Parque Estadual da Serra do Tabuleiro [47], our assessment suggests that the records are outside the limits of that park (M.R. Bornschein, pers. obs.). On the other hand, in the state of Paraná, eight out of nine species occur at least partially within this type of protected areas. In this state, most species are classified as Least Concern (N=4) or Vulnerable (N=3), mainly due to the absence of ongoing population declines and the lack of plausible future threats, contrasting with Santa Catarina, where most of the species are Critically Endangered (N=5). These findings highlight the critical role of IPCU areas in safeguarding the microendemic *Brachycephalus* species.

There is a particular type of IPCU established in Brazil law that does not imply expropriation, unless there are uses in the area incompatible with the protection of natural resources. This IPCU is the Wildlife Refuge (*Refúgio de Vida Silvestre*; see the National System of Conservation Units – SNUC, Law Nº 9.985/2000). The original objective of *Refúgio de Vida Silvestre* was the protection of environments (SNUC, Law Nº 9.985/2000), but is has also incorporated the objective to protect particular species of the fauna and fauna [93]. Thus, we propose the creation of an IPCU called Refúgio de Vida Silvestre Serra do Quiriri, with 6,600 ha located in Serra do Quiriri, municipalities of Garuva and Campo Alegre, Santa Catarina (Fig 14). Its objectives would be the protection of *B. lulai* sp. nov., but also *B. quiririiensis*, *B. auroguttatus*, and *Melanophryniscus biancae*, in addition to cloud forests and grasslands (*campos de altitude*). The limits of this conservation unit include all points of occurrence of *B. lulai* sp. nov. (this study) and *B. auroguttatus* as well as likely the entire potential habitat of *B. quiririiensis* [2]. To comply with these purposes, we delimited the conservation unit based on the altitude records of the three species. *Melanophryniscus biancae* is endemic to the highland grasslands of Serra do Quiriri and the adjacent Serra do Araçatuba, in Paraná state [19,88], and was recently recognized as endangered (EN) by Brazilian government (Portaria MMA Nº 148/2022) and also globally by IUCN [94].

The delimitation of the Refúgio de Vida Silvestre Serra do Quiriri encompasses a relatively small area with a low human population density. This refuge would have a greater strategic importance for conservation than any other unit that might be created for the other species of *Brachycephalus* in Santa Catarina (Table 8). Its creation and implementation would maintain the Green Score of *B. lulai* sp. nov. and *B. auroguttatus*, and, especially, change the status of *B. quiririiensis* from Critically Depleted to Fully Recovered and performing in its baseline potential.

Among the anthropic impacts in the Serra do Quiriri are the regular burning of grasslands (that also affect adjacent forests), cattle grazing, invasion of *Pinus* spp., and kaolin mining [88,95]. Additional impacts include erosion along roads (MR Bornschein, per. obs.) and touristic trails [96]. Part of the area proposed as a conservation unit contains many cattle that negatively impact cloud forests by creating deep holes formed along cattle paths and, secondarily, erosion by rainwater. Under these conditions, in 2004, we still recorded *B. quiririiensis* vocalizing in the vegetation between hollows, but we were not able to return to the area after 2010 and to confirm its permanence in the site (MR Bornschein, pers. obs.).

Possible management actions in the Refúgio de Vida Silvestre Serra do Quiriri could include a diagnosis of the impact of fire, management of grasslands with plots for controlled and rotational burning, establishment of a low-impact cattle stocking rate, reassessment of the impact and mitigation actions for kaolin mining, the cutting of *Pinus*, and the establishment of a plan to regulate tourism activities. Environmental compensation using governmental or municipal resources should be considered to reward landowners for the efforts to adapt to the management plan and to encourage alternatives to native wood used in the construction of barbed-wire fences. Due to its scenic beauty, the region attracts many tourists annually. Tourism can be expanded and provide funding to the region, also with a possible financial return to landowners.

## Conclusions

Over the past 15 years, our research group has concentrated efforts to sample the richness and geographical distribution of the species of the *B. pernix* group. In addition, our team also sought to better document the individual variation of all

*Brachycephalus* species in southern Brazil, looking for them in the field over the past seven years. We carried out a careful review of their original descriptions and their diagnostic characteristics based on newly collected specimens, including of some populations that remain to be properly identified. As a result of this work, we discovered and herein described a population collected on the eastern slope of Serra do Quiriri as a new species based on several diagnostic features.

The region of occurrence of the new species, the Serra do Quiriri, experienced semi-arid times in the past [69,70]. This implies an influence on the distribution of arboreal formations, corroborating the hypothesis of climatic variations causing altitudinal displacement of forest and promoting *Brachycephalus* speciation by vicariance at higher altitudes [14,33,34]. Because three sister species occur in the same mountain range of Serra do Quiriri, the displacement of forests to higher altitudes may have occurred with this vegetation isolated in natural patches (*capões*) likely leading to speciation as microrefugia.

Cloud forests may have served as the interface between past forest microrefugia and grasslands. Characteristics of the cloud forest that make it suitable to play this role include that (1) it can advance quickly over herbaceous plants, (2) it has a simplified structure, (3) it transitions to a lower altitude forest type, and (4) it is occupied by *Brachycephalus* even if the vegetation height is less than 2 m. These vegetational and faunal attributes shed light on the ecological implications of the altitudinal dispersion of biotas, the basis for the biogeographic hypothesis of vicariance speciation at high altitudes.

Although *Brachycephalus lulai* sp. nov. is currently classified as Least Concern, this status is based on the absence of observed ongoing decline and the apparent lack of plausible future threats. Nevertheless, it is essential to continue systematically monitoring this scenario. The new species is found close to other endemic and threatened anurans, justifying a proposition for the Refúgio de Vida Silvestre Serra do Quiriri, a specific type of Integral Protection Conservation Unit that would not necessitate expropriation of land by the government. This unit would help ensure both the maintenance and potential improvement of the conservation status of these species.

## Supporting information

**S1 Table. Primers used to amplify the loci used in the present study and their corresponding annealing temperatures and durations.**
(DOCX)

**S2 Table. GenBank accession numbers of all sequences used in our analyses, as well as the new sequences generated in the present study.**
(DOCX)

**S3 Table. Minimum and maximum pairwise genetic distance (%) matrix for 16S sequences among specimens of *Brachycephalus* in the present study.**
(DOCX)

**S1 Fig. Characterization of skin texture of *Brachycephalus*. A.** Smooth texture. B. Moderately rough texture. C. Densely rough texture. Scale bar equal 1 mm.
(TIF)

**S2 Fig. Distribution of *Brachycephalus actaeus*.** (A) Distribution considering the lower bound of area of occupancy (AOO) based on current records. (B) Distribution considering the upper bound of AOO incorporating suitable habitat. The black line represents the minimum convex polygon (MCP) of the extent of occurrence (EOO), pink polygons indicate mapped suitable habitat, red dots represent current records, and dark-shaded cells were accounted for in the estimation. All layers were created by the authors. Distribution was generated using field observations, habitat mapping, and altitude. No copyrighted or third-party material was used for the figure.
(TIF)

**S3 Fig.  Distribution of *Brachycephalus albolineatus*.** (A) Distribution considering the lower bound of area of occupancy (AOO) based on current records. (B) Distribution considering the upper bound of AOO incorporating suitable habitat. The black line represents the minimum convex polygon (MCP) of the extent of occurrence (EOO), pink polygons indicate mapped suitable habitat, red dots represent current records, and dark-shaded cells were accounted for in the estimation. All layers were created by the authors. Distribution was generated using field observations, habitat mapping, and altitude. No copyrighted or third-party material was used for the figure.
(TIF)

**S4 Fig.  Distribution of *Brachycephalus auroguttatus*.** Red dots represent current records, and the dark-shaded area was accounted for in the estimation. All layers were created by the authors. Distribution was generated using field observations, habitat mapping, and altitude. No copyrighted or third-party material was used for the figure.
(TIF)

**S5 Fig.  Distribution of *Brachycephalus boticario*.** The black line represents the minimum convex polygon (MCP) of the extent of occurrence (EOO), the pink polygon indicates mapped suitable habitat, the red dot represents the current record, and the dark-shaded area was accounted for in the estimation. All layers were created by the authors. Distribution was generated using field observations, habitat mapping, and altitude. No copyrighted or third-party material was used for the figure.
(TIF)

**S6 Fig.  Distribution of *Brachycephalus brunneus*.** (A) Distribution considering the lower bound of area of occupancy (AOO) based on current records. (B) Distribution considering the upper bound of AOO incorporating suitable habitat. The black line represents the minimum convex polygon (MCP) of the extent of occurrence (EOO), pink polygons indicate mapped suitable habitat, red dots represent current records, and dark-shaded cells were accounted for in the estimation. All layers were created by the authors. Distribution was generated using field observations, habitat mapping, and altitude. No copyrighted or third-party material was used for the figure.
(TIF)

**S7 Fig.  Distribution of *Brachycephalus coloratus*.** (A) Distribution considering the lower bound of area of occupancy (AOO) based on current records. (B) Distribution considering the upper bound of AOO incorporating suitable habitat. The black line represents the minimum convex polygon (MCP) of the extent of occurrence (EOO), pink polygons indicate mapped suitable habitat, red dots represent current records, and dark-shaded cells were accounted for in the estimation. All layers were created by the authors. Distribution was generated using field observations, habitat mapping, and altitude. No copyrighted or third-party material was used for the figure.
(TIF)

**S8 Fig.  Distribution of *Brachycephalus curupira*.** (A) Distribution considering the lower bound of area of occupancy (AOO) based on current records. (B) Distribution considering the upper bound of AOO incorporating suitable habitat. The black line represents the minimum convex polygon (MCP) of the extent of occurrence (EOO), pink polygons indicate mapped suitable habitat, red dots represent current records, and dark-shaded cells were accounted for in the estimation. All layers were created by the authors. Distribution was generated using field observations, habitat mapping, and altitude. No copyrighted or third-party material was used for the figure.
(TIF)

**S9 Fig.  Distribution of *Brachycephalus ferruginus*.** (A) Distribution considering the lower bound of area of occupancy (AOO) based on current records. (B) Distribution considering the upper bound of AOO incorporating suitable habitat. The black line represents the minimum convex polygon (MCP) of the extent of occurrence (EOO), pink polygons indicate

mapped suitable habitat, the red dot represents the current record, and dark-shaded cells were accounted for in the estimation. All layers were created by the authors. Distribution was generated using field observations, habitat mapping, and altitude. No copyrighted or third-party material was used for the figure.
(TIF)

**S10 Fig. Distribution of *Brachycephalus fuscolineatus*.** The black line represents the minimum convex polygon (MCP) of the extent of occurrence (EOO), pink polygons indicate mapped suitable habitat, red dots represent current records, and dark-shaded cells were accounted for in the estimation. All layers were created by the authors. Distribution was generated using field observations, habitat mapping, and altitude. No copyrighted or third-party material was used for the figure.
(TIF)

**S11 Fig. Distribution of *Brachycephalus izecksohni*.** (A) Distribution considering the lower bound of area of occupancy (AOO) based on current records. (B) Distribution considering the upper bound of AOO incorporating suitable habitat. The black line represents the minimum convex polygon (MCP) of the extent of occurrence (EOO), pink polygons indicate mapped suitable habitat, the red dot represents the current record, and dark-shaded cells were accounted for in the estimation. All layers were created by the authors. Distribution was generated using field observations, habitat mapping, and altitude. No copyrighted or third-party material was used for the figure.
(TIF)

**S12 Fig. Distribution of *Brachycephalus leopardus*.** (A) Distribution considering the lower bound of area of occupancy (AOO) based on current records. (B) Distribution considering the upper bound of AOO incorporating suitable habitat. The black line represents the minimum convex polygon (MCP) of the extent of occurrence (EOO), pink polygons indicate mapped suitable habitat, red dots represent current record, and dark-shaded cells were accounted for in the estimation. All layers were created by the authors. Distribution was generated using field observations, habitat mapping, and altitude. No copyrighted or third-party material was used for the figure.
(TIF)

**S13 Fig. Distribution of *Brachycephalus lulai* sp. nov.** Red dots represent current records, and dark-shaded cells were accounted for in the estimation. All layers were created by the authors. Distribution was generated using field observations, habitat mapping, and altitude. No copyrighted or third-party material was used for the figure.
(TIF)

**S14 Fig. Distribution of *Brachycephalus mariaeterezae*.** The red dot represents the current record, and the dark-shaded area was accounted for in the estimation. All layers were created by the authors. Distribution was generated using field observations, habitat mapping, and altitude. No copyrighted or third-party material was used for the figure.
(TIF)

**S15 Fig. Distribution of *Brachycephalus mirissimus*.** (A) Distribution considering the lower bound of area of occupancy (AOO) based on current records. (B) Distribution considering the upper bound of AOO incorporating suitable habitat. The black line represents the minimum convex polygon (MCP) of the extent of occurrence (EOO), pink polygons indicate mapped suitable habitat, red dots represent current records, and dark-shaded cells were accounted for in the estimation. All layers were created by the authors. Distribution was generated using field observations, habitat mapping, and altitude. No copyrighted or third-party material was used for the figure.
(TIF)

**S16 Fig. Distribution of *Brachycephalus olivaceus*.** The black line represents the minimum convex polygon (MCP) of the extent of occurrence (EOO), dots represent current records, and dark-shaded cells were accounted for in the

estimation. All layers were created by the authors. Distribution was generated using field observations, habitat mapping, and altitude. No copyrighted or third-party material was used for the figure.
(TIF)

**S17 Fig. Distribution of *Brachycephalus pernix*.** (A) Distribution considering the lower bound of area of occupancy (AOO) based on current records. (B) Distribution considering the upper bound of AOO incorporating suitable habitat. The black line represents the minimum convex polygon (MCP) of the extent of occurrence (EOO), pink polygons indicate mapped suitable habitat, the red dot represents the current record, and dark-shaded cells were accounted for in the estimation. All layers were created by the authors. Distribution was generated using field observations, habitat mapping, and altitude. No copyrighted or third-party material was used for the figure.
(TIF)

**S18 Fig Distribution of *Brachycephalus pombali*.** Red dots represent current records, and the dark-shaded area was accounted for in the estimation. All layers were created by the authors. Distribution was generated using field observations, habitat mapping, and altitude. No copyrighted or third-party material was used for the figure.
(TIF)

**S19 Fig. Distribution of *Brachycephalus quiririensis*.** (A) Distribution considering the lower bound of area of occupancy (AOO) based on current records. (B) Distribution considering the upper bound of AOO incorporating suitable habitat. The black line represents the minimum convex polygon (MCP) of the extent of occurrence (EOO), pink polygons indicate mapped suitable habitat, red dots represent current records, and dark-shaded cells were accounted for in the estimation. All layers were created by the authors. Distribution was generated using field observations, habitat mapping, and altitude. No copyrighted or third-party material was used for the figure.
(TIF)

**S20 Fig. Distribution of *Brachycephalus tabuleiro*.** The red dot represents the current record, and the dark-shaded area was accounted for in the estimation. All layers were created by the authors. Distribution was generated using field observations, habitat mapping, and altitude. No copyrighted or third-party material was used for the figure.
(TIF)

**S21 Fig. Distribution of *Brachycephalus tridactylus*.** (A) Distribution considering the lower bound of area of occupancy (AOO) based on current records. (B) Distribution considering the upper bound of AOO incorporating suitable habitat. The black line represents the minimum convex polygon (MCP) of the extent of occurrence (EOO), pink polygons indicate mapped suitable habitat, red dots represent current records, and dark-shaded cells were accounted for in the estimation. All layers were created by the authors. Distribution was generated using field observations, habitat mapping, and altitude. No copyrighted or third-party material was used for the figure.
(TIF)

**S22 Fig. Distribution of *Brachycephalus verrucosus*.** The red dot represents the current record, and the dark-shaded area was accounted for in the estimation. All layers were created by the authors. Distribution was generated using field observations, habitat mapping, and altitude. No copyrighted or third-party material was used for the figure.
(TIF)

**S1 Appendix. Examined specimens of *Brachycephalus*.** Abbreviations: CFBH = Célio F. B. Haddad collection, Departamento de Zoologia, Universidade Estadual Paulista, Campus de Rio Claro, São Paulo; DZUP = Coleção Herpetológica do Departamento de Zoologia, Universidade Federal do Paraná, Curitiba, Paraná; MHNCI = Museu de História Natural Capão da Imbuia, Curitiba, Paraná; MNRJ = Museu Nacional, Rio de Janeiro, Rio de Janeiro; MZUSP = Museu de

Zoologia da Universidade de São Paulo, São Paulo, São Paulo; and ZUEC = Museu de História Natural, Universidade Estadual de Campinas, Campinas, São Paulo.
(DOCX)

**S2 Appendix. Advertisement calls and territorial calls of *Brachycephalus* analyzed.** Sample size indicates number of analyzed calls and individuals (some recordings can contain more than one call). Abbreviation: MHNCI = Museu de História Natural Capão da Imbuia, Curitiba, Paraná; CASA = Coleção Audiovisual do Semiárido, Mossoró, Rio Grande do Norte; FNJV = Fonoteca Neotropical Jacques Vielliard, Campinas, São Paulo. Recordings at MHNCI were made by the authors with the following devices: digital recorder Sony PCM-D50 with a Sennheiser ME 66/K6 microphone, digital recorder Marantz PMD660 with a Sennheiser ME 66/K6 microphone, and/or digital recorder Tascam DR-44WL with a Sennheiser ME 67/K6 microphone, with sampling frequency rate of 44.1 kHz and 16-bit resolution.
(DOCX)

**S1 File. Limits of the Refúgio de Vida Silvestre Serra do Quiriri, integral protection conservation unit proposed for *Brachycephalus lulai* sp. nov., *B. auroguttatus*, and *B. quiririensis*.**
(SHP)

## Acknowledgments

Julio Cesar Moura Leite provided access to the collection at MHNCI. Milena Wachlevski provided access to the recordings of *B. tabuleiro* at CASA. Liliane Pires, André Luiz Ferreira Silva, Philippe Fumaneri Teixeira, and Tainara Thais Jory assisted during fieldwork. John Measey, Pedro Ivo Simões, Ulrich Sinsch and an anonymous reviewer revised the text and improved its quality.

## Author contributions

**Conceptualization:** Marcos R. Bornschein, Marcio R. Pie, Luiz F. Ribeiro.

**Formal analysis:** Marcos R. Bornschein, Marcio R. Pie, Júnior Nadaline, André E. Confetti, David C. Blackburn, Edward L. Stanley, Renata de Britto Mari, Gabriel Silveira Alves, Giovanna Sandretti-Silva, Felipe Farias de Andrade Lima, Luiz F. Ribeiro.

**Funding acquisition:** Marcos R. Bornschein, Marcio R. Pie, Júnior Nadaline, David C. Blackburn, Edward L. Stanley, Renata de Britto Mari, Luiz F. Ribeiro.

**Visualization:** Marcos R. Bornschein, Marcio R. Pie, Júnior Nadaline, André E. Confetti, David C. Blackburn, Edward L. Stanley, Renata de Britto Mari, Gabriel Silveira Alves, Giovanna Sandretti-Silva, Felipe Farias de Andrade Lima, Luiz F. Ribeiro.

**Writing – original draft:** Marcos R. Bornschein, Marcio R. Pie, André E. Confetti, David C. Blackburn, Edward L. Stanley, Renata de Britto Mari, Giovanna Sandretti-Silva, Luiz F. Ribeiro.

**Writing – review & editing:** Marcos R. Bornschein, Marcio R. Pie, Júnior Nadaline, André E. Confetti, David C. Blackburn, Edward L. Stanley, Renata de Britto Mari, Gabriel Silveira Alves, Giovanna Sandretti-Silva, Felipe Farias de Andrade Lima, Luiz F. Ribeiro.

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
