## [Decision Letter · Decision Letter 0]

28 Mar 2025

PONE-D-24-43253A new species of *Brachycephalus* (Anura: Brachycephalidae) from Serra do Quiriri, northeastern Santa Catarina state, southern Brazil, with a review of the diagnosis among species of the B. pernix group and conservation measuresPLOS ONE

Dear Dr. Pie,

Thank you for submitting your manuscript to PLOS ONE. After careful consideration, we feel that it has merit but does not fully meet PLOS ONE’s publication criteria as it currently stands. Therefore, we invite you to submit a revised version of the manuscript that addresses the points raised during the review process. A reviewer and academic editor have reviewed your manuscript. The comments provided by both are at the bottom of the email and in the attached document. While revising the manuscript make sure to provide point wise explanation to the steps taken to address the queries.

We look forward to receiving your revised manuscript.

Kind regards,

Neelesh Dahanukar, Ph.D.

Academic Editor

PLOS ONE

Journal Requirements:

“The field work was funded by Fundação Grupo Boticário de Proteção à Natureza (through grant 1149_20191) through project conducted by Mater Natura – Instituto de Estudos Ambientais. MRP received a grant from CNPq/MCT (301636/2016-8). GS-S received  grant from São Paulo Research Foundation (FAPESP; processes #2022/04847-7 and # 2023/09718-3). There was no additional external funding received for this study.”

4. We note that Figure 11 and 12 in your submission contain map/satellite images which may be copyrighted. All PLOS content is published under the Creative Commons Attribution License (CC BY 4.0), which means that the manuscript, images, and Supporting Information files will be freely available online, and any third party is permitted to access, download, copy, distribute, and use these materials in any way, even commercially, with proper attribution. For these reasons, we cannot publish previously copyrighted maps or satellite images created using proprietary data, such as Google software (Google Maps, Street View, and Earth). For more information, see our copyright guidelines: http://journals.plos.org/plosone/s/licenses-and-copyright.

 a. You may seek permission from the original copyright holder of Figure 11 and 12 to publish the content specifically under the CC BY 4.0 license. 

5. We are unable to open your Supporting Information file S1_File.nex and doc.kml. Please kindly revise as necessary and re-upload.

6. Please take this opportunity to be sure you have met all of our guidelines for new species. For proper registration of a new zoological taxon, we require two specific statements to be included in your manuscript.

1.        In the Results section, the globally unique identifier (GUID), currently in the form of a Life Science Identifier (LSID), should be listed under the new species name, for example:

 Anochetus boltoni Fisher sp. nov. urn:lsid:zoobank.org:act:B6C072CF-1CA6-40C7-8396-534E91EF7FBB

Another LSID for the manuscript itself should also appear within the Nomenclature statement. You will need to contact Zoobank (zoobank.org/About) to obtain a GUID (LSID). You should receive one LSID for your manuscript and a separate, unique LSID for the new species.

2.        Please also insert the following text into the Methods section, in a sub-section to be called "Nomenclatural Acts":

 The electronic edition of this article conforms to the requirements of the amended International Code of Zoological Nomenclature, and hence the new names contained herein are available under that Code from the electronic edition of this article. This published work and the nomenclatural acts it contains have been registered in ZooBank, the online registration system for the ICZN. The ZooBank LSIDs (Life Science Identifiers) can be resolved and the associated information viewed through any standard web browser by appending the LSID to the prefix "http://zoobank.org/". The LSID for this publication is: urn:lsid:zoobank.org:pub: XXXXXXX. The electronic edition of this work was published in a journal with an ISSN, and has been archived and is available from the following digital repositories: PubMed Central, LOCKSS [author to insert any additional repositories].

 All PLOS ONE articles are deposited in PubMed Central and LOCKSS. If your institute, or those of your co-authors, has its own repository, we recommend that you also deposit the published online article there and include the name in your article.

Following a recent ruling by the International Commission on Zoological Nomenclature, electronic journals are now a valid format for publication of new zoological taxa. In order to ensure the valid publication of your new species, please be sure to include the updated version of Nomenclatural Acts (above). A complete explanation of our guidelines for publishing new species can be found on our website: http://www.plosone.org/static/guidelines#zoological.

Additional Editor Comments:

There are several issues with the manuscript that authors will have to address before the manuscript can be considered further for review.

(1) Defining the *Brachycephalus* pernix group: Authors mention the B. pernix group in the introduction without properly defining it and providing information about the number of species under this group. The references used for defining the group are old and several species have been published ever since. So there is a need to provide the number of species currently present in the group and also list them for easy reference to the readers. Based on Table 2, it seems that there are 20 species currently know in this species group. If this is the case, why does the number of species differ substantially for different types of analysis and comparison. For example, the genetic analysis is based on only six species, while the conservation analysis is for 12 species. Authors need to keep some consistency in the arguments and explain why other species are not considered for the given analysis.

(2) Phylogenetic analysis: Phylogenetic analysis is not presented properly. Authors are not allowed to pick and choose some species for phylogenetic analysis. Molecular data are available for several species of B. pernix group and authors should use all available species in the group for analysis, even if some of the genetic markers are missing. In fact, it is advisable that authors use all member so the genus for larger phylogeny to show the monophyly of the group and then the taxonomic position of the new species. The genetic analysis is also not presented properly. The outgroups are not defined. There are large branches within the clades and there are politomies, suggesting that the trees are not resolved. Authors should increase the taxonomic sampling and perform maximum likelihood analysis in addition to Bayesian analysis. The genetic distances provided by the authors make very little sense. If there are multiple individuals then there will be a range of genetic distance and not just one value. It is essential to provide this range (minimum-maximum) for the intra and inter species genetic distance. I will refrain from providing the mean as the number of individuals are low and even if means are provided they should be accompanied with the standard deviation or some measure of uncertainty. The long branches in the proposed species are worrisome, and it is essential that authors prove that the new species is genetically distinct with species delimitation methods such as barcode gap (using ASAP) and Poisson tree process.

(3) Conservation status: This is the weakest part of the manuscript. Authors are using methodology that they themselves have devised for defining IUCN categories and they are not referring to proper IUCN guidelines for defining the categories based on the various criteria that are defined by the IUCN Redlist of Threatened Taxa documentation. If authors are using their own methodologies, there is no point in using the IUCN categories for defining the conservation status. The IUCN guidelines provides proper rationale for when to use different categories based on five different criteria used for assessments. Author's argument that the species is DD as a polygon cannot be constructed is flawed. The point localities are enough to define the Area of Occupancy based on 2km grids and have been used for defining the conservation status where the species are point endemics. Either authors should understand how the various criteria are used for finding the IUCN categories for red listing or authors should not used the categories used by the IUCN. In the current case if the threats to the species are know and the distribution is known, the species is not DD.

Reviewers' comments:

Reviewer's Responses to Questions

**Comments to the Author**

1. Is the manuscript technically sound, and do the data support the conclusions?

Reviewer #1: Partly

2. Has the statistical analysis been performed appropriately and rigorously? 

Reviewer #1: Yes

3. Have the authors made all data underlying the findings in their manuscript fully available?

Reviewer #1: Yes

4. Is the manuscript presented in an intelligible fashion and written in standard English?

Reviewer #1: No

5. Review Comments to the Author

Reviewer #1: This manuscript proposes a new species of the anuran genus *Brachycephalus* . The authors present several lines of evidence, including external morphology, osteology, vocalizations and molecular data to support the status of this taxon as a new species. They also acknowledge the difficulty in differentiating many species of the B. pernix group due to their morphological similarity and/or shallow genetic distances and highlight the importance of using as many types of data as possible in species descriptions.

However, I think the manuscript has a number of problems that should be addressed before it can be considered for publication, including minor issues such as wording, grammar, sentence structure, etc.

The complete review and comments on the ms are uploaded here as a file, due to its size.

6. PLOS authors have the option to publish the peer review history of their article (what does this mean? ). If published, this will include your full peer review and any attached files.

**Do you want your identity to be public for this peer review?** For information about this choice, including consent withdrawal, please see our Privacy Policy .

Reviewer #1: No

---

## [Author Response · Author response to Decision Letter 1]

21 Jul 2025

Reply to Journal Requirements

1) Please ensure that your manuscript meets PLOS ONE's style requirements, including those for file naming. The PLOS ONE style templates can be found at:

Authors: Done.

2) Thank you for stating the following financial disclosure:

“The field work was funded by Fundação Grupo Boticário de Proteção à Natureza (through grant 1149_20191) through project conducted by Mater Natura – Instituto de Estudos Ambientais. MRP received a grant from CNPq/MCT (301636/2016-8). GS-S received grant from São Paulo Research Foundation (FAPESP; processes #2022/04847-7 and # 2023/09718-3). There was no additional external funding received for this study.”

Please state what role the funders took in the study. If the funders had no role, please state: “The funders had no role in study design, data collection and analysis, decision to publish, or preparation of the manuscript.”

Authors: Done (see above in our response).

3) We note that you have indicated that there are restrictions to data sharing for this study. PLOS only allows data to be available upon request if there are legal or ethical restrictions on sharing data publicly. For more information on unacceptable data access restrictions, please see http://journals.plos.org/plosone/s/data-availability#loc-unacceptable-data-access-restrictions.

b) If there are no restrictions, please upload the minimal anonymized data set necessary to replicate your study findings to a stable, public repository and provide us with the relevant URLs, DOIs, or accession numbers. For a list of recommended repositories, please see https://journals.plos.org/plosone/s/recommended-repositories. You also have the option of uploading the data as Supporting Information files, but we would recommend depositing data directly to a data repository if possible.

Authors: There may have been an error on our part if we indicated that there are restrictions on the data. The examined specimens and vocalizations are housed in public collections, and the list of materials, along with their catalog numbers, is provided in the manuscript as appendices. Osteological data are also deposited in an open-access repository, with the access link provided in the manuscript. Genetic data are available in GenBank, and the accession numbers have been included in the manuscript. The sequences were included as supplementary material.

4) We note that Figure 11 and 12 in your submission contain map/satellite images which may be copyrighted. All PLOS content is published under the Creative Commons Attribution License (CC BY 4.0), which means that the manuscript, images, and Supporting Information files will be freely available online, and any third party is permitted to access, download, copy, distribute, and use these materials in any way, even commercially, with proper attribution. For these reasons, we cannot publish previously copyrighted maps or satellite images created using proprietary data, such as Google software (Google Maps, Street View, and Earth). For more information, see our copyright guidelines: http://journals.plos.org/plosone/s/licenses-and-copyright.

a. You may seek permission from the original copyright holder of Figure 11 and 12 to publish the content specifically under the CC BY 4.0 license.

Please upload the completed Content Permission Form or other proof of granted permissions as an “Other” file with your submission.

Authors: Done. The updated map now uses an OpenStreetMap basemap (ODbL-licensed), following the author guidelines

5) We are unable to open your Supporting Information file S1_File.nex and doc.kml. Please kindly revise as necessary and re-upload.

Authors: Done.

6) Please take this opportunity to be sure you have met all of our guidelines for new species. For proper registration of a new zoological taxon, we require two specific statements to be included in your manuscript.

6.1. In the Results section, the globally unique identifier (GUID), currently in the form of a Life Science Identifier (LSID), should be listed under the new species name, for example:

Anochetus boltoni Fisher sp. nov. urn:lsid:zoobank.org:act:B6C072CF-1CA6-40C7-8396-534E91EF7FBB. Another LSID for the manuscript itself should also appear within the Nomenclature statement. You will need to contact Zoobank (zoobank.org/About) to obtain a GUID (LSID). You should receive one LSID for your manuscript and a separate, unique LSID for the new species.

6.2. Please also insert the following text into the Methods section, in a sub-section to be called “Nomenclatural Acts”:

The electronic edition of this article conforms to the requirements of the amended International Code of Zoological Nomenclature, and hence the new names contained herein are available under that Code from the electronic edition of this article. This published work and the nomenclatural acts it contains have been registered in ZooBank, the online registration system for the ICZN. The ZooBank LSIDs (Life Science Identifiers) can be resolved and the associated information viewed through any standard web browser by appending the LSID to the prefix “http://zoobank.org/”. The LSID for this publication is: urn:lsid:zoobank.org:pub: XXXXXXX. The electronic edition of this work was published in a journal with an ISSN, and has been archived and is available from the following digital repositories: PubMed Central, LOCKSS [author to insert any additional repositories].

All PLOS ONE articles are deposited in PubMed Central and LOCKSS. If your institute, or those of your co-authors, has its own repository, we recommend that you also deposit the published online article there and include the name in your article.

Following a recent ruling by the International Commission on Zoological Nomenclature, electronic journals are now a valid format for publication of new zoological taxa. In order to ensure the valid publication of your new species, please be sure to include the updated version of Nomenclatural Acts (above). A complete explanation of our guidelines for publishing new species can be found on our website: http://www.plosone.org/static/guidelines#zoological.

Authors: Done.

Reply to Editor’s comments

7.1) Defining the *Brachycephalus* pernix group: Authors mention the B. pernix group in the introduction without properly defining it and providing information about the number of species under this group. The references used for defining the group are old and several species have been published ever since. So there is a need to provide the number of species currently present in the group and also list them for easy reference to the readers.

Authors: We defined the characteristics of the three groups in the second paragraph on the Introduction. Indeed, we had not mentioned the number of species in each group in the Introduction, but we did so in the Diagnosis section, with the appropriate references. We have now included the number of species in each group in the Introduction. However, we believe that listing the species belonging to each group would require bringing into the Introduction all the specific comments from the Diagnosis section regarding which author assigned each species described after 2015 to which group. This would disrupt the flow of the Introduction with highly technical content. In any case, we remain open to making this adjustment if the Editor still considers it beneficial for the readers.

7.2) Based on Table 2, it seems that there are 20 species currently know in this species group [B. pernix group]. If this is the case, why does the number of species differ substantially for different types of analysis and comparison. For example, the genetic analysis is based on only six species, while the conservation analysis is for 12 species. Authors need to keep some consistency in the arguments and explain why other species are not considered for the given analysis.

Authors: Regarding the number of species included in the genetic analysis, please see our response under Question #8.3. As for the species used in the conservation analysis, we have included the 12 species of the B. pernix group that occur in the state of Santa Catarina, since we are proposing the creation of a conservation unit in this state. We have now analyzed all 20 species of the B. pernix group.

8.1) Phylogenetic analysis: Phylogenetic analysis is not presented properly.

Authors: We thank the editor for the thoughtful suggestions on our molecular work. We now provide a more appropriate account of our goals and analyses. However, before we start, we want to emphasize that the validity of our new species, according to the ICZN, does not require any genetic data. We include it in our study to provide as much information as we can at this point about the new species.

8.2) Authors are not allowed to pick and choose some species for phylogenetic analysis.

Authors: We thank this opportunity to clarify this point. As will become clear below, our choice of species to be included in our analysis was far from arbitrary. However, we agree that the rationale for that choice was not explicit in our manuscript, and we added text to inform the reader about it.

8.3) Molecular data are available for several species of B. pernix group and authors should use all available species in the group for analysis, even if some of the genetic markers are missing. In fact, it is advisable that authors use all member so the genus for larger phylogeny to show the monophyly of the group and then the taxonomic position of the new species.

Authors: We understand this concern. However, including all of the genetic information for the group would not solve this issue, as we explain below.

The first comprehensive phylogeny of the B. pernix species group was provided by our research group in 2016 (Firkowski et al. 2016) and included only four loci through Sanger sequencing (i.e. 16S, cytb, ND2, and RPL). This was followed up by a phylogenomic study using UCEs and thousands of loci (Pie et al. 2018). There is only one locus in common (16S) between the current study and Firkowski et al. (2016), and there is no UCE data for the new species. Therefore, there would not be any shared information to place the new species beyond 16S. In addition, given the vast difference in the sizes of the datasets, in practice that would be tantamount to imposing topological constraints of the tree obtained with the phylogenomic dataset, which had very high support. In other words, it would be impossible for a single locus to overcome the influence of hundreds of UCE loci, and we would end up with the same tree as the UCE tree, except that the new species would be placed only with 16S. We believe that a more reasonable approach would be to use the phylogenomic analysis to inform our choice of taxa and rooting, as now made more explicit in the text, as follows. In the methods section, we wrote:

“To determine the phylogenetic position of the new species, we analyzed DNA sequences from four paratypes (MHNCI 11596, MHNCI 11598–11600). The choice of species to be analyzed alongside the new species was based on previous work on the phylogeny of the group (i.e. Firkowski et al. 2016, Pie et al. 2018). Preliminary analyses indicated that the new species was closely related to a well-supported clade that included B. auroguttatus, B. quiririensis, B. ferruginus, B. pernix, and B. pombali. The monophyly of this clade was first suggested by Firkowski et al. (2016) using Sanger sequencing and was later strongly supported by a phylogenomic analysis using UCEs by Pie et al. (2018). We therefore chose those species for exploring the phylogenetic position of the new species. The resulting tree was rooted according to the phylogenomic work by Pie et al. (2018).”

In the results section, we rewrote the following:

Phylogenetic relationships

The inferred phylogenetic relationships and phylogenetic distances between B. lulai and other species of the B. pernix species group are shown Fig 9 and Table 6, respectively. All specimens of B. lulai form a clade with strong support, which itself forms a larger clade together with B. auroguttatus and B. quiririensis, also with strong support…

8.4) The genetic analysis is also not presented properly.

Authors: We now provide more information (see above).

8.5) The outgroups are not defined.

Authors: As indicated above, the tree was rooted according to the relationships found in the phylogenomic study by Pie et al. (2018).

8.6) There are large branches within the clades and there are politomies, suggesting that the trees are not resolved.

Authors: We are not sure what is meant by large branches. If phylogenies are generated with only a few Sanger loci, such a level of unevenness in distances from the root to the tips is very common (we c

---

## [Decision Letter · Decision Letter 1]

6 Aug 2025

PONE-D-24-43253R1A new species of *Brachycephalus* (Anura: Brachycephalidae) from Serra do Quiriri, northeastern Santa Catarina state, southern Brazil, with a review of the diagnosis among species of the B. pernix group and conservation measuresPLOS ONE

Dear Dr. Pie,

Thank you for submitting your manuscript to PLOS ONE. After careful consideration, we feel that it has merit but does not fully meet PLOS ONE’s publication criteria as it currently stands. Therefore, we invite you to submit a revised version of the manuscript that addresses the points raised during the review process. In general the comments raised on the earlier draft by both the reviewer and the academic editor are addressed by the authors. However, the reviewer has provided some more comments, including reiteration of some of the comments made on the earlier draft. It is advisable that authors take all the comments made on the earlier and the current draft seriously and make appropriate changes to the manuscript. Authors should also provide a point to point rebuttal to the concerns raised by the reviewer. 

We look forward to receiving your revised manuscript.

Kind regards,

Neelesh Dahanukar, Ph.D.

Academic Editor

PLOS ONE

Journal Requirements:

Additional Editor Comments:

Authors have revised the text substantially with respect to the comments provided by the reviewer and the academic editor. However, the reviewer has provided some more comments, including reiterating earlier comments, and it is advisable that authors should take these comments seriously and provide proper justification to the issues raised and include the justification even int he main text of the article.

Reviewers' comments:

Reviewer's Responses to Questions

**Comments to the Author**

1. If the authors have adequately addressed your comments raised in a previous round of review and you feel that this manuscript is now acceptable for publication, you may indicate that here to bypass the “Comments to the Author” section, enter your conflict of interest statement in the “Confidential to Editor” section, and submit your "Accept" recommendation.

Reviewer #1: (No Response)

2. Is the manuscript technically sound, and do the data support the conclusions?

Reviewer #1: Yes

3. Has the statistical analysis been performed appropriately and rigorously? 

Reviewer #1: Yes

4. Have the authors made all data underlying the findings in their manuscript fully available?

Reviewer #1: Yes

5. Is the manuscript presented in an intelligible fashion and written in standard English?

Reviewer #1: Yes

6. Review Comments to the Author

Reviewer #1: Major issues:

As stated in my first review of this ms, I still discourage the continuing use of the B. didactylus group as a taxonomic entity. The authors have responded to this critic by arguing that the B. didactylus group is “a very well-defined group of species based on morphological characteristics”. Species groups are treated in current zoological systematics as monophyletic groupings of species rather than purely phenetic ones (unless when the phylogenetic relationships among species are not known). And it is now widely known that the characteristics uniting the species in the B. didactylus group (including B. clarissae, which does not share all of them) are plesiomorphic for the genus. Yet, even after addressing the non-monophyly of the B. didactylus group in the introduction, the authors continue to refer to it throughout the text as if it was a taxonomic entity equivalent to the other two species groups in the genus (for which there is strong evidence of monophyly in the literature). I consider this misleading, and continue to advocate for the use of the term “flea toads” instead, which unlike the “species group” category, corresponds to a morphotype rather than a clade. Indeed, the term is adopted in the work of Toledo et al. (2024), which included B. clarissae among the species with that morphotype (as mentioned by the authors in their response to my previous review).

The authors also seem to use the B. ephippium group in the sense of Ribeiro et al. 2015 (i.e. including both the ephippium and vertebralis lineages proposed by Condez et al. 2020) rather than in the sense of Folly et al. 2022 (i.e. excluding the vertebralis lineage, which is considered as a separate species group). The latter definition seems more conservative, since not all phylogenetic analyses published so far presented strong support for the monophyly of the B. ephippium group sensu Ribeiro et al. 2015. Anyway, since there are currently two different definitions of this group being used in the literature, the authors should explicitly state in the text which one they are using.

I believe that in all tables, as well as in table and figure legends that feature the name “B. lulai”, it should be followed by “sp. nov.”

Minor corrections:

Line 4 – I suggest adding the word “proposed” before “conservation measures”.

Line 26 – Exclude the hyphen in “currently-recognized”.

Line 29 – Change “call comparisons” to “acoustic comparisons”.

Line 52 – “The species are diurnal”. This should be changed to “…mostly diurnal”, since some of the cryptically colored “flea toads” often present nocturnal activity.

Line 98 – Change “also inhabited de occurrence of” to “also inhabited by”.

Line 102 – Change “occurs B. leopardus, a species known from two localities” to “B. leopardus occurs, being known from two localities”.

Lines 106-107 – Change “annual mean of temperature” to “mean annual temperature”.

Line 306 – Change “we did not mapped” to “we have not mapped”.

Line 468 – “discrete irregular greenish or brown spots”. Do you mean “discrete” or “discreet”? These are different terms.

Line 479 – Add the word “species” after “Sexually dimorphic”.

Line 483 – Change “green spots” to “green patches”.

Line 486 – Change “becoming dark, brownish black” to “becoming dark gray”.

Line 549 – “in B and C, von Kossa staining”. I believe there is an error here: it should be “B and D”, right?

Line 643 – Change “in the southeastern hillside from” to “on the southeastern hillside of”.

Line 645 – Change “it is likely that the B. lulai populations” to “it is likely that B. lulai populations”.

Line 652 – Exclude the word “with”.

Lines 662-663 – Change “of B. lulai” to “of one individual of B. lulai”.

Lines 664-665 – Change “previously studied parasites [58] and found under the same conditions” to “previously recorded parasites found under similar conditions [58]”.

Line 665 – Change “without apparent morphological structures and in a young stage” to “without apparent diagnostic morphological structures and at a young stage”.

Lines 667 – Add the word “presumably” before “makes”.

Lines 668-670 – This sentence starting with “Furthermore…” is confusing and it is not clear what it is trying to say. I suggest deleting this sentence and replacing it with this: “Indeed, Ophiotaenia spp. are generalist parasites that use not only amphibians, but also snakes and turtles as hosts [58].”

Line 671 – Change “the Ophiotaenia parasite was found in” to “Ophiotaenia parasites have been found in”.

Lines 671-672 – I think the record of Ophiotaenia in B. pernix by Ribeiro et al. (2014) should be cited here too.

Line 676 – Change “of the parasite at the gular region” to “of the parasites in the gular region”.

Line 692 – Change “even with the small distribution.” To “even considering its small distribution area.”

Lines 693 and 699-700 – Change “From the 20 remaining species” to “Of the 20 remaining species”.

Line 706 – Add the word “states” after “Santa Catarina”.

Table 7 – In the line of B. pombali, change 1300 to 1,300.

Line 735 – After “digital reduction”, I suggest adding “(President Lula lost the little finger on his left hand in an accident)”. I think an explanation is appropriate, since this pun may not be obvious for some non-Brazilian readers.

Line 749 – Change “support our interpretation” to “reinforce our interpretation”.

Line 759 – Change “vertebrae” to “vertebra”.

Lines 783 and 784 - Change “other words” to “other works”.

Line 784 – Change “described to as” to “described as”.

Line 819 – Change “did not note this feature” to “did not mention this feature”.

Line 825 – Change “latter it was demonstrated” to “it was later demonstrated”.

Line 862 – Change “would be as an example” to “would be an example”.

Line 883 – Change “expanding the knowledge of intraspecific variation” to “the expanding knowledge on intraspecific variation”.

Line 907 – Change “increasingly” to “increasing”.

Line 907 – Change “which we have sometimes achieved” to “which is sometimes achieved”.

Line 926 – Change “mountain top” to “mountaintop”.

Line 932 – Change “vegetative front” to “arboreal stratum”.

Line 964 – Change “*Brachycephalus* ’s geographic range” to “the geographic range of species in the genus *Brachycephalus* ”.

Line 970 – Change “propose” to “hypothesize”.

Line 971 – Change “upslope” to “upward”.

Line 976 – Change “forest” to “vegetation”.

Line 983 – Add “to” after “according”.

Line 986 – Change “to Santa Catarina” to “to the state of Santa Catarina”.

Line 999 – Change “possibly” to “possible”.

Line 1000 – Change “conservation target species and key areas” to “target species and key areas for conservation”

Line 1034 – Change “and, mainly, take” to “and, especially, change the status of”.

Line 1035 – Delete “the”.

Line 1036 – Add the word “anthropic” before “impacts”.

Line 1043 -Change “and confirm” to “to confirm”.

Line 1069 – Add “may have” before “occurred”.

Line 1079 – “the lacks of” to “the apparent lack of”.

Line 1092 - Change “improve” to “improved”.

Line 1383 – Put the name “Pleroma” in italics.

7. PLOS authors have the option to publish the peer review history of their article (what does this mean? ). If published, this will include your full peer review and any attached files.

**Do you want your identity to be public for this peer review?** For information about this choice, including consent withdrawal, please see our Privacy Policy .

Reviewer #1: No

---

## [Author Response · Author response to Decision Letter 2]

20 Sep 2025

Curitiba, September 20, 2025

Dr. Neelesh Dahanukar,

Academic Editor

PLOS ONE

Dear Dr. Dahanukar,

We thank the reviewer for their thorough analysis of the entire manuscript, down to its smallest details. We have accepted all suggestions, except for the one regarding a more detailed explanation of the etymology of the new species. This is because we have chosen to simplify the etymology, and therefore the suggested addition is no longer necessary.

Despite fully accepting all other points, we have still chosen to respond to each comment individually, in order to highlight the breadth of the review.

Below, you will find our responses to the reviewer’s comments, presented in bold for ease of reading. We have numbered the comments sequentially ourselves, so that we can refer back to our responses in the case of recurring issues.

Kind regards,

Marcio Pie

Reply to Reviewer’s comments

1) As stated in my first review of this ms, I still discourage the continuing use of the B. didactylus group as a taxonomic entity. The authors have responded to this critic by arguing that the B. didactylus group is “a very well-defined group of species based on morphological characteristics”. Species groups are treated in current zoological systematics as monophyletic groupings of species rather than purely phenetic ones (unless when the phylogenetic relationships among species are not known). And it is now widely known that the characteristics uniting the species in the B. didactylus group (including B. clarissae, which does not share all of them) are plesiomorphic for the genus. Yet, even after addressing the non-monophyly of the B. didactylus group in the introduction, the authors continue to refer to it throughout the text as if it was a taxonomic entity equivalent to the other two species groups in the genus (for which there is strong evidence of monophyly in the literature). I consider this misleading, and continue to advocate for the use of the term “flea toads” instead, which unlike the “species group” category, corresponds to a morphotype rather than a clade. Indeed, the term is adopted in the work of Toledo et al. (2024), which included B. clarissae among the species with that morphotype (as mentioned by the authors in their response to my previous review).

The authors also seem to use the B. ephippium group in the sense of Ribeiro et al. 2015 (i.e. including both the ephippium and vertebralis lineages proposed by Condez et al. 2020) rather than in the sense of Folly et al. 2022 (i.e. excluding the vertebralis lineage, which is considered as a separate species group). The latter definition seems more conservative, since not all phylogenetic analyses published so far presented strong support for the monophyly of the B. ephippium group sensu Ribeiro et al. 2015. Anyway, since there are currently two different definitions of this group being used in the literature, the authors should explicitly state in the text which one they are using.

Authors: DONE.

We have started using “flea toads” instead of the “B. didactylus group”. In the case of the B. ephippium group, we have taken the liberty of continuing to adopt the original proposal by Ribeiro et al. (2015). Since the new species described belongs to the B. pernix group, it seems more didactic and understandable to diagnose it in relation to the B. ephippium group lato sensu.

Perhaps the reviewer did not notice that we made explicit which proposal we adopted regarding the B. ephippium group in the following passage of the diagnosis: “*Brachycephalus* lulai sp. nov. can be distinguished from the seven flea toad species (…) by having bufoniform body shape instead leptodactyliform body shape (…) and from 15 species of the B. ephippium group (sensu Ribeiro et al. [13]; …”

2) I believe that in all tables, as well as in table and figure legends that feature the name “B. lulai”, it should be followed by “sp. nov.”

Authors: DONE.

However, we standardized by adding “sp. nov.” to every mention of B. lulai.

3) Line 4 – I suggest adding the word “proposed” before “conservation measures”.

Authors: DONE.

4) Line 26 – Exclude the hyphen in “currently-recognized”.

Authors: DONE.

5) Line 29 – Change “call comparisons” to “acoustic comparisons”.

Authors: DONE.

6) Line 52 – “The species are diurnal”. This should be changed to “…mostly diurnal”, since some of the cryptically colored “flea toads” often present nocturnal activity.

Authors: DONE.

7) Line 98 – Change “also inhabited de occurrence of” to “also inhabited by”.

Authors: DONE.

8) Line 102 – Change “occurs B. leopardus, a species known from two localities” to “B. leopardus occurs, being known from two localities”.

Authors: DONE.

9) Lines 106-107 – Change “annual mean of temperature” to “mean annual temperature”.

Authors: DONE.

10) Line 306 – Change “we did not mapped” to “we have not mapped”.

Authors: DONE.

11) Line 468 – “discrete irregular greenish or brown spots”. Do you mean “discrete” or “discreet”? These are different terms.

Authors: DONE

We did mean “discrete”.

12) Line 479 – Add the word “species” after “Sexually dimorphic”.

Authors: DONE.

13) Line 483 – Change “green spots” to “green patches”.

Authors: We referred to the spots of B. lulai as “spots” seven times, and never as “patches.” Replacing only one instance of “spots” with “patches” could introduce ambiguity in the interpretation of the color pattern. Moreover, replacing all seven occurrences of “spots” with “patches” did not seem appropriate to us. That said, we remain open to adjusting the terminology used to describe the markings, depending on the reviewer’ suggestions.

14) Line 486 – Change “becoming dark, brownish black” to “becoming dark gray”.

Authors: DONE.

15) Line 549 – “in B and C, von Kossa staining”. I believe there is an error here: it should be “B and D”, right?

Authors: DONE

Exactly. Thank you very much for noticing this error.

16) Line 643 – Change “in the southeastern hillside from” to “on the southeastern hillside of”.

Authors: DONE.

17) Line 645 – Change “it is likely that the B. lulai populations” to “it is likely that B. lulai populations”.

Authors: DONE.

18) Line 652 – Exclude the word “with”.

Authors: DONE.

19) Lines 662-663 – Change “of B. lulai” to “of one individual of B. lulai”.

Authors: DONE.

20) Lines 664-665 – Change “previously studied parasites [58] and found under the same conditions” to “previously recorded parasites found under similar conditions [58]”.

Authors: DONE.

21) Line 665 – Change “without apparent morphological structures and in a young stage” to “without apparent diagnostic morphological structures and at a young stage”.

Authors: DONE.

22) Lines 667 – Add the word “presumably” before “makes”.

Authors: DONE.

23) Lines 668-670 – This sentence starting with “Furthermore…” is confusing and it is not clear what it is trying to say. I suggest deleting this sentence and replacing it with this: “Indeed, Ophiotaenia spp. are generalist parasites that use not only amphibians, but also snakes and turtles as hosts [58].”

Authors: DONE.

24) Line 671 – Change “the Ophiotaenia parasite was found in” to “Ophiotaenia parasites have been found in”.

Authors: DONE.

25) Lines 671-672 – I think the record of Ophiotaenia in B. pernix by Ribeiro et al. (2014) should be cited here too.

Authors: DONE.

Perhaps the reviewer did not notice, but the work by Ribeiro et al. (2014) was indeed cited [...a proteocephalid tapeworm, due to the morphological similarity with previously recorded parasites found under similar conditions [58],...].

26) Line 676 – Change “of the parasite at the gular region” to “of the parasites in the gular region”.

Authors: DONE.

27) Line 692 – Change “even with the small distribution.” To “even considering its small distribution area.”

Authors: DONE.

28) Lines 693 and 699-700 – Change “From the 20 remaining species” to “Of the 20 remaining species”.

Authors: DONE.

29) Line 706 – Add the word “states” after “Santa Catarina”.

Authors: DONE.

30) Table 7 – In the line of B. pombali, change 1300 to 1,300.

Authors: DONE.

31) Line 735 – After “digital reduction”, I suggest adding “(President Lula lost the little finger on his left hand in an accident)”. I think an explanation is appropriate, since this pun may not be obvious for some non-Brazilian readers.

Authors: As mentioned above, we opted to simplify the etymology, so the reviewer’ suggestion is no longer applicable.

32) Line 749 – Change “support our interpretation” to “reinforce our interpretation”.

Authors: DONE.

33) Line 759 – Change “vertebrae” to “vertebra”.

Authors: DONE.

34) Lines 783 and 784 - Change “other words” to “other works”.

Authors: DONE.

35) Line 784 – Change “described to as” to “described as”.

Authors: DONE.

36) Line 819 – Change “did not note this feature” to “did not mention this feature”.

Authors: DONE.

37) Line 825 – Change “latter it was demonstrated” to “it was later demonstrated”.

Authors: DONE.

38) Line 862 – Change “would be as an example” to “would be an example”.

Authors: DONE.

39) Line 883 – Change “expanding the knowledge of intraspecific variation” to “the expanding knowledge on intraspecific variation”.

Authors: DONE.

40) Line 907 – Change “increasingly” to “increasing”.

Authors: DONE.

41) Line 907 – Change “which we have sometimes achieved” to “which is sometimes achieved”.

Authors: DONE.

42) Line 926 – Change “mountain top” to “mountaintop”.

Authors: DONE.

43) Line 932 – Change “vegetative front” to “arboreal stratum”.

Authors: DONE.

44) Line 964 – Change “*Brachycephalus* ’s geographic range” to “the geographic range of species in the genus *Brachycephalus* ”.

Authors: DONE.

45) Line 970 – Change “propose” to “hypothesize”.

Authors: DONE.

46) Line 971 – Change “upslope” to “upward”.

Authors: DONE.

47) Line 976 – Change “forest” to “vegetation”.

Authors: In this case, we are referring to the name of a forest type, based on the literature (highland dense ombrophylous forest = Floresta Ombrófila Densa Altomontana), so replacing “forest” with “vegetation” is not appropriate.

48) Line 983 – Add “to” after “according”.

Authors: DONE.

49) Line 986 – Change “to Santa Catarina” to “to the state of Santa Catarina”.

Authors: DONE.

50) Line 999 – Change “possibly” to “possible”.

Authors: DONE.

51) Line 1000 – Change “conservation target species and key areas” to “target species and key areas for conservation”

Authors: DONE.

52) Line 1034 – Change “and, mainly, take” to “and, especially, change the status of”.

Authors: DONE.

53) Line 1035 – Delete “the”.

Authors: DONE.

54) Line 1036 – Add the word “anthropic” before “impacts”.

Authors: DONE.

55) Line 1043 -Change “and confirm” to “to confirm”.

Authors: DONE.

56) Line 1069 – Add “may have” before “occurred”.

Authors: DONE.

57) Line 1079 – “the lacks of” to “the apparent lack of”.

Authors: DONE.

58) Line 1092 - Change “improve” to “improved”.

Authors: DONE.

59) Line 1383 – Put the name “Pleroma” in italics.

Authors: DONE.

---

## [Editor Report · Decision Letter 2]

2 Oct 2025

A new species of *Brachycephalus* (Anura: Brachycephalidae) from Serra do Quiriri, northeastern Santa Catarina state, southern Brazil, with a review of the diagnosis among species of the B. pernix group and proposed conservation measures

PONE-D-24-43253R2

Dear Dr. Pie,

We’re pleased to inform you that your manuscript has been judged scientifically suitable for publication and will be formally accepted for publication once it meets all outstanding technical requirements.

Kind regards,

Neelesh Dahanukar, Ph.D.

Academic Editor

PLOS ONE

---

## [Editor Report · Acceptance letter]

PONE-D-24-43253R2

PLOS ONE

Dear Dr. Pie,

I'm pleased to inform you that your manuscript has been deemed suitable for publication in PLOS ONE. Congratulations! Your manuscript is now being handed over to our production team.

Kind regards,

on behalf of

Dr. Neelesh Dahanukar

Academic Editor

PLOS ONE